# FEDSGM: A UNIFIED FRAMEWORK FOR CONSTRAINT AWARE, BIDIRECTIONALLY COMPRESSED, MULTI-STEP FEDERATED OPTIMIZATION

## ABSTRACT

We introduce FEDSGM, a unified framework for federated constrained optimization that addresses four major challenges in federated learning (FL): functional constraints, communication bottlenecks, local updates, and partial client participation. Building on the switching gradient method, FEDSGM provides projection-free, primal-only updates, avoiding expensive dual-variable tuning or inner solvers. To handle communication limits, FEDSGM incorporates bidirectional error feedback, correcting the bias introduced by compression while explicitly understanding the interaction between compression noise and multi-step local updates. We derive convergence guarantees showing that the averaged iterate achieves the canonical $\mathcal{O}(1/\sqrt{T})$ rate, with additional high-probability bounds that decouple optimization progress from sampling noise due to partial participation. Additionally, we introduce a soft switching version of FEDSGM to stabilize updates near the feasibility boundary. To our knowledge, FEDSGM is the first framework to unify functional constraints, compression, multiple local updates, and partial client participation, establishing a theoretically grounded foundation for constrained federated learning. Finally, we validate the theoretical guarantees of FEDSGM via experimentation on Neyman–Pearson classification and constrained Markov decision process (CMDP) tasks.

## 1 INTRODUCTION

We study federated constrained optimization problems of the form

$$w^* = \underset{w \in \mathcal{X}}{\arg\min} \left\{ f(w) := \tfrac{1}{n} \sum_{j=1}^{n} f_j(w) \quad \text{s.t.} \quad g(w) := \tfrac{1}{n} \sum_{j=1}^{n} g_j(w) \leq 0 \right\}, \tag{1}$$

where $\mathcal{X} \subseteq \mathbb{R}^d$ is a compact, convex set, $f_j : \mathbb{R}^d \to \mathbb{R}$ and $g_j : \mathbb{R}^d \to \mathbb{R}$ are the local objective and constraint functions on client $j \in [n]$. This setting captures a wide range of real-world federated learning (FL) applications (Konečný et al., 2016; Kairouz et al., 2021; McMahan et al., 2017) such as mobile keyboards, autonomous fleets, or battery management systems (Kalashnikov et al., 2018; Brohan et al., 2022; Zhang et al., 2023), where privacy, latency, and safety requirements prohibit centralizing data. The constraint $g(w) \leq 0$ encodes critical feasibility criteria such as fairness mandates (Agarwal et al., 2018), energy budgets, or safety margins (Zhang et al., 2023).

Addressing 1 in the FL setting requires navigating **four tightly coupled challenges**, whose combined effect fundamentally limits existing algorithmic approaches.

• **Functional constraints.** Many federated tasks require guarantees beyond minimizing loss, such as fairness across subpopulations, risk limits in financial models, or physical safety constraints in batteries (e.g., maximum temperature rise). Enforcing these constraints demands algorithms that ensure feasibility without expensive projections or nested inner solves.

• **Severe bandwidth limits.** FL often operates over wireless networks, where transmitting millions of parameters per round is infeasible. Communication-efficient updates rely on contractive compressors such as $\text{Top}-K$, $\text{Rand}-K$, or quantization (Alistarh et al., 2017; Stich & Karimireddy, 2019).

These introduce bias, which must be corrected via error feedback (EF) (Stich & Karimireddy, 2019; Fatkhullin et al., 2021) to guarantee convergence.

• **Local updates & Partial participation.** Edge devices vary significantly in terms of FLOPS, memory, and energy capacity. Allowing each client to perform multiple local steps ($E > 1$) between communications improves utilization and reduces latency (Khaled et al., 2019; Li et al., 2020a;a), but introduces drift between local and global iterates, complicating both theory and practice. Moreover, in practice, only a small subset $m < n$ of clients are active each round due to device availability, battery levels, or network schedules. This stochastic sampling introduces additional variance (Li et al., 2020b), degrading convergence guarantees.

**Limitations of existing approaches.** Existing methods address only subsets of these challenges in FL, but not all at once. Projection-based approaches, such as constrained FEDAVG (He et al., 2024) and AL/ADMM-type methods (Nemirovski, 2004; Hamedani & Aybat, 2021; Müller et al., 2024; Ding et al., 2023; Kim et al., 2024), can guarantee feasibility but typically assume full participation and uncompressed updates, while requiring dual variable tuning, penalty scheduling, or costly inner solvers that burden edge devices. Compression methods with EF (EF-SGD, SAFE-EF) (Stich & Karimireddy, 2019; Alistarh et al., 2017; Fatkhullin et al., 2021; Islamov et al., 2025) provide robustness to biased compression, yet their analyses do not accommodate multiple local updates ($E > 1$) or client sampling. Similarly, FEDAVG and its variants (Khaled et al., 2019; Karimireddy et al., 2020; Li et al., 2020a) are designed to handle heterogeneity, but they do not provide feasibility guarantees under functional constraints. Finally, switching (sub-)gradient method (SGM) (Polyak, 1967; Lan & Zhou, 2020; Upadhyay et al., 2025) offers primal-only, projection-free updates by alternating between $\nabla f$ and $\nabla g$, achieving the optimal $\mathcal{O}(\epsilon^{-2})$ rate for convex non-smooth problems. However, existing SGM formulations do not simultaneously address the challenges above. Additional discussion of related work is provided in the Appendix G.

To address these limitations, our goal is a method that is duality-free, projection-free, and communication efficient, yet admits clean convergence guarantees under convexity.

**Contributions: FEDSGM**

1. **Duality-free FL with general non-smooth convex functional constraints.** We extend the projection-free, primal-only concept of SGM to FL with convex functional constraints, avoiding the dual-variable tuning and penalty scheduling required by AL/ADMM-type methods. This keeps per-round computation light while still certifying feasibility at the averaged iterate.

2. **Compression with EF under switching.** We extend bi-directional compression with EF to FEDSGM, explicitly analyzing the effect of compression, switching, and local updates.

3. **Multi-step local updates: drift analysis.** We provide the first convergence analysis of SGM in federated learning with $E > 1$ local steps, showing that the averaged iterate $\bar{w}$ achieves

$$\max\left\{ f(\bar{w}) - f(w^*), g(\bar{w}) \right\} = \mathcal{O}\left( \frac{DG\sqrt{E}}{\sqrt{T}} \cdot \Gamma(q, q_0) \right),$$

where $D = \|w_0 - w^*\|$, $T$ is the number of rounds, and $\Gamma(q, q_0)$ captures the effect of uplink and downlink compression accuracies $q$ and $q_0$ such that $\Gamma = 1$ means no compression. This result isolates the impact of drift while preserving the canonical $1/\sqrt{T}$ rate.

4. **Partial participation with high-probability guarantees.** When only $m(< n)$ clients participate, we prove that with probability at least $(1 - \delta)$ for probability tolerance, $\delta \in (0, 1)$,

$$(i) f(\bar{w}) - f(w^*) \le \epsilon + 2\sigma\sqrt{\frac{2}{m} \log\left(\frac{6T}{\delta}\right)} \quad (ii) g(\bar{w}) \le \epsilon + 2\sigma\sqrt{\frac{2}{m} \log\left(\frac{6T}{\delta}\right)},$$

cleanly decoupling optimization and estimation errors.

5. **Soft switching.** In addition to hard switching, we adopt soft switching inspired by Upadhyay et al. (2025) to interpolate between $\nabla f$ and $\nabla g$ based on the magnitude of constraint violation. This mitigates instability near the feasibility boundary under noise and heterogeneity, while preserving the projection-free, primal-only structure.

To the best of our knowledge, FEDSGM is the first unified framework that simultaneously handles constraints, compression, local multi-step updates, and partial participation in federated learning, while delivering lightweight, primal-only updates with provable convergence guarantees, establishing a principled foundation for reliable and communication-efficient constrained FL at scale.

## 2 PROBLEM SETUP AND PRELIMINARIES

**Notation** We denote the set of communication rounds as $[T] := \{0, \ldots, T-1\}$, and the set of clients as $[n] := \{1, \ldots, n\}$. $|\mathcal{A}|$ denotes cardinality of set $\mathcal{A}$. At round $t$, a subset of clients $\mathcal{S}_t \subseteq [n]$ participates, where $|\mathcal{S}_t| = m$. Let $\mathbf{Id}(x) = x$ be the identity mapping, $\Pi_{\mathcal{X}}(\cdot)$ be the projection mapping to $\mathcal{X}$, and $[z]_+ := \max\{0, z\}$. Additionally, $\mathbb{1}_{\{\cdot\}}$ denotes the standard binary indicator function.

We consider the constrained optimization problem defined in eq. (1). For this standard FL setting, we make the following assumptions:

**Assumption 1.** *Each local objective $f_j$ and constraint $g_j$ is convex and $G-$Lipschitz continuous on $\mathcal{X}$, i.e., for all $w_1, w_2 \in \mathcal{X}$, $f_j(w_1) \geq f_j(w_2) + \langle \nabla f_j(w_2), w_1 - w_2 \rangle$ and $|f_j(w_1) - f_j(w_2)| \leq G\|w_1 - w_2\|$; similarly for $g_j$. Consequently, the global functions $f$ and $g$ are also convex and $G-$Lipschitz.*

**Assumption 2.** *We have that $\mathcal{X} \subseteq \mathbb{R}^d$ is convex and compact, meaning the set is bounded, thus, $diam(\mathcal{X}) := \sup_{w_1, w_2 \in \mathcal{X}} \|w_1 - w_2\| \leq D$.*

In the standard FL setting, communication happens between the clients and the server, and data never leaves the clients. After $E$ local updates, clients traditionally transmit their accumulated gradients or model updates to the server. However, the high dimensionality of these updates creates a severe communication bottleneck. A common solution is to apply compression, reducing bandwidth at the cost of introducing bias. We focus on the widely studied class of contractive compressors, examples include sparsification methods such as Top$-K$, which retain only the $K$ entries of largest magnitude, as well as various quantization schemes (Stich et al., 2018; Alistarh et al., 2018; Wen et al., 2017; Horvóth et al., 2022; Islamov et al., 2024a; Beznosikov et al., 2023). In this work, we consider bi-directional compression, applying compression both on client-to-server (uplink) and server-to-client (downlink) communications.

**Assumption 3.** *The compression operators $\mathcal{C}_0$ (server) and $\mathcal{C}_j$ (client $j$) are contractive in expectation: $\mathbb{E}[\|\mathcal{C}_0(w) - w\|^2] \leq (1 - q_0)\|w\|^2$, $\mathbb{E}[\|\mathcal{C}_j(w) - w\|^2] \leq (1 - q)\|w\|^2$, for all $w \in \mathbb{R}^d$, with $q_0, q \in (0, 1]$. Expectations are taken over the randomness of the respective compression mechanisms. It holds trivially for deterministic compressors such as Top$-K$ compressor with $q = K/d$.*

To mitigate the bias introduced by compression (Seide et al., 2014; Karimireddy et al., 2019; Islamov et al., 2025), we employ error feedback on the uplink path, where each client $j$ maintains a residual vector $e_j^t$ to recover information lost during compression using an EF14 style update (Seide et al., 2014). For the downlink path, we adopt a primal EF21 variant (Gruntkowska et al., 2023; Islamov et al., 2025) that compresses the difference between successive model parameters, enabling efficient bidirectional communication while progressively correcting uplink bias.

Since our goal is constrained optimization, each client evaluates its local constraint using the global model before training begins. The server aggregates these evaluations and determines whether clients should update using the gradient of the objective or that of the constraint. In practice, partial participation is inevitable due to device availability, energy constraints, or connectivity issues. At each round $t$, only a subset $\mathcal{S}_t \subseteq [n]$ of $m$ clients participates, making it impossible to compute the exact $g(\cdot)$. Instead, we estimate, $\hat{G}(w) := \frac{1}{m} \sum_{j \in \mathcal{S}_t} g_j(w)$.

**Assumption 4.** *The constraint evaluation gap, $\hat{G}(w_t) - g(w_t)$ is $\sigma^2/m-$sub-Gaussian for all $t \in [T]$. Additionally, constraint evaluation and gradient computation are independent.*

Our goal is to compute an $\epsilon$-solution $\bar{w}$, i.e., $f(\bar{w}) - f(w^*) \leq \epsilon$ and $g(\bar{w}) \leq \epsilon$, while simultaneously handling (a) local updates under client drift, (b) bi-directional compression with error feedback, and (c) noisy constraint aggregation due to partial participation.

## 3 FEDSGM

In this section, we present FEDSGM, which combines **Fed**erated Learning with **S**witching **G**radient **M**ethod and accommodates (i) both full and partial client participation, (ii) hard and soft switching between objective and constraint minimization, and (iii) error feedback and compression in both uplink and downlink communication. We provide formal convergence guarantees for both hard and soft switching regimes, under appropriate assumptions.

### 3.1 HARD SWITCHING

Under hard switching, each client optimizes either the objective or the constraint, based on whether the average constraint violation $\hat{G}(w_t)$ is within a pre-specified tolerance $\epsilon$. It is formally presented as,

$$\nu_{j,\tau}^t = \nabla f_j(w_{j,\tau}^t)\, \mathbb{1}\{\hat{G}(w_t) \leq \epsilon\} + \nabla g_j(w_{j,\tau}^t)\, \mathbb{1}\{\hat{G}(w_t) > \epsilon\}.$$

This enforces hard switching between the gradients of the objectives and constraints in response to global feedback. Next, we state the theorem that provides the convergence guarantees for FEDSGM in Algorithm 1.

**Theorem 1.** *(**Hard Switching**) Consider the problem in 1 and Algorithm 1. Let $D := \|w_0 - w^*\|$ and define $\mathcal{A} := \{t \in [T] \mid \hat{G}(w_t) \leq \epsilon\}, \quad \bar{w} := \frac{1}{|\mathcal{A}|}\sum_{t\in\mathcal{A}} w_t$.*
*• **Full Participation** ($m = n$): Then under Assumption 1 and Assumption 3 (only for compression), let $\Gamma = 2E^2 + \underbrace{\frac{2E\sqrt{1-q}}{q} + \frac{4E\sqrt{10(1-q_0)}}{q_0 q}}_{\text{only for bidirectional compression w/ EF}}$, where $q$ and $q_0$ are the client's and server's compression accuracies. Now, set the constraint threshold and step size as $\epsilon = \sqrt{\frac{2D^2G^2\Gamma}{ET}}$ and $\eta = \sqrt{\frac{D^2}{2G^2ET\Gamma}}$, respectively then $\mathcal{A}$ is nonempty, $\bar{w}$ is well-defined, and $\bar{w}$ is an $\epsilon$-solution of 1, i.e. $(i) f(\bar{w}) - f(w^*) \leq \epsilon$ and $(ii) g(\bar{w}) - g(w^*) \leq \epsilon$.*
*• **Partial Participation** ($m < n$): Let Assumptions 1, 2, and 4 hold, and distinct clients are sampled uniformly at random. Then, for deterministic compressors $\{\mathcal{C}_j\}_{j=0}^n$, with probability tolerance $\delta \in (0,1)$, let $\Gamma = 2E^2 + \underbrace{16E\frac{n}{m}\frac{\sqrt{10(1-q)(1-q_0)}}{q_0 q^2} + 8E\frac{\sqrt{10(1-q_0)}}{q_0 q} + \frac{20E}{q^2} + \frac{n}{m}\frac{4E\sqrt{10(1-q)}}{q^2}}_{\text{only for bidirectional compression w/ EF}}$, where $q$ and $q_0$ are the client's and server's compression accuracies. Now, set the constraint threshold and step size as $\epsilon = \sqrt{\frac{2D^2G^2\Gamma}{ET}} + \frac{n}{m}\frac{2DG\sqrt{1-q}}{qT} + \frac{4GD}{\sqrt{mT}}\sqrt{2\log\left(\frac{3}{\delta}\right)} + 2\sigma\sqrt{\frac{2}{m}\log\left(\frac{6T}{\delta}\right)}$ and $\eta = \sqrt{\frac{D^2}{2G^2ET\Gamma}}$, then with probability at least $(1-\delta)$, $\mathcal{A}$ is nonempty, $\bar{w}$ is well-defined and $(i) f(\bar{w}) - f(w^*) \leq \epsilon$, $(ii) g(\bar{w}) \leq \epsilon + \sqrt{\frac{3\sigma^2}{m}\log\left(\frac{T}{\delta}\right)}$.*

The details of the proof are present in the Appendix C. Now, we discuss the implications of the statement of Theorem 1 as well as the main challenges in the convergence proof.

**Centralized with no compression**, i.e., $n = 1, q_0 = q = 1, E = 1$: In this case, we can infer from Theorem 1, that the rate we receive is $\mathcal{O}\left(DG/\sqrt{T}\right)$, corresponding to the rates present in Nesterov et al. (2018); Lan & Zhou (2020); Upadhyay et al. (2025).

**FedSGM w/ full participation w/o compression**, i.e., $m = n, q_0 = q = 1$: In this case, we can infer from Theorem 1 that the sub-optimality gap and constraint value diminish at the rate of $\mathcal{O}\left(DG\sqrt{E}/\sqrt{T}\right)$. The scaling of $\sqrt{E}$ captures the effect of client-drift in this federated constrained setting, deviating from the previous centralized results.

**Full participation**, i.e., $m = n, E = 1$: In this case, we obtain $f(\bar{w}) - f(w^*) \leq \mathcal{O}\left(\frac{DG}{\sqrt{q_0 qT}}\right)$ and $g(\bar{w}) \leq \mathcal{O}\left(\frac{DG}{\sqrt{q_0 qT}}\right)$, which recovers the rates present in Islamov et al. (2025).

---

**Algorithm 1** FEDSGM (UNIFIED): partial/full participation, hard/soft switching, bi-directional EF compression

---

**Input:** rounds $T$, local steps $E$, # participants $m$, $\eta$, $\epsilon \geq 0$, switching mode mode $\in$ {hard, soft} with soft parameter $\beta$, compression flag comp $\in$ {off, on}, client compressors $\{\mathcal{C}_j\}$, server compressor $\mathcal{C}_0$, initial model $w_0$; set $x_0 \leftarrow w_0$ and $e_j^0 \leftarrow 0$ for all $j$ (if comp = on).

1: **for** $t = 0, \ldots, T-1$ **do**
2:     **Sample** $\mathcal{S}_t \subseteq [n]$ with $|\mathcal{S}_t| = m$
3:     **Constraint query:** $\forall j \in \mathcal{S}_t$ sends scalar $g_j(w_t)$ to server
4:     **Server** sends $\hat{G}(w_t) = \frac{1}{m}\sum_{j \in \mathcal{S}_t} g_j(w_t)$ to all clients
5:     **for all** $j \in \mathcal{S}_t$ **in parallel do**
6:         $w_{j,0}^t \leftarrow w_t$
7:         **for** $\tau = 0$ to $E-1$ **do**
8:             **if** mode = hard **then**
9:                 **if** $\hat{G}(w_t) \leq \epsilon$ **then**
10:                     $\nu_{j,\tau}^t \leftarrow \nabla f_j(w_{j,\tau}^t)$
11:                 **else**
12:                     $\nu_{j,\tau}^t \leftarrow \nabla g_j(w_{j,\tau}^t)$
13:                 **end if**
14:             **else**
15:                 $\sigma_t \leftarrow \sigma_\beta(\hat{G}(w_t) - \epsilon)$;
16:                 $\nu_{j,\tau}^t \leftarrow (1-\sigma_t)\nabla f_j + \sigma_t \nabla g_j$
17:             **end if**
18:             $w_{j,\tau+1}^t \leftarrow w_{j,\tau}^t - \eta \nu_{j,\tau}^t$
19:         **end for**
20:         $\Delta_j^t \leftarrow \frac{w_t - w_{j,E}^t}{\eta} = \sum_{\tau=0}^{E-1} \nu_{j,\tau}^t$
21:         **if** comp = on **then**
22:             $v_j^t \leftarrow \mathcal{C}_j(e_j^t + \Delta_j^t)$;
23:             $e_j^{t+1} \leftarrow e_j^t + \Delta_j^t - v_j^t$
24:             **Client** sends $v_j^t$ to server
25:         **else**
26:             **Client** sends $\Delta_j^t$ to server
27:         **end if**
28:     **end for**
29:     **Server aggregation:**
30:     **if** comp = on **then**
31:         $v_t \leftarrow \frac{1}{m}\sum_{j \in \mathcal{S}_t} v_j^t$;
32:         $x_{t+1} \leftarrow \Pi_\mathcal{X}(x_t - \eta v_t)$
33:         **Downlink EF compression:**
34:         Send $\mathcal{C}_0(x_{t+1} - w_t)$ to all clients
35:         **All clients:**
36:         Set $w_{t+1} \leftarrow w_t + \mathcal{C}_0(x_{t+1} - w_t)$
37:     **else**
38:         $w_{t+1} \leftarrow \Pi_\mathcal{X}\left(w_t - \eta \frac{1}{m}\sum_{j \in \mathcal{S}_t} \Delta_j^t\right)$
39:     **end if**
40: **end for**

---

**Unidirectional uplink compression**, i.e., $q_0 = 1$ and $\mathcal{C}_0 \equiv \mathbf{Id}$: In this scenario, we observe that the sub-optimality gap as well as the constraint violation converge at the rate of $\mathcal{O}\left(\frac{\sqrt{E}}{q\sqrt{T}}\right)$, in the full participation case. In the case of partial participation, we recover the slowdown factor of $\frac{\sqrt{n}}{\sqrt{m}}$ matching the unconstrained case (Li & Li, 2023), in addition to the constraint estimation error.

**No constraint and uplink compression**, i.e., $g_j \equiv 0, \forall j \in [n], E = 1$ and $\mathcal{C}_0 \equiv \mathbf{Id}$: In this case our algorithm reduces to the EF-14 (Seide et al., 2014). It has been analyzed in the non-smooth case for $n = 1$ by Karimireddy et al. (2019), and our rates are consistent with their results.

**Bi-directional compression**: We study the general case where both uplink and downlink transmissions are compressed. Prior work in this setting largely focuses on restricted compressor families such as absolute (Tang et al., 2019) or unbiased operators (Philippenko & Dieuleveut, 2021; Gruntkowska et al., 2023; 2024; Tyurin & Richtarik, 2023) and does not extend to the more flexible, potentially biased operators considered here. Existing studies on biased compression (Beznosikov et al., 2023; Islamov et al., 2024a; Safaryan et al., 2021) address unconstrained optimization, assume full participation, single-step local updates, and omit bidirectional compression. The closest work to ours, Islamov et al. (2025), treats constrained FL with bidirectional compression but still assumes full participation, $E = 1$, and restricts switching to the hard case.

Our framework relaxes all of these assumptions. The convergence result in Theorem 1 holds under partial participation, multiple local updates, and general biased compressors with EF. Moreover, we extend the analysis to both hard and soft switching, with the latter addressed in Section 3.2.

**Principal theoretical hurdles**: This work presents the first principled analysis of constrained federated optimization under the simultaneous presence of multiple local updates, bi-directional compression with error feedback, and partial client participation. At each communication round, we assume

that distinct clients are sampled uniformly at random[1]. The main technical challenge is to control the deviation of the empirical constraint estimate $\hat{G}(w_t)$ from the true global constraint $g(w_t)$ under such random sampling. Under Assumption 4, the deviations $\hat{G}(w_t) - g(w_t)$ are sub-Gaussian with variance proxy $\sigma^2/m$, where $m$ is the number of sampled clients. This bound ensures that $\hat{G}(w_t)$ remains sufficiently close to $g(w_t)$ to make correct switching decisions. In particular, the feasibility set $\mathcal{A} := \{t \in [T] \mid g(w_t) \leq \epsilon\}$ is guaranteed to be nonempty if $\epsilon$ is chosen above the concentration radius implied by the sub-Gaussian tail bound, namely $\epsilon \gtrsim$ optimization error $+ \sigma\sqrt{2\log(6T/\delta)/m^2}$ with probability $(1 - \delta)$ for $\delta \in (0, 1)$. Once such an $\epsilon$ is identified, the rest of the convergence argument proceeds from that point.

## 3.2 SOFT SWITCHING

Hard switching enforces a binary rule: at each round, the algorithm either optimizes the objective or enforces the constraint. While simple, such sharp transitions can be unstable, especially when the iterates lie close to the feasibility boundary. Indeed, the continuous-time limit of SGM dynamics reveals a skew-symmetric component in the Jacobian of the flow (see Proposition 1 in Upadhyay et al. (2025)). This skew-symmetric structure induces rotational behavior, causing oscillations whenever $\hat{G}(w_t)$ fluctuates around the tolerance $\epsilon$. Consequently, hard switching may lead to frequent back-and-forth updates near the constraint boundary.

**Geometric Source of Oscillations.** To better understand this instability, let $a = \nabla f(w)$ and $b = \nabla g(w)$ denote the global objective and constraint gradients. The interaction between $a$ and $b$ is encoded in the skew-symmetric matrix $K_{\text{glob}} := ab^\top - ba^\top$. Whenever $K_{\text{glob}} \neq 0$, the dynamics inherit a rotational component, leading to oscillations in the trajectory. In contrast, when $K_{\text{glob}} = 0$, the gradients are aligned, i.e., $\nabla f(w) = \lambda \nabla g(w)$ for some $\lambda \in \mathbb{R}$, and the dynamics become piecewise linear across feasibility boundaries without intrinsic oscillations.

In federated settings, however, even if $K_{\text{glob}} = 0$, local heterogeneity induces additional skewness. Specifically, define

$$K_{\text{loc}} := \frac{1}{n} \sum_{j=1}^{n} \left( \nabla f_j(w) \nabla g_j(w)^\top - \nabla g_j(w) \nabla f_j(w)^\top \right),$$

with the bound $\|K_{\text{loc}}\|_F \leq \sqrt{2V_f V_g}$, where $V_f := \frac{1}{n}\sum_{j=1}^n \|\nabla f_j - \nabla f\|^2$, $V_g := \frac{1}{n}\sum_{j=1}^n \|\nabla g_j - \nabla g\|^2$ quantify the heterogeneity of objective and constraint gradients (see proof in Appendix C). Thus, instability may arise not only from the global geometry but also from client-level heterogeneity.

**Remark 1.** *Even if the global gradients are perfectly aligned ($K_{glob} = 0$), federated optimization may still exhibit rotational drift due to $K_{loc} \neq 0$. Such client-induced skewness can be mitigated by reducing the number of local steps $E$, by explicitly regularizing client updates, or by tuning the soft switching parameter $\beta$, which dampens the amplification of heterogeneity-driven rotations. In this way, $\beta$ acts as a geometric stabilizer, complementing algorithmic controls on local dynamics.*

These observations motivate a continuous relaxation of the switching rule. Instead of enforcing a hard decision between objective and constraint updates, we introduce soft switching, where the tradeoff is governed continuously through a weighting function. Formally, each client forms a convex combination of the two gradients:

$$\nu_{j,\tau}^t = (1 - \sigma_t)\nabla f_j(w_{j,\tau}^t) + \sigma_t \nabla g_j(w_{j,\tau}^t),$$

where $\sigma_t = \sigma_\beta(\hat{G}(w_t) - \epsilon)$ is the switching weight depending on the global constraint violation. The mapping $\sigma_\beta(\cdot)$ is a activation with sharpness controlled by $\beta$, i.e., a trimmed hinge $\sigma_\beta(x) := \min\{1, [1 + \beta x]_+\} = \text{Proj}_{[0,1]}(1 + \beta x)$ that maps $\mathbb{R}$ to $[0, 1]$.

---

[1] The partial participation analysis remains valid without loss of generality for the case of sampling without replacement, due to Lemma 1.1 in Bardenet & Maillard (2015) (or Theorem 4 in Hoeffding (1963)). These results establish that, for any continuous convex function $f$, the expected value of $f$ applied to the sum of a sample without replacement is never larger than that for a sample with replacement.

[2] The $\log T$ term arises from applying a union bound to ensure the high-probability bound on $\hat{G}(w_t)$ holds uniformly over all $t \in 1, \ldots, T$, a common artifact in stochastic method analyses (Lan & Zhou, 2020). This factor disappears in the full participation case, where $\hat{G}(w_t) = g(w_t)$ and no client sampling variance occurs.

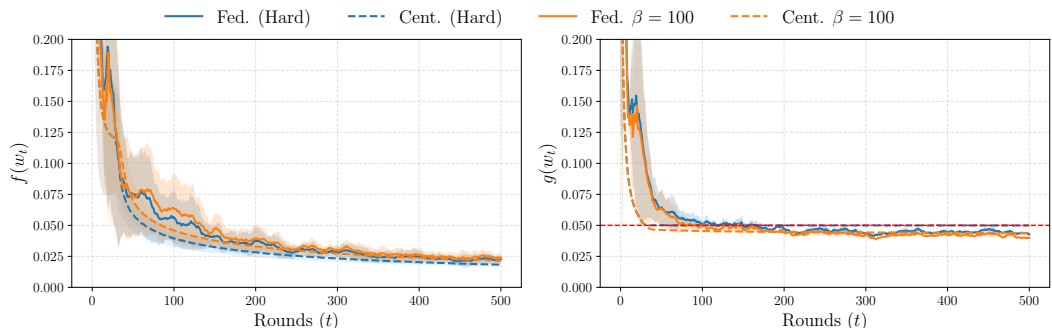

Figure 1: *NP classification:* Progress per round with hard and soft switching.

As $\beta \to \infty$, we observe that it becomes hard switching; otherwise, the two gradients are blended. This removes sharp transitions near the feasibility boundary and dampens oscillatory behavior, while retaining convergence guarantees. We now state the main convergence guarantee for the unified soft switching method under full participation.

**Theorem 2.** *(Soft Switching) Consider the problem in 1 and Algorithm 1, under Assumption 1 and Assumption 3 (only for compression). Let $D := \|w_0 - w^*\|$ and define $\mathcal{A} = \{t \in [T] \mid g(w_t) < \epsilon\}, \bar{w} = \sum_{t \in \mathcal{A}} \alpha_t w_t$, where $\alpha_t = \frac{1 - \sigma_\beta(g(w_t) - \epsilon)}{\sum_{s \in \mathcal{A}} \left[ 1 - \sigma_\beta(g(w_s) - \epsilon) \right]}$. Let $\Gamma = 2E^2 + \underbrace{\frac{2E\sqrt{1-q}}{q} + \frac{4E\sqrt{10(1-q_0)}}{q_0 q}}_{\text{only for bidirectional compression w/ EF}}$, where $q$ and $q_0$ are the client's and server's contraction parame-*

*ters. Now, set the constraint threshold and step size as $\epsilon = \sqrt{\frac{2D^2 G^2 \Gamma}{ET}}, \eta = \sqrt{\frac{D^2}{2G^2 ET\Gamma}}$, and $\beta \geq \frac{2}{\epsilon}$, respectively then $\mathcal{A}$ is nonempty, $\bar{w}$ is well-defined and an $\epsilon$-solution of 1.*

By setting the sharpness parameter $\beta \geq 2/\epsilon$, as stated in Theorem 2, we obtain the same rates as in Theorem 1. This parameter choice ensures a continuous transition but may be overly conservative when $\epsilon$ is very small, effectively approximating a hard switch. Nevertheless, all findings from the hard switching setting, such as convergence rates in special cases, the impact of local updates, and the effects of compression, continue to apply, with the key difference that soft switching reduces the rotational component without altering the bounds' asymptotic order.

## 4 EXPERIMENTS

In this section, we provide experiments on two key applications of constrained optimization. We begin with a classical problem of Neyman-Pearson (NP) classification, and then extend to the highly stochastic deep reinforcement learning setting in constrained Markov decision processes (CMDP).

**NP Classification Task Setting.** NP classification studies the constrained optimization problem in 1, where the goal is to minimize the empirical loss on the majority class while keeping the loss on the minority class below a specific tolerance. For each client $j \in [n]$, the local objective and constraint are defined as $f_j(w) := \frac{1}{m_{j0}} \sum_{x \in \mathcal{D}_j^{(0)}} \phi(w; (x, 0)), g_j(w) := \frac{1}{m_{j1}} \sum_{x \in \mathcal{D}_j^{(1)}} \phi(w; (x, 1))$, where $\mathcal{D}_j^{(0)}$ and $\mathcal{D}_j^{(1)}$ are local class-0 and class-1 datasets of sizes $m_{j0}$ and $m_{j1}$, and $\phi$ is the binary logistic loss. This setup follows the NP paradigm where $f(w)$ enforces the performance on the majority class, while constraint $g(w) \leq \epsilon$ maintains a loss bound on the minority class. Here, we set $n = 20, m = 10, E = 5, T = 500, \text{Top}-K, K/d = 0.1$, and $\epsilon = 0.05$ unless stated otherwise, with experiments conducted on the breast cancer dataset (Wolberg et al., 1993). We report the mean and variance bands over three random seeds. From Figure 1, we observe that both hard and soft switching achieve convergence of the majority-class objective $f(w)$ while satisfying the constraint $g(w)$. Additional experimental details are presented in Appendix F.2.

**Discussion: NP Classification.** To validate the theoretical efficacy of our approach, we conduct experiments by varying local epochs, participation rate, and compression factor. Intuitively, it is apparent that increasing local updates on each client reduces the burden on global communication;

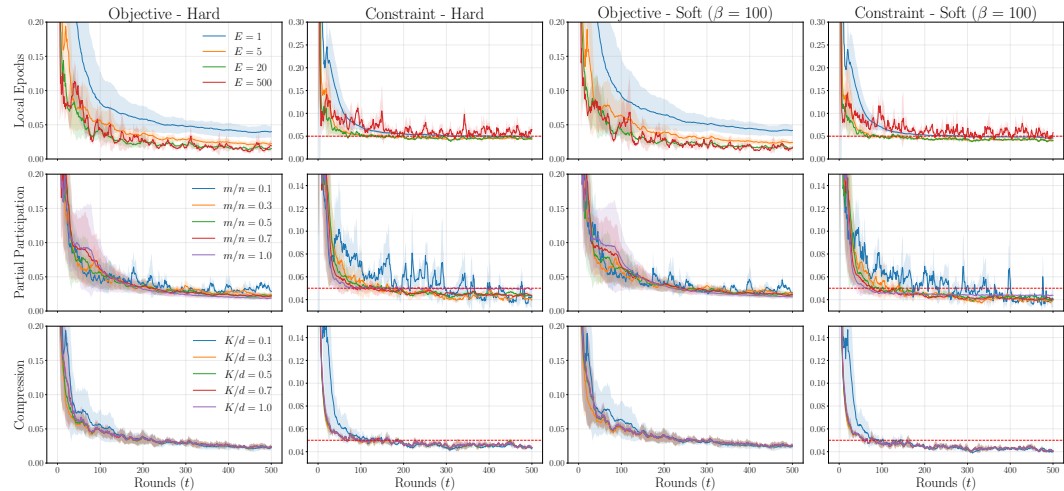

Figure 2: Top: Effect of local updates. Middle: Effect of partial participation. Bottom: Effect of compression on objective and feasibility for hard (left) and soft (right) switching.

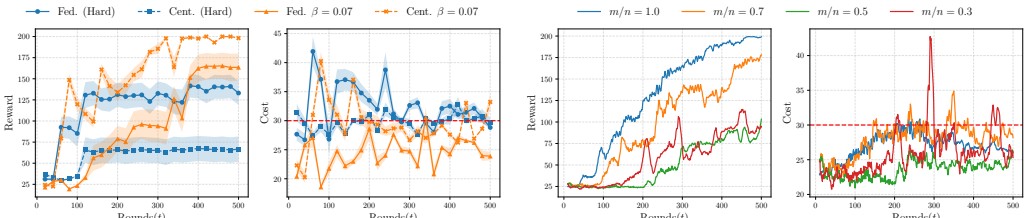

Figure 3: CMDP: Episodic reward vs. Cost. Dotted: centralized, solid: federated with $m/n = 0.7$ and Top$-K$, $K/d = 0.5$. Red dotted line: safety budget of 30.

Figure 4: Effect of partial participation on episodic reward and cost. Total clients $n = 10$, no compression. Red dotted line: safety budget of 30.

however, this can exacerbate the issue of drift and start to harm the learning process (Li et al., 2020b). As we can observe from Figure 2 (top row), increasing the number of local epochs leads to improved convergence of the objective function. However, as $E$ increases, the gains in convergence start to diminish, indicating a trade-off between local computation and global progress. At the same time, we observe that the oscillations around the threshold become more pronounced for increased $E$. It is also noteworthy to mention that we discussed this phenomenon in Section 3.2, where we talk about the source of this geometric oscillation. Next, varying the participation rate shows that higher participation improves convergence for both soft and hard switching, matching the centralized case. Finally, when analyzing the impact of the compression factor $K/d$, we find that even an aggressive compression factor of $0.1$ achieves convergence due to EF, though at a slower rate in Figure 2. However, we observe instability in the constraint dynamics, mitigated by soft switching.

**CMDP Task Setting.** We define our constrained Markov decision process (CMDP) by the tuple $(\mathcal{S}, \mathcal{A}, r, c, P, \gamma, \rho)$, which extends a standard episodic discounted MDP with a cost function $c : \mathcal{S} \times \mathcal{A} \times \mathcal{S} \to \mathbb{R}$ and a corresponding safety constraint. By standard notation, $\mathcal{S}$ and $\mathcal{A}$ are the state and action spaces, respectively, $r : \mathcal{S} \times \mathcal{A} \times \mathcal{S} \to \mathbb{R}$ is the reward function, $P : \mathcal{S} \times \mathcal{A} \times \mathcal{S} \to [0, 1]$ is the stochastic transition kernel, $\gamma \in (0, 1]$ is the discount factor and $\rho : \mathcal{S} \to [0, 1]$ is the initial state distribution. Each of the $n$ clients interacts with a unique CMDP instance, distinguished by a client-specific safety budget $d_i$. The trajectories collected by each client are then used to compute their local constraint and gradient. The objective of each client is defined as $f_i(w) = -\mathbb{E}_{w, P} \left[ \sum_{\tau=0}^{T-1} \gamma^t r(s_\tau, a_\tau, s_{\tau+1}) \right]$ and the constraint as $g_i(w) = \mathbb{E}_{w, P} \left[ \sum_{\tau=0}^{T-1} \gamma^t c(s_\tau, a_\tau, s_{\tau+1}) \right] - d_i$ to solve the global optimization in 1, where $w$ represents the policy parameter. To solve policy optimization, we adopt TRPO (Schulman et al., 2015), which calculates policy gradients in a centralized, unconstrained setting. For our purposes, value function training is subsumed under policy training because value functions are trained on their own

objective regardless of the safety constraint. We use the continuous Cartpole environment (Brockman et al., 2016; Towers et al., 2024), augmented with safety constraints. The agent incurs a cost of 1 at each timestep for violating either of two conditions: the cart entering a prohibited area or the pole's angle exceeding the threshold (Xu et al., 2021). Each client is assigned a different safety budget $d_i \in [25, 35]$ and collects their own data to train. This makes the task extremely heterogeneous, because each data point is sampled from different distribution determined by client drift, stochastic policy and initial state. More details on the specific settings of the experiment can be found in Appendix F.1.

**Discussion: Cartpole.** Figure 3 plots the mean episodic reward and cost, averaged over 5 runs with different random seeds. The shaded area represents 0.2 standard deviation. The results demonstrate that soft switching stabilizes the learning process, leading to a faster convergence for both constraint satisfaction and objective optimization. Also, it is notable that, with hard switching, the FL setting benefited the learning of the objective, despite the complications arising from compression, partial participation, and varying constraint limits. Indeed, the noise and implicit regularization introduced by these factors can smooth the optimization landscape, stabilize the switching mechanism, and encourage exploration. The experimental results in Islamov et al. (2025) and Li et al. (2020b) also describe the effects of these

Table 1: Effect of quantization and Top$-K$ compression in terms of average episodic rewards and costs after 100 and 500 rounds. The arrows indicate that the episodic reward is higher the better, with a maximum of 200, and the episodic cost is lower the better, with a safety margin of 30. The asterisks indicate that the episodic costs next to them exceed the safety margin. The bold numbers indicate the maximum feasible episodic reward across different compression methods in specific rounds.

| Compression | $\hat{f}(w_{100})$ ↑ | $\hat{g}(w_{100})$ ↓ | $\hat{f}(w_{500})$ ↑ | $\hat{g}(w_{500})$ ↓ |
|---|---|---|---|---|
| Centralized | 119.9 | 33.6* | 198.2 | 33.2* |
| No comp. (float32) | **67.2** | 26.9 | **199.4** | 27.6 |
| float16 | 90.5 | 31.2* | 198.8 | 26.9 |
| float8 | 91.4 | 31.4* | 199.1 | 26.5 |
| float4 | 49.1 | 25.2 | 197.2 | 25.6 |
| $K/d = 0.5$ | 46.5 | 23.8 | 131.6 | 24.0 |
| $K/d = 0.25$ | 25.1 | 22.7 | 25.5 | 22.2 |

complications, which do not necessarily hinder the learning of the objective. Figure 4 demonstrates the effect of partial participation under soft switching. Lower participation exhibits greater instability and slower convergence in the learning curve. Table 1 explores the effect of two types of common compression methods under soft switching: Top$-K$ compression and quantization. For this experiment, Top$-K$ compression is simulated by zeroing out the gradient vector with dimensions exceeding $K$ on the order of their magnitude. Instead of converting data types, quantization is simulated by rounding the number that exceeds the precision given by the number of bits to use. It is notable that the algorithm stably satisfies constraints and maximizes episodic reward, and quantization with error feedback had minimal impact in the number of required rounds to maximize episodic rewards while satisfying constraints.

## 5 CONCLUSION, LIMITATIONS, AND FUTURE WORKS

FEDSGM provides a unified algorithmic framework for constrained FL by integrating projection and duality free switching gradient method with multiple local updates, bi-directional compression with EF, and partial participation. Additionally, we employ soft switching that matches the convergence rates of hard switching when $\beta \geq 2/\epsilon$, while ensuring performance stability. To the best of our knowledge, this is the first algorithm to analyze a constrained FL under these regimes. Overall, FEDSGM robustly balances feasibility, client drift, and communication efficiency, and lays the foundation for extensions to other federated paradigms.

Although our work advances the understanding of the constrained FL problem with practical motivations, certain limitations remain. Our theoretical analysis relies on the convexity of the objectives and constraints, though we show the efficacy of the proposed algorithm on RL, which is highly non-convex. Moreover, extending the analysis to a weakly convex setting is the next natural step, potentially using the ideas from Huang & Lin (2023). We focus on gradient descent for simplicity, which may be restrictive in large-scale systems. Though we believe an analogous SGD version

could be analyzed theoretically within the current framework, following approaches in Lan & Zhou (2020); Islamov et al. (2025). Empirically, we address this by evaluating on an RL task, which is highly stochastic in both data sampling and client participation. Future research could extend FEDSGM by incorporating advanced optimizers, privacy-preserving techniques, and asynchronous update schemes. Establishing theoretical lower bounds for constrained FL is another key direction, offering insights to guide the principled design of FEDSGM and beyond.

## REPRODUCIBILITY STATEMENT

We provide the code in a `.zip` file as the supplementary material. In addition, we provide all the details to run the code in the Section 4 and Appendix F.

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

APPENDIX

This appendix provides additional lemmas, proofs, and extended experimental results that complement the main text.

## A  GEOMETRY OF SKEWNESS

For each client $j \in [n]$ let $a_j = \nabla f_j(w)$ and $b_j = \nabla g_j(w)$, and let $a = \frac{1}{n} \sum_{j=1}^n a_j = \nabla f(w)$, $b = \frac{1}{n} \sum_{j=1}^n b_j = \nabla g(w)$.

Define the *global* skew–symmetric matrix $K_{\mathrm{glob}} := ab^\top - ba^\top$ and the mean *local* skew matrix $K_{\mathrm{loc}} := \frac{1}{n} \sum_{j=1}^n \left( a_j b_j^\top - b_j a_j^\top \right)$. Writing the client drifts $\Delta a_j := a_j - a$ and $\Delta b_j := b_j - b$, one obtains the exact decomposition

$$K_{\mathrm{loc}} - K_{\mathrm{glob}} = \frac{1}{n} \sum_{j=1}^n (\Delta a_j \Delta b_j^\top - \Delta b_j \Delta a_j^\top),$$

and the Frobenius-norm bound is just a result of applying the Triangle inequality and the Cauchy-Schwarz inequality,

$$\|K_{\mathrm{loc}} - K_{\mathrm{glob}}\|_F \leq \sqrt{2\,V_f\,V_g}, \qquad V_f := \frac{1}{n}\sum_{j=1}^{n}\|\Delta a_j\|^2, \;\; V_g := \frac{1}{n}\sum_{j=1}^{n}\|\Delta b_j\|^2.$$

## B   ADDITIONAL ASSUMPTIONS: STOCHASTIC FEDSGM

We will use Assumptions 1 and 2 as is. In addition to that, we add a couple more assumptions as follows.

**Assumption 5.**     *1. The noise, $\delta_t^i$ at each user $i \in [n]$ is $\sigma_\xi^2$-sub-Gaussian, i.e., for all $t \geq 0$ and any $\mathcal{F}_{t-1}-$measurable $x \in \mathbb{R}^d$*

$$\mathbb{E}\left[\exp\left(\frac{\|\delta_t^i\|^2}{\sigma_\xi^2}\right)\Big|\mathcal{F}_{t-1}\right] \leq \exp(1),$$

*where $\delta_t^i := \nabla f_i(x_t^i) - \nabla f_i(x_t^i, \xi_t^i)$ for $t \in \mathcal{A}$ or $\delta_t^i := \nabla g_i(x_t^i) - \nabla g_i(x_t^i, \xi_t^i)$ for $t \in \mathcal{B}$.*

*2. The difference between the stochastic and true function evaluation, $(g_i(w, \xi_i) - g(w))$ is $\sigma_b^2/N_b-$ sub-Gaussian. Additionally, constraint evaluation and gradient computation are independent.*

**Proposition 1.** *We define $\tilde{G}(w_t) := \frac{1}{m}\sum_{j \in \mathcal{S}_t} g_j(w_t, \xi_t^j)$. Then the quantity $\tilde{G}(w) - g(w)$ is $\left(\frac{\sigma^2}{m} + \frac{\sigma_b^2}{mN_b}\right)$ -sub-Gaussian.*

*Proof sketch.* We can write

$$\tilde{G}(w) - g(w) = \underbrace{\hat{G}(w) - g(w)}_{\text{client level noise}} + \underbrace{\tilde{G}(w) - \hat{G}(w)}_{\text{stochastic gradient noise}}$$

Now, it only remains to utilize Assumptions 4 and 5(2) in addition to Theorem 2.7 from Rivasplata (2012). □

## C   THEOREMS AND LEMMAS

### C.1   LEMMAS

Here we state some Lemmas that we will use throughout in the proofs of different Theorems.

**Lemma 1.** *Under Assumption 1, for all rounds $t \in [T]$, the following bound holds,*

$$-2\eta\left\langle w_t - w^*, \frac{1}{n}\sum_{j=1}^{n}\sum_{\tau=0}^{E-1}\nu_{j,\tau}^t\right\rangle$$

$$\leq \frac{1}{n}\sum_{j=1}^{n}\sum_{\tau=0}^{E-1}\begin{cases}\frac{\eta}{\alpha}\|w_t - w_{j,\tau}^t\|^2 + \eta\alpha G^2 + 2\eta(f_j(w^*) - f_j(w_{j,\tau}^t)), & \text{if } t \in \mathcal{A},\\[2mm]\frac{\eta}{\alpha}\|w_t - w_{j,\tau}^t\|^2 + \eta\alpha G^2 + 2\eta(g_j(w^*) - g_j(w_{j,\tau}^t)), & \text{if } t \in \mathcal{B}.\end{cases}$$

*Proof.* We now analyze the cross-term in the squared distance recursion,

$$-2\eta\left\langle w_t - w^*, \frac{1}{n}\sum_{j=1}^{n}\sum_{\tau=0}^{E-1}\nu_{j,\tau}^t\right\rangle = \frac{1}{n}\sum_{j=1}^{n}\sum_{\tau=0}^{E-1}\left[\underbrace{-2\eta\left\langle w_t - w_{j,\tau}^t, \nu_{j,\tau}^t\right\rangle}_{TermA}\underbrace{-2\eta\left\langle w_{j,\tau}^t - w^*, \nu_{j,\tau}^t\right\rangle}_{TermB}\right].$$

We handle the second term by applying the convexity of $f_j$ or $g_j$. For $t \in \mathcal{A}$, where updates use $\nabla f_j$ we apply:

$$f_j(w^*) \geq f_j(w_{j,\tau}^t) + \left\langle\nabla f_j(w_{j,\tau}^t), w^* - w_{j,\tau}^t\right\rangle,$$

which implies:

$$-\left\langle w_{j,\tau}^t - w^*, \nabla f_j(w_{j,\tau}^t)\right\rangle \le f_j(w^*) - f_j(w_{j,\tau}^t).$$

Thus,

$$TermB = -2\eta \left\langle w_{j,\tau}^t - w^*, \nabla f_j(w_{j,\tau}^t)\right\rangle \le 2\eta \left(f_j(w^*) - f_j(w_{j,\tau}^t)\right).$$

A similar argument holds for $t \in \mathcal{B}$ with $\nabla g_j$, resulting in $TermB \le 2\eta \left(g_j(w^*) - g_j(w_{j,\tau}^t)\right)$. Again, while upper bounding $TermA$, we need to deal with 2 cases depending on whether $t \in \mathcal{A}$ or $t \in \mathcal{B}$. Firstly, we start with the case where $t \in \mathcal{A}$, for any $\alpha > 0$

$$TermA = -2\eta \left\langle w_t - w_{j,\tau}^t, \nabla f_j(w_{j,\tau}^t)\right\rangle \overset{\text{Young's}}{\le} \frac{\eta}{\alpha}\|w_t - w_{j,\tau}^t\|^2 + \eta\alpha\|\nabla f_j(w_{j,\tau}^t)\|^2$$

$$\overset{G-Lip}{\le} \frac{\eta}{\alpha}\|w_t - w_{j,\tau}^t\|^2 + \eta\alpha G^2.$$

Similarly, for $t \in \mathcal{B}$, we get

$$TermA = -2\eta \left\langle w_t - w_{j,\tau}^t, \nabla f_j(w_{j,\tau}^t)\right\rangle \le \frac{\eta}{\alpha}\|w_t - w_{j,\tau}^t\|^2 + \eta\alpha G^2.$$

Substituting the bounds for both $TermA$ and $TermB$ back into the original expectation furnishes the proof. $\qquad\square$

**Lemma 2.** *(Only Client Sampling Case) Under Assumptions 1, for all rounds $t \in [T]$, the following bound holds,*

$$\sum_{\tau=0}^{E-1}\|w_t - w_{j,\tau}^t\|^2 \le \frac{1}{3}\eta^2 E^3 G^2$$

*Proof.*

$$\|w_t - w_{j,\tau}^t\|^2 = \left\|w_t - \left(w_t - \eta\sum_{k=0}^{\tau-1}\nu_{j,k}^t\right)\right\|^2$$

$$= \eta^2\left\|\sum_{k=0}^{\tau-1}\nu_{j,k}^t\right\|^2$$

$$\overset{Jensen's}{\le} \eta^2\tau\sum_{k=0}^{\tau-1}\|\nu_{j,k}^t\|^2$$

$$\overset{G-Lip}{\le} \eta^2\tau^2 G^2$$

Therefore,

$$\sum_{\tau=0}^{E-1}\|w_t - w_{j,\tau}^t\|^2 \overset{\text{sum of squares}}{\le} \frac{1}{3}\eta^2 E^3 G^2.$$

$\qquad\square$

**Lemma 3.** *(Stochastic Gradient Case) Fix integers $T \ge 1$, $E \ge 1$, number of clients $n \ge 1$, and failure probability $\delta \in (0,1)$. Under Assumptions 1 and 5, we have that, with probability at least $1 - \delta$, the following holds simultaneously for all $t \in [T]$ and all $j \in [n]$:*

$$\sum_{\tau=0}^{E-1}\left\|w_t - w_{j,\tau}^t\right\|^2 \le \eta^2 E^3\left[\frac{2}{3}G^2 + \sigma_\xi^2\left(1 + \log\frac{nTE}{\delta}\right)\right]. \qquad (2)$$

*Proof.*

$$\|w_t - w_{j,\tau}^t\|^2 = \left\|w_t - \left(w_t - \eta\sum_{k=0}^{\tau-1}\nu_{j,k}^t\right)\right\|^2$$

$$= \eta^2 \left\| \sum_{k=0}^{\tau-1} \nu_{j,k}^t \right\|^2$$

$$\leq \eta^2 \tau \sum_{k=0}^{\tau-1} \|\nu_{j,k}^t\|^2.$$

Let $\bar{\nu}_{j,k}^t$ denote the corresponding deterministic gradient (either $\nabla f_j(w_{j,k}^t)$ or $\nabla g_j(w_{j,k}^t)$, depending on whether $t \in \mathcal{A}$ or $t \in \mathcal{B}$), and define the noise $\delta_{j,k}^t := \bar{\nu}_{j,k}^t - \nu_{j,k}^t$. Then

$$\nu_{j,k}^t = \bar{\nu}_{j,k}^t - \delta_{j,k}^t \quad \Rightarrow \quad \|\nu_{j,k}^t\|^2 = \|\bar{\nu}_{j,k}^t - \delta_{j,k}^t\|^2 \leq 2\|\bar{\nu}_{j,k}^t\|^2 + 2\|\delta_{j,k}^t\|^2.$$

By Assumption 1, we have $\|\bar{\nu}_{j,k}^t\| \leq G$, hence $\|\bar{\nu}_{j,k}^t\|^2 \leq G^2$, and so $\|\nu_{j,k}^t\|^2 \leq 2G^2 + 2\|\delta_{j,k}^t\|^2$. Substituting into the previous inequality yields

$$\|w_t - w_{j,\tau}^t\|^2 \leq \eta^2 \tau \sum_{k=0}^{\tau-1} \|\nu_{j,k}^t\|^2$$

$$\leq \eta^2 \tau \sum_{k=0}^{\tau-1} (2G^2 + 2\|\delta_{j,k}^t\|^2)$$

$$= 2\eta^2 \tau^2 G^2 + 2\eta^2 \tau \sum_{k=0}^{\tau-1} \|\delta_{j,k}^t\|^2.$$

Now sum over $\tau = 0, \ldots, E-1$.

$$\sum_{\tau=0}^{E-1} \|w_t - w_{j,\tau}^t\|^2 \leq \sum_{\tau=1}^{E-1} \left( 2\eta^2 \tau^2 G^2 + 2\eta^2 \tau \sum_{k=0}^{\tau-1} \|\delta_{j,k}^t\|^2 \right)$$

$$= 2\eta^2 G^2 \sum_{\tau=1}^{E-1} \tau^2 + 2\eta^2 \sum_{\tau=1}^{E-1} \tau \sum_{k=0}^{\tau-1} \|\delta_{j,k}^t\|^2$$

$$\leq \frac{2}{3} \eta^2 E^3 G^2 + 2\eta^2 \sum_{\tau=1}^{E-1} \tau \sum_{k=0}^{\tau-1} \|\delta_{j,k}^t\|^2.$$

For the second term, exchange the order of summation. For a fixed $k$, the term $\|\delta_{j,k}^t\|^2$ appears in the inner sum for $\tau = k+1, \ldots, E-1$, hence

$$\sum_{\tau=1}^{E-1} \tau \sum_{k=0}^{\tau-1} \|\delta_{j,k}^t\|^2 = \sum_{k=0}^{E-2} \|\delta_{j,k}^t\|^2 \sum_{\tau=k+1}^{E-1} \tau.$$

For $k \in \{0, \ldots, E-2\}$, we have

$$\sum_{\tau=k+1}^{E-1} \tau \leq \sum_{\tau=1}^{E-1} \tau = \frac{E(E-1)}{2} \leq \frac{1}{2} E^2.$$

Therefore

$$\sum_{\tau=1}^{E-1} \tau \sum_{k=0}^{\tau-1} \|\delta_{j,k}^t\|^2 \leq \frac{1}{2} E^2 \sum_{k=0}^{E-1} \|\delta_{j,k}^t\|^2,$$

and thus

$$2\eta^2 \sum_{\tau=1}^{E-1} \tau \sum_{k=0}^{\tau-1} \|\delta_{j,k}^t\|^2 \leq \eta^2 E^2 \sum_{k=0}^{E-1} \|\delta_{j,k}^t\|^2.$$

Combining these pieces, we obtain the deterministic inequality

$$\sum_{\tau=0}^{E-1} \|w_t - w_{j,\tau}^t\|^2 \leq \frac{2}{3} \eta^2 E^3 G^2 + \eta^2 E^2 \sum_{k=0}^{E-1} \|\delta_{j,k}^t\|^2. \tag{3}$$

It remains to control $\sum_{k=0}^{E-1} \|\delta_{j,k}^t\|^2$ in high probability, uniformly over all $t, j, k$. Now, by Assumption 5, there exists $\sigma_\xi^2 > 0$ such that

$$\mathbb{E}\left[\exp\left(\frac{\|\delta_{j,k}^t\|^2}{\sigma_\xi^2}\right) \mid \mathcal{F}_{t,k-1}\right] \le e \quad \text{for all } t, j, k.$$

Define

$$Z_{j,k}^t := \frac{\|\delta_{j,k}^t\|^2}{\sigma_\xi^2}.$$

Then

$$\mathbb{E}\left[e^{Z_{j,k}^t} \mid \mathcal{F}_{t,k-1}\right] \le e.$$

For any $a > 0$, apply the conditional Markov inequality with $Y = e^{Z_{j,k}^t}$ and $c = e^a$:

$$\mathbb{P}\left(e^{Z_{j,k}^t} \ge e^a \mid \mathcal{F}_{t,k-1}\right) \le e^{-a}\mathbb{E}\left[e^{Z_{j,k}^t} \mid \mathcal{F}_{t,k-1}\right] \le e^{-a} \cdot e = e^{1-a}.$$

Since $\{Z_{j,k}^t \ge a\} = \{e^{Z_{j,k}^t} \ge e^a\}$, this gives

$$\mathbb{P}\left(Z_{j,k}^t \ge a \mid \mathcal{F}_{t,k-1}\right) \le e^{1-a}.$$

Taking expectation, $\mathbb{P}\left(Z_{j,k}^t \ge a\right) \le e^{1-a}$. Now set $N := nTE$, $a := 1 + \log\frac{N}{\delta}$. Then

$$e^{1-a} = e^{1-(1+\log\frac{N}{\beta})} = \frac{\delta}{N},$$

so for each fixed triple $(t, j, k)$,

$$\mathbb{P}\left(Z_{j,k}^t \ge a\right) \le \frac{\delta}{N}.$$

Applying a union bound over all $t \in [T]$, $j \in [n]$, and $k \in [E]$ yields with probability at least $1 - \delta$, we have

$$\|\delta_{j,k}^t\|^2 \le \sigma_\xi^2 \left(1 + \log\frac{nTE}{\beta}\right) \quad \text{for all } t, j, k. \tag{4}$$

On this same event, for any fixed $t, j$,

$$\sum_{k=0}^{E-1} \|\delta_{j,k}^t\|^2 \le E\,\sigma_\xi^2 \left(1 + \log\frac{nTE}{\delta}\right).$$

Substituting this into equation 3, we conclude that, with probability at least $1 - \delta$, for all $t \in [T]$ and all $j \in [n]$,

$$\sum_{\tau=0}^{E-1} \|w_t - w_{j,\tau}^t\|^2 \le \eta^2 E^3 \left[\frac{2}{3}G^2 + \sigma_\xi^2 \left(1 + \log\frac{nTE}{\delta}\right)\right].$$

$\square$

**Lemma 4** (Bounding Inner Product-Client Sampling). *Under Assumptions 1, 2, and independence of gradient and constraint evaluation, the following holds for $\delta \in (0, 1)$ with probability at least $1 - \delta$*

$$\sum_{t \in \mathcal{A}} 2\eta \left\langle \frac{1}{n}\sum_{j=1}^n \sum_{\tau=0}^{E-1} \nabla f_j(w_{j,\tau}^t) - \frac{1}{m}\sum_{j \in \mathcal{S}_t}\sum_{\tau=0}^{E-1} \nabla f_j(w_{j,\tau}^t), w_t - w^* \right\rangle \le 4\eta EGD\sqrt{\frac{2T}{m}\log(\frac{1}{\delta})}.$$

$$\sum_{t \in \mathcal{B}} 2\eta \left\langle \frac{1}{n}\sum_{j=1}^n \sum_{\tau=0}^{E-1} \nabla g_j(w_{j,\tau}^t) - \frac{1}{m}\sum_{j \in \mathcal{S}_t}\sum_{\tau=0}^{E-1} \nabla g_j(w_{j,\tau}^t), w_t - w^* \right\rangle \le 4\eta EGD\sqrt{\frac{2T}{m}\log(\frac{1}{\delta})}.$$

*Proof.* Define $\nabla f_t := \frac{1}{n} \sum_{j=1}^{n} \sum_{\tau=0}^{E-1} \nabla f_j(w_{j,\tau}^t)$ and $\tilde{\nabla} f_t := \frac{1}{m} \sum_{j \in \mathcal{S}_t} \sum_{\tau=0}^{E-1} \nabla f_j(w_{j,\tau}^t)$. Also, define $\Delta_t := \nabla f_t - \tilde{\nabla} f_t$. Now, we define a filtration $\mathcal{F}_t := \sigma(w_0, \mathcal{S}_0, \mathcal{S}_1, \ldots, \mathcal{S}_t)$, and therefore $\mathbb{E}[\delta_t \mid \mathcal{F}_{t-1}] = 0$.

Our goal is to bound the following term in high probability:

$$\sum_{t=0}^{T-1} 2\eta \langle \Delta_t, w_t - w^* \rangle \mathbb{1}_{t \in \mathcal{A}}.$$

Let's start by observing that $\langle \Delta_t, w_t - w^* \rangle \mathbb{1}_{t \in \mathcal{A}}$ is a martingale difference sequence. We can check this by

$$\begin{aligned}
\mathbb{E}[2\eta \langle \Delta_t, w_t - w^* \rangle \mathbb{1}_{t \in \mathcal{A}} \mid \mathcal{F}_{t-1}] &= \mathbb{E}[2\eta \langle \Delta_t, w_t - w^* \rangle \mathbb{1}_{t \in \mathcal{A}} \mid \mathcal{F}_{t-1}, \mathbb{1}_{t \in \mathcal{A}} = 1] \mathrm{Prob}\{\mathbb{1}_{t \in \mathcal{A}} = 1\} \\
&\quad + \mathbb{E}[2\eta \langle \Delta_t, w_t - w^* \rangle \mathbb{1}_{t \in \mathcal{A}} \mid \mathcal{F}_{t-1}, \mathbb{1}_{t \in \mathcal{A}} = 0] \mathrm{Prob}\{\mathbb{1}_{t \in \mathcal{A}} = 0\} \\
&= 0.
\end{aligned}$$

We can observe that $2\eta \langle \Delta_t, w_t - w^* \rangle$ for all $t \in [T]$ is $16\eta^2 E^2 G^2 D^2 / m-$sub-Gaussian. So, finally, by using Azuma-Hoeffding for sub-Gaussian random variables, we have

$$\mathrm{Prob} \left\{ \sum_{t=0}^{T-1} 2\eta \langle \Delta_t, w_t - w^* \rangle \mathbb{1}_{t \in \mathcal{A}} > \lambda \right\} \leq \exp\left( \frac{-\lambda^2 m}{32\eta^2 E^2 G^2 D^2 T} \right).$$

Set $\lambda = 4\eta EGD \sqrt{\frac{2T}{m} \log(\frac{1}{\delta})}$, we get for $\delta \in (0,1)$ with probability at least $1 - \delta$,

$$\sum_{t \in \mathcal{A}} 2\eta \left\langle \frac{1}{n} \sum_{j=1}^{n} \sum_{\tau=0}^{E-1} \nabla f_j(w_{j,\tau}^t) - \frac{1}{m} \sum_{j \in \mathcal{S}_t} \sum_{\tau=0}^{E-1} \nabla f_j(w_{j,\tau}^t), w_t - w^* \right\rangle \leq 4\eta EGD \sqrt{\frac{2T}{m} \log(\frac{1}{\delta})}. \tag{5}$$

Similarly, following this, we can bound

$$\sum_{t \in \mathcal{B}} 2\eta \left\langle \frac{1}{n} \sum_{j=1}^{n} \sum_{\tau=0}^{E-1} \nabla g_j(w_{j,\tau}^t) - \frac{1}{m} \sum_{j \in \mathcal{S}_t} \sum_{\tau=0}^{E-1} \nabla g_j(w_{j,\tau}^t), w_t - w^* \right\rangle \leq 4\eta EGD \sqrt{\frac{2T}{m} \log(\frac{1}{\delta})}. \tag{6}$$

$\square$

**Lemma 5** (Bounding Inner Product-Data Sampling). *Under Assumptions 2, 5, and independence of gradient and constraint evaluation, the following holds for $\delta \in (0,1)$ with probability at least $1 - \delta$*

$$\sum_{t \in \mathcal{A}} 2\eta \left\langle \frac{1}{m} \sum_{j \in \mathcal{S}_t} \sum_{\tau=0}^{E-1} \nabla f_j(w_{j,\tau}^t) - \frac{1}{m} \sum_{j \in \mathcal{S}_t} \sum_{\tau=0}^{E-1} \nabla f_j(w_{j,\tau}^t, \xi_{j,\tau}^t), w_t - w^* \right\rangle \leq 2\eta D\sigma_\xi \sqrt{\frac{2ET}{m} \log(\frac{1}{\delta})}.$$

$$\sum_{t \in \mathcal{B}} 2\eta \left\langle \frac{1}{m} \sum_{j \in \mathcal{S}_t} \sum_{\tau=0}^{E-1} \nabla g_j(w_{j,\tau}^t) - \frac{1}{m} \sum_{j \in \mathcal{S}_t} \sum_{\tau=0}^{E-1} \nabla g_j(w_{j,\tau}^t, \xi_{j,\tau}^t), w_t - w^* \right\rangle \leq 2\eta D\sigma_\xi \sqrt{\frac{2ET}{m} \log(\frac{1}{\delta})}.$$

*Proof.* Define $\hat{\nabla} f_t := \frac{1}{m} \sum_{j \in \mathcal{S}_t} \sum_{\tau=0}^{E-1} \nabla f_j(w_{j,\tau}^t, \xi_{j,\tau}^t)$ and $\tilde{\nabla} f_t := \frac{1}{n} \sum_{j \in \mathcal{S}_t} \sum_{\tau=0}^{E-1} \nabla f_j(w_{j,\tau}^t)$. Also, define $\tilde{\Delta}_t := \tilde{\nabla} f_t - \hat{\nabla} f_t$. Now, we define a filtration $\mathcal{F}_t := \sigma(w_0, \mathcal{S}_0, \xi_0, \ldots, \mathcal{S}_t, \xi_t)$ where $\xi_t := \{\xi_{j,\tau}^t\}_{j \in \mathcal{S}_t, \tau \in [E]}$, and therefore $\mathbb{E}[\tilde{\delta}_t \mid \mathcal{F}_{t-1}] = 0$.

Our goal is to bound the following term in high probability:

$$\sum_{t=0}^{T-1} 2\eta \left\langle \tilde{\Delta}_t, w_t - w^* \right\rangle \mathbb{1}_{t \in \mathcal{A}}.$$

Let's start by observing that $\left\langle \tilde{\Delta}_t, w_t - w^* \right\rangle \mathbb{1}_{t \in \mathcal{A}}$ is a martingale difference sequence. We can check this by

$$\mathbb{E}[2\eta \left\langle \tilde{\Delta}_t, w_t - w^* \right\rangle \mathbb{1}_{t \in \mathcal{A}} \mid \mathcal{F}_{t-1}] = \mathbb{E}[2\eta \left\langle \tilde{\Delta}_t, w_t - w^* \right\rangle \mathbb{1}_{t \in \mathcal{A}} \mid \mathcal{F}_{t-1}, \mathbb{1}_{t \in \mathcal{A}} = 1] \mathrm{Prob}\{\mathbb{1}_{t \in \mathcal{A}} = 1\}$$

$$+ \mathbb{E}[2\eta \left\langle \tilde{\Delta}_t, w_t - w^* \right\rangle \mathbb{1}_{t \in \mathcal{A}} \mid \mathcal{F}_{t-1}, \mathbb{1}_{t \in \mathcal{A}} = 0] \text{Prob}\{\mathbb{1}_{t \in \mathcal{A}} = 0\}$$
$$= 0.$$

Using Assumption 5, we can observe that $\tilde{\Delta}_t$ is $E\sigma_\xi^2/m-$sub-Gaussian. Thus for all $t \in [T]$, we have that the term $2\eta \left\langle \tilde{\Delta}_t, w_t - w^* \right\rangle$ is $4\eta^2 D^2 E\sigma_\xi^2/m-$sub-Gaussian. So, finally, by using Azuma-Hoeffding for sub-Gaussian random variables, we have

$$\text{Prob}\left\{ \sum_{t=0}^{T-1} 2\eta \left\langle \tilde{\Delta}_t, w_t - w^* \right\rangle \mathbb{1}_{t \in \mathcal{A}} > \lambda \right\} \leq \exp\left( \frac{-\lambda^2 m}{8\eta^2 D^2 \sigma_\xi^2 ET} \right).$$

Set $\lambda = 2\eta D\sigma_\xi \sqrt{2ET \frac{1}{m} \log(\frac{1}{\delta})}$, we get for $\delta \in (0,1)$ with probability at least $1 - \delta$,

$$\sum_{t \in \mathcal{A}} 2\eta \left\langle \frac{1}{m} \sum_{j \in \mathcal{S}_t} \sum_{\tau=0}^{E-1} \nabla f_j(w_{j,\tau}^t) - \frac{1}{m} \sum_{j \in \mathcal{S}_t} \sum_{\tau=0}^{E-1} \nabla f_j(w_{j,\tau}^t, \xi_{j,\tau}^t), w_t - w^* \right\rangle \leq 2\eta D\sigma_\xi \sqrt{\frac{2ET}{m} \log(\frac{1}{\delta})}.$$
$$(7)$$

Similarly, following this, we can bound

$$\sum_{t \in \mathcal{A}} 2\eta \left\langle \frac{1}{m} \sum_{j \in \mathcal{S}_t} \sum_{\tau=0}^{E-1} \nabla g_j(w_{j,\tau}^t) - \frac{1}{m} \sum_{j \in \mathcal{S}_t} \sum_{\tau=0}^{E-1} \nabla g_j(w_{j,\tau}^t, \xi_{j,\tau}^t), w_t - w^* \right\rangle \leq 2\eta D\sigma_\xi \sqrt{\frac{2ET}{m} \log(\frac{1}{\delta})}.$$
$$(8)$$
$$\square$$

---

**Lemma 6** (Client-sampling + SGD noise). *Fix integers $T \geq 1$, $E \geq 1$, number of clients $n \geq 1$, and per-round sample size $m \in [n]$. Define, for each round $t$,*

$$\Delta_t := \frac{1}{n} \sum_{j=1}^{n} \sum_{\tau=0}^{E-1} \bar{\nu}_{j,\tau}^t - \frac{1}{m} \sum_{j \in \mathcal{S}_t} \sum_{\tau=0}^{E-1} \nu_{j,\tau}^t.$$

*Then for any $\delta \in (0,1)$ and any stepsize $\eta > 0$, with probability at least $1 - \delta$,*

$$2\eta^2 \|\Delta_t\|^2 \leq \frac{8\eta^2 E}{m} \left( EG^2 + \sigma_\xi^2 \right) \log\left(\frac{4T}{\delta}\right) \quad \text{for all } t \in [T]. \quad (9)$$

*Proof.* Fix a round $t$. Decompose $\Delta_t$ into a *client-sampling* part and a *noise* part:

$$\Delta_t = \underbrace{\frac{1}{n} \sum_{j=1}^{n} \sum_{\tau=0}^{E-1} \bar{\nu}_{j,\tau}^t - \frac{1}{m} \sum_{j \in \mathcal{S}_t} \sum_{\tau=0}^{E-1} \bar{\nu}_{j,\tau}^t}_{=:A_t} + \underbrace{\frac{1}{m} \sum_{j \in \mathcal{S}_t} \sum_{\tau=0}^{E-1} \delta_{j,\tau}^t}_{=:B_t}.$$

Thus $\|\Delta_t\|^2 \leq 2\|A_t\|^2 + 2\|B_t\|^2$. We have $v_j^t := \sum_{\tau=0}^{E-1} \bar{\nu}_{j,\tau}^t$, $\|v_j^t\| \leq EG$. Then

$$A_t = \frac{1}{n} \sum_{j=1}^{n} v_j^t - \frac{1}{m} \sum_{j \in \mathcal{S}_t} v_j^t.$$

By Hoeffding's inequality, we have,

$$\mathbb{P}\Big( \|A_t\| \geq \lambda \mid \mathcal{F}_{t-1} \Big) \leq 2\exp\left( -\frac{m\lambda^2}{2E^2 G^2} \right).$$

Set $\lambda_1 = GE\sqrt{\frac{2}{m} \log\left(\frac{4T}{\delta}\right)}$. Then

$$\mathbb{P}\Big( \|A_t\| \geq \lambda_1 \Big) \leq 2\exp\left( -\log \frac{4T}{\delta} \right) = \frac{\delta}{2T}.$$

Now, using Assumption 5 we have $\mathbb{E}\left[\exp\left(\frac{\|\delta_{j,\tau}^t\|^2}{\sigma_\xi^2}\right) \,\Big|\, \mathcal{F}_{t-1}\right] \le \exp(1)$, and the family $\{\delta_{j,\tau}^t\}$ is conditionally independent given $\mathcal{F}_{t-1}$. By standard sub-Gaussian vector concentration for averages of independent sub-Gaussian vectors, we get

$$\mathbb{P}\Big(\|B_t\| \ge \lambda \,\Big|\, \mathcal{F}_{t-1}\Big) \le 2\exp\left(-\frac{m\,\lambda^2}{2E\sigma_\xi^2}\right).$$

Set

$$\lambda_2 = \sigma_\xi \sqrt{\frac{2E}{m}\log\Big(\frac{4T}{\delta}\Big)}.$$

Then

$$\mathbb{P}\Big(\|B_t\| \ge \lambda_2\Big) \le 2\exp\left(-\log\frac{4T}{\delta}\right) = \frac{\delta}{2T}.$$

Finally, by a union bound over $t = 0, \ldots, T-1$, with probability at least $1 - \delta$, we have for all $t \in [T]$

$$\|\Delta_t\|^2 \le 2\lambda_1^2 + 2\lambda_2^2 = 2\left(G^2\frac{2E^2}{m}\log\frac{4T}{\delta} + \sigma_\xi^2\frac{2E}{m}\log\frac{4T}{\delta}\right) = \frac{4E}{m}(EG^2 + \sigma_\xi^2)\log\frac{4T}{\delta}.$$

Multiplying by $2\eta^2$ gives, for all $t$,

$$2\eta^2\|\Delta_t\|^2 \le \frac{8\eta^2 E}{m}(EG^2 + \sigma_\xi^2)\log\frac{4T}{\delta},$$

which is exactly equation 9. $\qquad\square$

---

**Lemma 7** (Constraint Gap-Client Sampling). *Under Assumption 4, the following holds for $\delta \in (0,1)$ with probability at least $1 - \delta$*

- 
$$|\hat{G}(w_t) - g(w_t)| \le \sigma\sqrt{\frac{2}{m}\log\left(\frac{2T}{\delta}\right)}, \qquad \forall\, t \in [T].$$

- 
$$\sum_{t\in\mathcal{B}} \hat{G}(w_t) - g(w_t) \le \sigma T\sqrt{\frac{2}{m}\log\left(\frac{2T}{\delta}\right)}.$$

*Proof.* We need to bound the term $\sum_{t=0}^{T-1}(\hat{G}(w_t) - g(w_t))\mathbb{1}_{t\in\mathcal{B}}$ in high probability. We know from Assumption 4 that $\hat{G}(w_t) - g(w_t)$ is $\sigma^2/m-$sub-Gaussian. Then, by utilizing Chernoff bounds for sub-Gaussian random variables, we have

$$\text{Prob}\left\{|\hat{G}(w_t) - g(w_t)| \ge \lambda_t\right\} \le 2\exp\left(-\frac{\lambda_t^2 m}{2\sigma^2}\right).$$

Set, $\lambda_t := \sigma\sqrt{\frac{2}{m}\log\left(\frac{2}{\delta^t}\right)}$. Then we have

$$\text{Prob}\left\{|\hat{G}(w_t) - g(w_t)| \ge \sigma\sqrt{\frac{2}{m}\log\left(\frac{2}{\delta^t}\right)}\right\} \le \delta^t.$$

Applying the union bound over all $t \in [T]$, we get for $\delta^t = \frac{\delta}{T}$

$$\text{Prob}\left\{|\hat{G}(w_t) - g(w_t)| \le \sigma\sqrt{\frac{2}{m}\log\left(\frac{2T}{\delta}\right)}\right\} \ge 1 - \delta, \qquad \forall\, t \in [T].$$

So, finally we have for $\delta \in (0,1)$ with probability at least $1 - \delta$

$$\sum_{t\in\mathcal{B}} \hat{G}(w_t) - g(w_t) \le \sigma T\sqrt{\frac{2}{m}\log\left(\frac{2T}{\delta}\right)}.$$

$\qquad\square$

**Lemma 8** (Constraint Gap-Client + Data Sampling). *Under Proposition 1, the following holds for* $\delta \in (0,1)$ *with probability at least* $1 - \delta$

- 
$$|\tilde{G}(w_t) - g(w_t)| \leq \sqrt{\frac{2}{m}(\sigma^2 + \frac{\sigma_b^2}{N_b})\log\left(\frac{2T}{\delta}\right)}, \qquad \forall\, t \in [T].$$

- 
$$\sum_{t \in \mathcal{B}} \tilde{G}(w_t) - g(w_t) \leq T\sqrt{\frac{2}{m}(\sigma^2 + \frac{\sigma_b^2}{N_b})\log\left(\frac{2T}{\delta}\right)}.$$

*Proof.* We need to bound the term $\sum_{t=0}^{T-1}(\tilde{G}(w_t) - g(w_t))\mathbb{1}_{t \in \mathcal{B}}$ in high probability. We know from Proposition 1 that $\tilde{G}(w_t) - g(w_t)$ is $\left(\frac{\sigma^2}{m} + \frac{\sigma_b^2}{mN_b}\right)$−sub-Gaussian. Then, by utilizing Chernoff bounds for sub-Gaussian random variables, we have

$$\text{Prob}\left\{|\tilde{G}(w_t) - g(w_t)| \geq \lambda_t\right\} \leq 2\exp\left(-\frac{\lambda_t^2 m}{2(\sigma^2 + \sigma_b^2/N_b)}\right).$$

Set, $\lambda_t := \sqrt{\frac{2}{m}(\sigma^2 + \frac{\sigma_b^2}{N_b})\log\left(\frac{2}{\delta^t}\right)}$. Then we have

$$\text{Prob}\left\{|\tilde{G}(w_t) - g(w_t)| \geq \sqrt{\frac{2}{m}(\sigma^2 + \frac{\sigma_b^2}{N_b})\log\left(\frac{2}{\delta^t}\right)}\right\} \leq \delta^t.$$

Applying the union bound over all $t \in [T]$, we get for $\delta^t = \frac{\delta}{T}$

$$\text{Prob}\left\{|\tilde{G}(w_t) - g(w_t)| \leq \sqrt{\frac{2}{m}(\sigma^2 + \frac{\sigma_b^2}{N_b})\log\left(\frac{2T}{\delta}\right)}\right\} \geq 1 - \delta, \qquad \forall\, t \in [T].$$

So, finally we have for $\delta \in (0,1)$ with probability at least $1 - \delta$

$$\sum_{t \in \mathcal{B}} \tilde{G}(w_t) - g(w_t) \leq T\sqrt{\frac{2}{m}(\sigma^2 + \frac{\sigma_b^2}{N_b})\log\left(\frac{2T}{\delta}\right)}.$$

$\square$

## C.2 Proofs of Theorems

The proof presented in the main paper is provided in a compressed form and combined into two theorems for clarity. In this appendix, we expand upon each part in detail. We begin with the analysis of the FEDSGM framework, followed by the bi-directional Compression component. For both of these, we provide complete proofs under hard switching for both full and partial participation settings. We then turn to the soft switching case, where we include the proof for the full participation setting only.

## C.3 Main Theorems FEDSGM

### C.3.1 Hard Switching — Full Participation

**Theorem 3.** *Consider the problem in eq. (1), where* $\mathcal{X} = \mathbb{R}^d$ *and Algorithm 1, under Assumption 1. Define* $D := \|w_0 - w^*\|$ *and*

$$\mathcal{A} = \{t \in [T] \mid g(w_t) \leq \epsilon\}, \qquad \bar{w} = \frac{1}{|\mathcal{A}|}\sum_{t \in \mathcal{A}} w_t.$$

*Then, if*

$$\epsilon = \sqrt{\frac{4D^2G^2E}{T}}, \text{ and } \eta = \sqrt{\frac{D^2}{4G^2E^3T}}$$

*it holds that* $\mathcal{A}$ *is nonempty,* $\bar{w}$ *is well-defined, and* $\bar{w}$ *is an* $\epsilon$*-solution for P.*

*Proof.* Using Algorithm 1, the update rule for the global model is

$$w_{t+1} = w_t - \eta \cdot \frac{1}{n} \sum_{j=1}^{n} \sum_{\tau=0}^{E-1} \nu_{j,\tau}^t, \tag{10}$$

We analyze the squared distance to the optimal point $w^*$ as follows,

$$\|w_{t+1} - w^*\|^2 = \left\| w_t - \eta \cdot \frac{1}{n} \sum_{j=1}^{n} \sum_{\tau=0}^{E-1} \nu_{j,\tau}^t - w^* \right\|^2$$

$$= \|w_t - w^*\|^2 + \underbrace{\eta^2 \left\| \frac{1}{n} \sum_{j=1}^{n} \sum_{\tau=0}^{E-1} \nu_{j,\tau}^t \right\|^2}_{Term-A} \underbrace{-2\eta \left\langle w_t - w^*, \frac{1}{n} \sum_{j=1}^{n} \sum_{\tau=0}^{E-1} \nu_{j,\tau}^t \right\rangle}_{Term-B}$$

Firstly, we start with upper-bounding $Term - A$.

$$Term - A = \eta^2 \left\| \frac{1}{n} \sum_{j=1}^{n} \sum_{\tau=0}^{E-1} \nu_{j,\tau}^t \right\|^2 \overset{Jensen's}{\leq} \eta^2 \frac{1}{n} \sum_{j=1}^{n} E \sum_{\tau=0}^{E-1} \|\nu_{j,\tau}^t\|^2$$

$$\overset{G-Lip}{\leq} \eta^2 E^2 G^2$$

Now we can use Lemma 1 to bound $Term - B$. So, putting $Term - A$ and $Term - B$ back into the expression we get

$$\|w_{t+1} - w^*\|^2 \leq \|w_t - w^*\|^2 + \eta^2 E^2 G^2 + \eta \alpha E G^2$$

$$+ \frac{\eta}{\alpha n} \sum_{j=1}^{n} \sum_{\tau=0}^{E-1} \|w_t - w_{j,\tau}^t\|^2$$

$$+ \frac{2\eta}{n} \sum_{j=1}^{n} \sum_{\tau=0}^{E-1} (f_j(w^*) - f_j(w_{j,\tau}^t)) \mathbb{1}\{t \in \mathcal{A}\}$$

$$+ \frac{2\eta}{n} \sum_{j=1}^{n} \sum_{\tau=0}^{E-1} (g_j(w^*) - g_j(w_{j,\tau}^t)) \mathbb{1}\{t \in \mathcal{B}\}. \tag{11}$$

Now our goal is to handle the term $f_j(w^*) - f_j(w_{j,\tau}^t)$, so we first rewrite,

$$f_j(w_{j,\tau}^t) \geq f_j(w_t) + \langle \nabla f_j(w_t), w_{j,\tau}^t - w_t \rangle \quad \text{(by convexity)}$$

$$\Rightarrow f_j(w^*) - f_j(w_{j,\tau}^t) \leq f_j(w^*) - f_j(w_t) - \langle \nabla f_j(w_t), w_{j,\tau}^t - w_t \rangle.$$

Using Young's inequality with parameter $\alpha > 0$, we get

$$-\langle \nabla f_j(w_t), w_{j,\tau}^t - w_t \rangle \leq \frac{1}{2\alpha} \|w_{j,\tau}^t - w_t\|^2 + \frac{\alpha}{2} \|\nabla f_j(w_t^t)\|^2.$$

Thus, we get,

$$f_j(w^*) - f_j(w_{j,\tau}^t) \leq f_j(w^*) - f_j(w_t) + \frac{1}{2\alpha} \|w_t - w_{j,\tau}^t\|^2 + \frac{\alpha}{2} \|\nabla f_j(w_t)\|^2$$

$$\overset{G-Lip}{\leq} f_j(w^*) - f_j(w_t) + \frac{1}{2\alpha} \|w_t - w_{j,\tau}^t\|^2 + \frac{\alpha}{2} G^2.$$

Similarly, we can handle the other term with $g$ and get,

$$g_j(w^*) - g_j(w_{j,\tau}^t) \leq g_j(w^*) - g_j(w_t) + \frac{1}{2\alpha} \|w_t - w_{j,\tau}^t\|^2 + \frac{\alpha}{2} G^2.$$

So, putting these inequalities back in equation 11 along with the use of Lemma 2 and eq. (1), we get

$$\|w_{t+1} - w^*\|^2 \leq \|w_t - w^*\|^2 + \eta^2 E^2 G^2 + 2\eta\alpha E G^2 + \frac{2}{3\alpha}\eta^3 E^3 G^2$$
$$+ 2\eta E(f(w^*) - f(w_t))\mathbb{1}\{t \in \mathcal{A}\} + 2\eta E(g(w^*) - g(w_t))\mathbb{1}\{t \in \mathcal{B}\}.$$

Now, for $\alpha = \eta$, and rearranging the terms we get

$$(f(w_t) - f(w^*))\mathbb{1}\{t \in \mathcal{A}\} + (g(w_t) - g(w^*))\mathbb{1}\{t \in \mathcal{B}\} \leq \frac{\|w_t - w^*\|^2 - \|w_{t+1} - w^*\|^2}{2\eta E}$$
$$+ \frac{\eta}{2}EG^2 + \eta G^2 + \frac{1}{3}\eta E^2 G^2.$$

Defining $D := \|w_0 - w^*\|$ and summing the expression above for $t = 0, 1, \cdots, T - 1$ and then dividing by $T$, we get

$$\frac{1}{T}\sum_{t \in \mathcal{A}}(f(w_t) - f(w^*)) + \frac{1}{T}\sum_{t \in \mathcal{B}}(g(w_t) - g(w^*)) \leq \frac{D^2}{2\eta ET} + \frac{\eta}{2}EG^2 + \eta G^2 + \frac{1}{3}\eta E^2 G^2.$$

Now choosing $\eta = \sqrt{\frac{D^2}{2G^2 ET\Gamma}}$, where $\Gamma = \frac{1}{2}E + 1 + \frac{1}{3}E^2$, we get

$$\frac{1}{T}\sum_{t \in \mathcal{A}}(f(w_t) - f(w^*)) + \frac{1}{T}\sum_{t \in \mathcal{B}}(g(w_t) - g(w^*)) \leq \sqrt{\frac{2D^2 G^2\Gamma}{ET}}.$$

Note that when $\epsilon$ is sufficiently large, $\mathcal{A}$ is nonempty. Assuming an empty $\mathcal{A}$, we can find the largest "bad" $\epsilon$:

$$\epsilon_{bad} < \frac{1}{T}\sum_{t \in \mathcal{B}}g(w_t) - g(w^*) \leq \sqrt{\frac{2D^2 G^2\Gamma}{ET}}$$

Thus, let us set $\epsilon = (N + 1)\sqrt{\frac{2D^2 G^2\Gamma}{ET}}$ for some $N \geq 0$. With this choice, $\mathcal{A}$ is guaranteed to be nonempty.

Now, we consider two cases. Either $\sum_{t \in \mathcal{A}} f(w_t) - f(w^*) \leq 0$ which implies by the convexity of $f$ and $g$ for $\bar{w} = \frac{1}{|\mathcal{A}|}\sum_{t \in \mathcal{A}} w_t$ we have

$$f(\bar{w}) - f(w^*) \leq 0 < \epsilon, \qquad g(\bar{w}) \leq \epsilon. \tag{12}$$

Otherwise, if $\sum_{t \in \mathcal{A}} f(w_t) - f(w^*) > 0$, then

$$\sqrt{\frac{2D^2 G^2\Gamma}{ET}} \geq \frac{1}{T}\sum_{t \in \mathcal{A}} f(w_t) - f(w^*) + \frac{1}{T}\sum_{t \in \mathcal{B}} g(w_t) - g(w^*)$$
$$> \frac{1}{T}\sum_{t \in \mathcal{A}} f(w_t) - f(w^*) + \frac{1}{T}\sum_{t \in \mathcal{B}}\epsilon$$
$$= \frac{|\mathcal{A}|}{T}\frac{1}{|\mathcal{A}|}\sum_{t \in \mathcal{A}} f(w_t) - f(w^*) + (1 - \frac{|\mathcal{A}|}{T})\epsilon \tag{13}$$
$$\geq \frac{|\mathcal{A}|}{T}\left(f(\bar{w}) - f(w^*)\right) + (1 - \frac{|\mathcal{A}|}{T})\epsilon.$$

By rearranging

$$\frac{|\mathcal{A}|}{T}\left(f(\bar{w}) - f(w^*) - \epsilon\right) < \sqrt{\frac{2D^2 G^2\Gamma}{ET}} - \epsilon \leq -N\sqrt{\frac{2D^2 G^2\Gamma}{ET}}, \tag{14}$$

Implying $f(\bar{w}) - f(w^*) < \epsilon$ and further by convexity of $g$ for $\bar{w} = \frac{1}{|\mathcal{A}|}\sum_{t \in \mathcal{A}} w_t$, we also have $g(\bar{w}) \leq \epsilon$. □

### C.3.2 Hard Switching — Partial Participation

**Theorem 4.** *Consider the optimization problem in equation 1 and Algorithm 1 under Assumptions 1, 2, and 4. Additionally, assume that at each global round, $m$ distinct clients are sampled uniformly at random. Define*

$$\mathcal{A} := \left\{ t \in [T] \,\middle|\, \hat{G}(w_t) \leq \epsilon \right\}, \qquad \bar{w} := \frac{1}{|\mathcal{A}|} \sum_{t \in \mathcal{A}} w_t.$$

*Suppose the step size $\eta$ and constraint threshold $\epsilon$ are set as*

$$\eta = \sqrt{\frac{D^2}{4G^2 E^3 T}} \qquad and \; \epsilon = \sqrt{\frac{4D^2 G^2 E}{T}} + \frac{4DG}{\sqrt{mT}} \sqrt{2 \log\left(\frac{3}{\delta}\right)} + 2\sigma \sqrt{\frac{2}{m} \log\left(\frac{6T}{\delta}\right)},$$

*where $\delta \in (0,1)$ is a confidence parameter. Then, with probability at least $1 - \delta$, the following hold:*

1. *The set $\mathcal{A}$ is non-empty, so $\bar{w}$ is well-defined.*

2. *The average iterate $\bar{w}$ satisfies*

$$f(\bar{w}) - f(w^*) \leq \epsilon, \qquad g(\bar{w}) \leq \epsilon.$$

*Proof.* Using Algorithm 1, the update rule for the global model under partial participation is

$$w_{t+1} = \Pi_{\mathcal{X}} \left( w_t - \eta \cdot \frac{1}{m} \sum_{j \in \mathcal{S}_t} \sum_{\tau=0}^{E-1} \nu_{j,\tau}^t \right). \tag{15}$$

We analyze the squared distance to the optimal point $w^*$ as follows,

$$\|w_{t+1} - w^*\|^2 \overset{1-Lip \; \Pi_{\mathcal{X}}(\cdot)}{\leq} \left\| w_t - \eta \cdot \frac{1}{m} \sum_{j \in \mathcal{S}_t} \sum_{\tau=0}^{E-1} \nu_{j,\tau}^t - w^* \right\|^2$$

$$= \|w_t - w^*\|^2 + \eta^2 \left\| \frac{1}{m} \sum_{j \in \mathcal{S}_t} \sum_{\tau=0}^{E-1} \nu_{j,\tau}^t \right\|^2 - 2\eta \left\langle w_t - w^*, \frac{1}{m} \sum_{j \in \mathcal{S}_t} \sum_{\tau=0}^{E-1} \nu_{j,\tau}^t \right\rangle$$

$$\leq \|w_t - w^*\|^2 + \eta^2 E^2 G^2 - 2\eta \left\langle w_t - w^*, \frac{1}{m} \sum_{j \in \mathcal{S}_t} \sum_{\tau=0}^{E-1} \nu_{j,\tau}^t \right\rangle$$

$$= \|w_t - w^*\|^2 + \eta^2 E^2 G^2 + 2\eta \left\langle w_t - w^*, \frac{1}{n} \sum_{j=1}^{n} \sum_{\tau=0}^{E-1} \nu_{j,\tau}^t - \frac{1}{m} \sum_{j \in \mathcal{S}_t} \sum_{\tau=0}^{E-1} \nu_{j,\tau}^t \right\rangle$$

$$- 2\eta \left\langle w_t - w^*, \frac{1}{n} \sum_{j=1}^{n} \sum_{\tau=0}^{E-1} \nu_{j,\tau}^t \right\rangle.$$

Now we can use Lemma 1 and Lemma 2 to bound LHS like we have done in previous proofs, we get

$$2\eta E(f(w_t) - f(w^*)) \mathbb{1}_{t \in \mathcal{A}} + 2\eta E(g(w_t) - g(w^*)) \mathbb{1}_{t \in \mathcal{B}} \leq \|w_t - w^*\|^2$$

$$- \|w_{t+1} - w^*\|^2 + \eta^2 E^2 G^2 + 2\eta^2 E G^2 + \frac{2}{3} \eta^2 E^3 G^2$$

$$+ 2\eta \left\langle \frac{1}{n} \sum_{j=1}^{n} \sum_{\tau=0}^{E-1} \nabla f_j(w_{j,\tau}^t) - \frac{1}{m} \sum_{j \in \mathcal{S}_t} \sum_{\tau=0}^{E-1} \nabla f_j(w_{j,\tau}^t), w_t - w^* \right\rangle \mathbb{1}_{t \in \mathcal{A}}$$

$$+ 2\eta \left\langle \frac{1}{n} \sum_{j=1}^{n} \sum_{\tau=0}^{E-1} \nabla g_j(w_{j,\tau}^t) - \frac{1}{m} \sum_{j \in \mathcal{S}_t} \sum_{\tau=0}^{E-1} \nabla g_j(w_{j,\tau}^t), w_t - w^* \right\rangle \mathbb{1}_{t \in \mathcal{B}}.$$

Again add and subtract $\hat{G}(w_t)$ from the 2nd term on LHS and rearrange and we get,

$$2\eta E(f(w_t) - f(w^*))\mathbb{1}_{t\in\mathcal{A}} + 2\eta E(\hat{G}(w_t) - g(w^*))\mathbb{1}_{t\in\mathcal{B}} \leq \|w_t - w^*\|^2$$

$$- \|w_{t+1} - w^*\|^2 + \eta^2 E^2 G^2 + 2\eta^2 E G^2 + \frac{2}{3}\eta^2 E^3 G^2$$

$$+ 2\eta \left\langle \frac{1}{n}\sum_{j=1}^{n}\sum_{\tau=0}^{E-1}\nabla f_j(w_{j,\tau}^t) - \frac{1}{m}\sum_{j\in\mathcal{S}_t}\sum_{\tau=0}^{E-1}\nabla f_j(w_{j,\tau}^t), w_t - w^* \right\rangle \mathbb{1}_{t\in\mathcal{A}}$$

$$+ 2\eta \left\langle \frac{1}{n}\sum_{j=1}^{n}\sum_{\tau=0}^{E-1}\nabla g_j(w_{j,\tau}^t) - \frac{1}{m}\sum_{j\in\mathcal{S}_t}\sum_{\tau=0}^{E-1}\nabla g_j(w_{j,\tau}^t), w_t - w^* \right\rangle \mathbb{1}_{t\in\mathcal{B}}$$

$$+ 2\eta E(\hat{G}(w_t) - g(w_t))\mathbb{1}_{t\in\mathcal{B}}.$$

Now sum both sides over $t \in [T]$, we get

$$\sum_{t\in\mathcal{A}} 2\eta E(f(w_t) - f(w^*)) + \sum_{t\in\mathcal{B}} 2\eta E(\hat{G}(w_t) - g(w^*)) \leq D^2 + \eta^2 E^2 G^2 T + 2\eta^2 E G^2 T + \frac{2}{3}\eta^2 E^3 G^2 T$$

$$+ \sum_{t\in\mathcal{A}} 2\eta \left\langle \frac{1}{n}\sum_{j=1}^{n}\sum_{\tau=0}^{E-1}\nabla f_j(w_{j,\tau}^t) - \frac{1}{m}\sum_{j\in\mathcal{S}_t}\sum_{\tau=0}^{E-1}\nabla f_j(w_{j,\tau}^t), w_t - w^* \right\rangle$$

$$+ \sum_{t\in\mathcal{B}} 2\eta \left\langle \frac{1}{n}\sum_{j=1}^{n}\sum_{\tau=0}^{E-1}\nabla g_j(w_{j,\tau}^t) - \frac{1}{m}\sum_{j\in\mathcal{S}_t}\sum_{\tau=0}^{E-1}\nabla g_j(w_{j,\tau}^t), w_t - w^* \right\rangle$$

$$+ \sum_{t\in\mathcal{B}} 2\eta E(\hat{G}(w_t) - g(w_t)).$$

Now, by using Lemmas 4 and 7, we can bound the term above for $\delta \in (0,1)$ with probability at least $1 - \delta$

$$\sum_{t\in\mathcal{A}} 2\eta E(f(w_t) - f(w^*)) + \sum_{t\in\mathcal{B}} 2\eta E(\hat{G}(w_t) - g(w^*)) \leq D^2 + \eta^2 E^2 G^2 T + 2\eta^2 E G^2 T + \frac{2}{3}\eta^2 E^3 G^2 T$$

$$+ 8\eta E G D\sqrt{\frac{2T}{m}\log\left(\frac{3}{\delta}\right)} + 2\eta E\sigma T\sqrt{\frac{2}{m}\log\left(\frac{6T}{\delta}\right)}.$$

Dividing both sides by $2\eta ET$ we get

$$\frac{1}{T}\sum_{t\in\mathcal{A}}(f(w_t) - f(w^*)) + \frac{1}{T}\sum_{t\in\mathcal{B}}(\hat{G}(w_t) - g(w^*)) \leq \frac{D^2}{2\eta ET} + \frac{\eta}{2}EG^2 + \eta G^2 + \frac{1}{3}\eta E^2 G^2$$

$$+ \frac{4GD}{\sqrt{mT}}\sqrt{2\log\left(\frac{3}{\delta}\right)} + \sigma\sqrt{\frac{2}{m}\log\left(\frac{6T}{\delta}\right)}.$$

Now choosing $\eta = \sqrt{\frac{D^2}{2G^2 ET\Gamma}}$, where $\Gamma = \frac{1}{2}E + 1 + \frac{1}{3}E^2$, we get

$$\frac{1}{T}\sum_{t\in\mathcal{A}}(f(w_t) - f(w^*)) + \frac{1}{T}\sum_{t\in\mathcal{B}}(\hat{G}(w_t) - g(w^*)) \leq \sqrt{\frac{2D^2 G^2 \Gamma}{ET}} + \frac{4GD}{\sqrt{mT}}\sqrt{2\log\left(\frac{3}{\delta}\right)}$$

$$+ \sigma\sqrt{\frac{2}{m}\log\left(\frac{6T}{\delta}\right)}.$$

Again, using a similar style of analysis as in previous proofs, if $\mathcal{A} = \phi$, then

$$\epsilon_{bad} < \sqrt{\frac{2D^2 G^2 \Gamma}{ET}} + \frac{4GD}{\sqrt{mT}}\sqrt{2\log\left(\frac{3}{\delta}\right)} + \sigma\sqrt{\frac{2}{m}\log\left(\frac{6T}{\delta}\right)}.$$

Thus, if we set $\epsilon = \sqrt{\frac{2D^2G^2\Gamma}{ET}} + \frac{4GD}{\sqrt{mT}}\sqrt{2\log\left(\frac{3}{\delta}\right)} + \sigma\sqrt{\frac{2}{m}\log\left(\frac{6T}{\delta}\right)}$, $\mathcal{A} \neq \phi$. Now, let's consider two different cases here as well. Either $\sum_{t\in\mathcal{A}} f(w_t) - f(w^*) \leq 0$ which implies by the convexity of $f$ for $\bar{w} = \frac{1}{|\mathcal{A}|}\sum_{t\in\mathcal{A}} w_t$, we have

$$f(\bar{w}) - f(w^*) \leq 0 < \epsilon. \tag{16}$$

Additionally, we know via convexity of g that

$$g(\bar{w}) \leq \frac{1}{|\mathcal{A}|}\sum_{t\in\mathcal{A}} g(w_t)$$

$$\leq \frac{1}{|\mathcal{A}|}\sum_{t\in\mathcal{A}}(g(w_t) - \hat{G}(w_t)) + \frac{1}{|\mathcal{A}|}\sum_{t\in\mathcal{A}}\hat{G}(w_t)$$

$$\overset{Lem.\ 7}{\leq} \epsilon + \sigma\sqrt{\frac{2}{m}\log\left(\frac{6T}{\delta}\right)}, \text{ with probability at least } 1 - \delta.$$

Otherwise, if $\sum_{t\in\mathcal{A}} f(w_t) - f(w^*) > 0$, then

$$\sqrt{\frac{2D^2G^2\Gamma}{ET}} + \frac{4GD}{\sqrt{mT}}\sqrt{2\log\left(\frac{3}{\delta}\right)} + \sigma\sqrt{\frac{2}{m}\log\left(\frac{6T}{\delta}\right)} \geq \frac{1}{T}\sum_{t\in\mathcal{A}} f(w_t) - f(w^*) + \frac{1}{T}\sum_{t\in\mathcal{B}}\hat{G}(w_t) - g(w^*)$$

$$> \frac{1}{T}\sum_{t\in\mathcal{A}} f(w_t) - f(w^*) + \frac{1}{T}\sum_{t\in\mathcal{B}}\epsilon$$

$$= \frac{|\mathcal{A}|}{T}\frac{1}{|\mathcal{A}|}\sum_{t\in\mathcal{A}} f(w_t) - f(w^*) + (1 - \frac{|\mathcal{A}|}{T})\epsilon$$

$$\geq \frac{|\mathcal{A}|}{T}\left(f(\bar{w}) - f(w^*)\right) + (1 - \frac{|\mathcal{A}|}{T})\epsilon. \tag{17}$$

This implies $f(\bar{w}) - f(w^*) < \epsilon$ and further using the same argument as above for $g$, we have for probability at least $1 - \delta$, $g(\bar{w}) \leq \epsilon + \sigma\sqrt{\frac{2}{m}\log\left(\frac{6T}{\delta}\right)}$. $\qquad\square$

### C.3.3 Soft Switching — Full participation

**Theorem 5.** *Consider the problem in equation 1, where $\mathcal{X} = \mathbb{R}^d$ and Algorithm 1, under Assumption 1. Define $D := \|w_0 - w^*\|$ and*

$$\mathcal{A} = \{t \in [T] \mid g(w_t) < \epsilon\}, \qquad \bar{w} = \sum_{t\in\mathcal{A}}\alpha_t w_t,$$

*where*

$$\alpha_t = \frac{1 - \sigma_\beta(g(w_t) - \epsilon)}{\sum_{t\in\mathcal{A}}[1 - \sigma_\beta(g(w_t) - \epsilon)]}. \tag{18}$$

*Then, if*

$$\epsilon = \sqrt{\frac{4D^2G^2E}{T}}, \qquad \eta = \sqrt{\frac{D^2}{4G^2E^3T}}, \quad and \quad \beta \geq \frac{2}{\epsilon}$$

*it holds that $\mathcal{A}$ is nonempty, $\bar{w}$ is well-defined, and $\bar{w}$ is an $\epsilon$-solution for $P$.*

*Proof.* Using Algorithm 1, the update rule for the global model is

$$w_{t+1} = w_t - \eta \cdot \frac{1}{n}\sum_{j=1}^{n}\sum_{\tau=0}^{E-1}\nu_{j,\tau}^t,$$

$$\text{where, } \nu_{j,\tau}^t = \sigma_\beta(g(w_t) - \epsilon)\nabla g_j(w_{j,\tau}^t) + (1 - \sigma_\beta(g(w_t) - \epsilon))\nabla f_j(w_{j,\tau}^t)$$

We analyze the squared distance to the optimal point $w^*$ as follows,

$$
\|w_{t+1} - w^*\|^2 = \left\| w_t - \eta \cdot \frac{1}{n} \sum_{j=1}^{n} \sum_{\tau=0}^{E-1} \nu_{j,\tau}^t - w^* \right\|^2
$$

$$
= \|w_t - w^*\|^2 + \underbrace{\eta^2 \left\| \frac{1}{n} \sum_{j=1}^{n} \sum_{\tau=0}^{E-1} \nu_{j,\tau}^t \right\|^2}_{Term-A} \underbrace{-2\eta \left\langle w_t - w^*, \frac{1}{n} \sum_{j=1}^{n} \sum_{\tau=0}^{E-1} \nu_{j,\tau}^t \right\rangle}_{Term-B} \tag{19}
$$

Firstly, we start with upper-bounding $Term - A$.

$$
Term - A = \eta^2 \left\| \frac{1}{n} \sum_{j=1}^{n} \sum_{\tau=0}^{E-1} \nu_{j,\tau}^t \right\|^2 \overset{Jensen's}{\leq} \eta^2 \frac{1}{n} \sum_{j=1}^{n} E \sum_{\tau=0}^{E-1} \|\nu_{j,\tau}^t\|^2
$$

$$
\overset{G-Lip \,\&\, \sigma_\beta(\cdot) \in [0,1]}{\leq} \eta^2 E^2 G^2.
$$

Now we aim to upper bound $Term - B$.

$$
Term - B = -2\eta \left\langle w_t - w^*, \frac{1}{n} \sum_{j=1}^{n} \sum_{\tau=0}^{E-1} \nu_{j,\tau}^t \right\rangle
$$

$$
= \frac{1}{n} \sum_{j=1}^{n} \sum_{\tau=0}^{E-1} -2\eta \left\langle w_t - w^*, \nu_{j,\tau}^t \right\rangle
$$

$$
= \frac{1}{n} \sum_{j=1}^{n} \sum_{\tau=0}^{E-1} \left[ \underbrace{-2\eta \left\langle w_t - w_{j,\tau}^t, \nu_{j,\tau}^t \right\rangle}_{Term-B_1} \underbrace{-2\eta \left\langle w_{j,\tau}^t - w^*, \nu_{j,\tau}^t \right\rangle)}_{Term-B_2} \right].
$$

Now we define $\sigma_\beta^t = \sigma_\beta(g(w_t) - \epsilon)$ before we start upper bounding $Term - B_1$ for any $\alpha > 0$.

$$
Term - B_1 = -2\eta \left\langle w_t - w_{j,\tau}^t, \nu_{j,\tau}^t \right\rangle
$$

$$
= -2\eta\sigma_\beta^t \left\langle w_t - w_{j,\tau}^t, g_j(w_{j,\tau}^t) \right\rangle - 2\eta(1 - \sigma_\beta^t) \left\langle w_t - w_{j,\tau}^t, f_j(w_{j,\tau}^t) \right\rangle
$$

$$
\overset{Young's}{\leq} \sigma_\beta^t \left[ \frac{\eta}{\alpha} \|w_t - w_{j,\tau}^t\|^2 + \eta\alpha \|\nabla g_j(w_{j,\tau}^t)\|^2 \right] + (1 - \sigma_\beta^t) \left[ \frac{\eta}{\alpha} \|w_t - w_{j,\tau}^t\|^2 \right.
$$

$$
\left. + \eta\alpha \|\nabla f_j(w_{j,\tau}^t)\|^2 \right]
$$

$$
\overset{G-Lip}{\leq} \frac{\eta}{\alpha} \|w_t - w_{j,\tau}^t\|^2 + \eta\alpha G^2.
$$

Similarly, we start to upper bound $Term - B_2$ as well.

$$
Term - B_2 = -2\eta \left\langle w_{j,\tau}^t - w^*, \nu_{j,\tau}^t \right\rangle)
$$

$$
= -2\eta\sigma_\beta^t \left\langle w_{j,\tau}^t - w^*, g_j(w_{j,\tau}^t) \right\rangle - 2\eta(1 - \sigma_\beta^t) \left\langle w_{j,\tau}^t - w^*, f_j(w_{j,\tau}^t) \right\rangle
$$

$$
\overset{Cvx.}{\leq} 2\eta\sigma_\beta^t \left( g_j(w^*) - g_j(w_{j,\tau}^t) \right) + 2\eta(1 - \sigma_\beta^t) \left( f_j(w^*) - f_j(w_{j,\tau}^t) \right)
$$

$$
\leq 2\eta\sigma_\beta^t \left[ g_j(w^*) - g_j(w_t) + \frac{1}{2\alpha} \|w_t - w_{j,\tau}^t\|^2 + \frac{\alpha}{2} G^2 \right] \qquad \text{(by Cvx., Young's, \& G-Lip)}
$$

$$
+ 2\eta(1 - \sigma_\beta^t) \left[ f_j(w^*) - f_j(w_t) + \frac{1}{2\alpha} \|w_t - w_{j,\tau}^t\|^2 + \frac{\alpha}{2} G^2 \right].
$$

Putting $Term - B_1$ and $Term - B_2$ back in $Term - B$, we get

$$Term - B \leq \frac{1}{n} \sum_{j=1}^{n} \sum_{\tau=0}^{E-1} \left[ \frac{2\eta}{\alpha} \|w_t - w_{j,\tau}^t\|^2 + 2\eta\alpha G^2 + 2\eta\sigma_\beta^t \left(g_j(w^*) - g_j(w_t)\right) \right.$$

$$\left. + 2\eta(1 - \sigma_\beta^t)\left(f_j(w^*) - f_j(w_t)\right)\right]$$

$$\overset{Lemma\ 2}{\leq} \frac{2}{3\alpha}\eta^3 E^3 G^2 + 2\eta\alpha E G^2 + 2\eta E \sigma_\beta^t \left(g(w^*) - g(w_t)\right)$$

$$+ 2\eta E(1 - \sigma_\beta^t)\left(f(w^*) - f(w_t)\right)$$

Substituting $Term - A$ and $Term - B$ back in eq. (19), we get for $\alpha = \eta$

$$\|w_{t+1} - w^*\|^2 \leq \|w_t - w^*\|^2 + \eta^2 E^2 G^2 + 2\eta^2 E G^2 + \frac{2}{3}\eta^2 E^3 G^2 + 2\eta E \sigma_\beta^t \left(g(w^*) - g(w_t)\right)$$

$$+ 2\eta E(1 - \sigma_\beta^t)\left(f(w^*) - f(w_t)\right)$$

Let $\mathcal{A} = \{t \in [T] | g(w_t) < \epsilon\}$ and $\mathcal{B} = [T] \backslash \mathcal{A} = \{t \in [T] | g(w_t) \geq \epsilon\}$. Note that for all $t \in \mathcal{B}$ it holds that $\sigma_\beta(g(w_t) - \epsilon) = 1$ and $g(w_t) - g(w^*) \geq \epsilon$. Further, for all $t \in \mathcal{A}$ if $\sigma_\beta(g(w_t) - \epsilon) \geq 0$ it holds that $g(w_t) - g(w^*) \geq g(w_t) \geq \epsilon - 1/\beta$. With these observations, using convexity of $f$ and $g$ and decomposing the sum over $t$ according to the definitions of $\mathcal{A}$ and $\mathcal{B}$ and division by $T$ yields,

$$\frac{D^2}{2\eta E} + \frac{1}{2}\eta E G^2 T + \eta G^2 T + \frac{1}{3}\eta E^2 G^2 T \geq \sum_{t \in \mathcal{A}} \sigma_\beta^t \left(g(w_t) - g(w^*)\right)$$

$$+ \sum_{t \in \mathcal{A}} (1 - \sigma_\beta^t)\left(f(w_t) - f(w^*)\right) + \sum_{t \in \mathcal{B}} \left(g(w_t) - g(w^*)\right).$$

Now choosing $\eta = \sqrt{\frac{D^2}{2G^2 E T \Gamma}}$, where $\Gamma = \frac{1}{2}E + 1 + \frac{1}{3}E^2$, we get

$$\sqrt{\frac{2D^2 G^2 T \Gamma}{E}} \geq \sum_{t \in \mathcal{A}} \sigma_\beta^t \left(g(w_t) - g(w^*)\right) + \sum_{t \in \mathcal{A}} (1 - \sigma_\beta^t)\left(f(w_t) - f(w^*)\right)$$

$$+ \sum_{t \in \mathcal{B}} \left(g(w_t) - g(w^*)\right) \tag{20}$$

$$\geq \sum_{t \in \mathcal{A}} (1 - \sigma_\beta^t)\left(f(w_t) - f(w^*)\right) + \epsilon|\mathcal{B}| + \left(\epsilon - \frac{1}{\beta}\right)\sum_{t \in \mathcal{A}} \sigma_\beta^t.$$

Similar to the previous proofs, we first need to find the smallest $\epsilon$ to ensure $\mathcal{A}$ is non-empty. So, to find a lower bound on $\epsilon$, assume $\mathcal{A}$ is empty in eq. (20) and observe that as long as condition $\sqrt{\frac{2D^2 G^2 \Gamma}{ET}} < \epsilon$ is met, $\mathcal{A}$ is non-empty. We choose to set $\epsilon = 2\sqrt{\frac{2D^2 G^2 \Gamma}{ET}}$.

Now, like before, we consider two cases based on the sign of $\sum_{t \in \mathcal{A}} \left(1 - \sigma_\beta^t\right)\left(f(w_t) - f(w^*)\right)$. As before, when the sum is non-positive, we are done by the definition of $\mathcal{A}$, which implies $0 < 1 - \sigma_\beta(g(w_t) - \epsilon) \leq 1$ and the convexity of $f$ and $g$.

Assuming the sum is positive, dividing eq. (20) by $\sum_{t \in \mathcal{A}} \left(1 - \sigma_\beta^t\right)$ (which by the definition of $\mathcal{A}$ is strictly positive), using convexity of $f$, and the definition of $\bar{w}$, we have

$$f(\bar{w}) - f(w^*) \leq \frac{0.5\epsilon T - \epsilon|\mathcal{B}| - (\epsilon - \frac{1}{\beta})\sum_{t \in \mathcal{A}} \sigma_\beta^t}{|\mathcal{A}| - \sum_{t \in \mathcal{A}} \sigma_\beta(g(w_t) - \epsilon)}$$

$$= \epsilon + \frac{-0.5\epsilon T + \beta^{-1}\sum_{t \in \mathcal{A}} \sigma_\beta^t}{|\mathcal{A}| - \sum_{t \in \mathcal{A}} \sigma_\beta^t},$$

where we used $|\mathcal{B}| = T - |\mathcal{A}|$.

Let us now find a lower bound on $\beta$ to ensure the second term in the bound is non-positive. Note this is done for simplicity, and as long as the second term is $\mathcal{O}(\epsilon)$, an $\epsilon$-solution can be found. Immediate calculations show the second term in the bound is non-positive when

$$\beta \geq \frac{2\sum_{t \in \mathcal{A}} \sigma_\beta^t}{\epsilon T}.$$

Since $\sum_{t \in \mathcal{A}} \sigma_\beta^t < T$, a sufficient (and highly conservative) condition for all $T \geq 1$ is to set $\beta \geq 2/\epsilon$. Thus, we proved the sub-optimality gap result. The feasibility result is immediate given the definition of $\mathcal{A}$ and the convexity of $g$. $\square$

### C.4 Main Theorems: FedSGM–bidirectional compression

#### C.4.1 Hard Switching — Full Participation

**Theorem 6.** *Consider the problem in eq. (1), where $\mathcal{X} = \mathbb{R}^d$ and Algorithm 1, under Assumption 1 and 3. Define $D := \|w_0 - w^*\|$ and*

$$\mathcal{A} = \{t \in [T] \mid g(w_t) \leq \epsilon\}, \qquad \bar{w} = \frac{1}{|\mathcal{A}|} \sum_{t \in \mathcal{A}} w_t.$$

*Then, if*

$$\epsilon = \sqrt{\frac{2D^2 G^2 \Gamma}{ET}}, \text{ and } \eta = \sqrt{\frac{D^2}{2G^2 ET\Gamma}}, \text{ where } \Gamma = 2E^2 + 2E\frac{\sqrt{1-q}}{q} + 4E\frac{\sqrt{10(1-q_0)}}{q_0 q}$$

*it holds that $\mathcal{A}$ is nonempty, $\bar{w}$ is well-defined, and $\bar{w}$ is an $\epsilon$-solution for $P$.*

*Proof.* We adopt the virtual iterate framework, a commonly used analytical tool in recent works such as Stich & Karimireddy (2019); Koloskova et al. (2023); Mishchenko et al. (2023); Islamov et al. (2024b; 2025). Specifically, we define the virtual sequence $\hat{w}_t := x_t - \eta e_t$, initialized as $\hat{w}_0 = w_0$ where $e_t = \frac{1}{n} \sum_{j=1}^n e_j^t$. This construction leads to a simplified update rule:

$$\hat{w}_{t+1} := \hat{w}_t - \eta \nu_t, \quad \text{where} \quad \nu_t = \frac{1}{n} \sum_{j=1}^n \sum_{\tau=0}^{E-1} \nu_{j,\tau}^t.$$

To see this, note that

$$\begin{aligned}
\hat{w}_{t+1} = x_{t+1} - \eta e_{t+1} &= (x_t - \eta \nu_t) - \eta(e_t + \nu_t - v_t) \\
&= (x_t - \eta e_t) - \eta \nu_t \\
&= \hat{w}_t - \eta \nu_t.
\end{aligned} \tag{21}$$

In addition, we define the auxiliary error term $\hat{e}_t = x_t - w_t$, which captures the discrepancy introduced by downlink (server-to-client) compression.

Starting with the update rule presented in eq. (21), we analyze the squared distance to the optimal point $w^*$ as follows,

$$\begin{aligned}
\|\hat{w}_{t+1} - w^*\|^2 &= \|\hat{w}_t - \eta \nu_t - w^*\|^2 \\
&= \|\hat{w}_t - w^*\|^2 + \underbrace{\eta^2 \|\nu_t\|^2}_{Term-A} \underbrace{-2\eta \langle \hat{w}_t - w^*, \nu_t \rangle}_{Term-B}.
\end{aligned} \tag{22}$$

Firstly, we start with an upper bound for $Term - A$.

$$Term - A = \eta^2 \|\nu_t\|^2 = \eta^2 \left\| \frac{1}{n} \sum_{j=1}^n \sum_{\tau=0}^{E-1} \nu_{j,\tau}^t \right\|^2 \overset{Jensen's}{\leq} \eta^2 \frac{1}{n} \sum_{j=1}^n E \sum_{\tau=0}^{E-1} \|\nu_{j,\tau}^t\|^2$$

$$\overset{G-Lip}{\leq} \eta^2 E^2 G^2.$$

Now, we start to upper bound $Term - B$.

$$\begin{aligned}
Term - B &= -2\eta \langle \hat{w}_t - w^*, \nu_t \rangle \\
&= -2\eta \langle \hat{w}_t - x_t + x_t - w^*, \nu_t \rangle \\
&= -2\eta \langle \hat{w}_t - x_t, \nu_t \rangle - 2\eta \langle x_t - w^*, \nu_t \rangle \\
&= 2\eta^2 \langle e_t, \nu_t \rangle + 2\eta \langle w_t - x_t, \nu_t \rangle - 2\eta \langle w_t - w^*, \nu_t \rangle
\end{aligned}$$

$$= \frac{2\eta^2}{n} \sum_{j=1}^{n} \sum_{\tau=0}^{E-1} \left\langle e_t, \nu_{j,\tau}^t \right\rangle + \frac{2\eta}{n} \sum_{j=1}^{n} \sum_{\tau=0}^{E-1} \left\langle -\hat{e}_t, \nu_{j,\tau}^t \right\rangle - \frac{2\eta}{n} \sum_{j=1}^{n} \sum_{\tau=0}^{E-1} \left\langle w_t - w^*, \nu_{j,\tau}^t \right\rangle$$

$$\overset{Cauchy-Schwarz}{\leq} \frac{2\eta^2}{n} \sum_{j=1}^{n} \sum_{\tau=0}^{E-1} \|e_t\| \|\nu_{j,\tau}^t\| + \frac{2\eta}{n} \sum_{j=1}^{n} \sum_{\tau=0}^{E-1} \|\hat{e}_t\| \|\nu_{j,\tau}^t\|$$

$$- \frac{2\eta}{n} \sum_{j=1}^{n} \sum_{\tau=0}^{E-1} \left\langle w_t - w^*, \nu_{j,\tau}^t \right\rangle$$

$$\overset{G-Lip}{\leq} \underbrace{\frac{2\eta^2 G}{n} \sum_{j=1}^{n} \sum_{\tau=0}^{E-1} \|e_t\|}_{Term-B_1} + \underbrace{\frac{2\eta G}{n} \sum_{j=1}^{n} \sum_{\tau=0}^{E-1} \|\hat{e}_t\|}_{Term-B_2} \underbrace{- \frac{2\eta}{n} \sum_{j=1}^{n} \sum_{\tau=0}^{E-1} \left\langle w_t - w^*, \nu_{j,\tau}^t \right\rangle}_{Term-B_3}.$$

We can upper bound $Term - B_3$ using Lemma 1. We now proceed to upper bound $Term - B_1$, considering the expectation taken over the stochasticity of the compression mechanisms.

$$\mathbb{E}[\|e_{t+1}\|^2] = \mathbb{E}\left[\left\| \frac{1}{n} \sum_{j=1}^{n} e_j^{t+1} \right\|^2\right]$$

$$\overset{Jensen's}{\leq} \frac{1}{n} \sum_{j=1}^{n} \mathbb{E}\left[\|e_j^{t+1}\|^2\right]$$

$$= \frac{1}{n} \sum_{j=1}^{n} \mathbb{E}\left[\left\| e_j^t + \sum_{\tau=0}^{E-1} \nu_{j,\tau}^t - \mathcal{C}_j\left(e_j^t + \sum_{\tau=0}^{E-1} \nu_{j,\tau}^t\right) \right\|^2\right]$$

$$\overset{Assump. \ 3}{\leq} \frac{1-q}{n} \sum_{j=1}^{n} \mathbb{E}\left[\left\| e_j^t + \sum_{\tau=0}^{E-1} \nu_{j,\tau}^t \right\|^2\right]$$

$$\overset{Young's}{\leq} (1-q)(1+\gamma) \frac{1}{n} \sum_{j=1}^{n} \mathbb{E}\left[\|e_j^t\|^2\right] + (1-q)(1+\gamma^{-1}) \frac{1}{n} \sum_{j=1}^{n} \left\| \sum_{\tau=0}^{E-1} \nu_{j,\tau}^t \right\|^2$$

$$\overset{Jensen's+G-Lip}{\leq} (1-q)(1+\gamma) \frac{1}{n} \sum_{j=1}^{n} \mathbb{E}\left[\|e_j^t\|^2\right] + (1-q)(1+\gamma^{-1})E^2 G^2$$

Let $A := (1-q)(1+\gamma)$ and $B := (1-q)(1+\gamma^{-1})E^2 G^2$. Also, define $\bar{E}_t := \frac{1}{n} \sum_{j=1}^{n} \mathbb{E}\left[\|e_j^t\|^2\right]$. Then,

$$\mathbb{E}[\|e_{t+1}\|^2] \leq \bar{E}_{t+1} \leq A \cdot \bar{E}_t + B.$$

So, for $\gamma = \frac{q}{2(1-q)}$, since $e_j^0 = 0$ implying $\bar{E}_0$, unrolling the recursion gives,

$$\mathbb{E}[\|e_{t+1}\|^2] \leq \sum_{l=0}^{t} A^{t-l} \cdot B$$

$$\leq \sum_{l=0}^{t} [(1-q)(1+\gamma)]^{t-l} (1-q)(1+\gamma^{-1})E^2 G^2$$

$$\overset{geometric \ sum}{\leq} \frac{(1-q)(1+\gamma^{-1})}{1 - (1-q)(1+\gamma)} E^2 G^2$$

$$= \frac{(1-q)(1+\gamma^{-1})}{q - \gamma(1-q)} E^2 G^2$$

$$\overset{\gamma=\frac{q}{2(1-q)}}{=} \frac{2(1-q)(2-q)}{q^2} E^2 G^2$$

$$\overset{(2-q)\leq 2}{\leq} \frac{4(1-q)}{q^2}E^2G^2.$$

So, we have for $Term - B_1$,

$$\mathbb{E}_{\mathcal{C}_j}[Term - B_1] \leq \frac{2\eta^2 G}{n} \sum_{j=1}^{n} \sum_{\tau=0}^{E-1} \sqrt{\frac{4(1-q)}{q^2}E^2G^2}$$

$$= 4\eta^2 E^2 G^2 \frac{\sqrt{(1-q)}}{q}.$$

Similarly, we start to deal with $\mathbb{E}[\|\hat{e}_{t+1}\|^2]$, to bound $Term - B_2$.

$$\mathbb{E}[\|\hat{e}_{t+1}\|^2] = \mathbb{E}\left[\|x_{t+1} - w_{t+1}\|^2\right]$$

$$= \mathbb{E}\left[\|x_{t+1} - w_t - \mathcal{C}_0(x_{t+1} - w_t)\|^2\right]$$

$$\overset{Assump. 3}{\leq} (1-q_0)\mathbb{E}\left[\|x_{t+1} - w_t\|^2\right] \qquad (23)$$

$$= (1-q_0)\mathbb{E}\left[\|x_t - \eta v_t - w_t\|^2\right] = (1-q_0)\mathbb{E}\left[\|\hat{e}_t - \eta v_t\|^2\right]$$

$$\overset{Young's}{\leq} (1-q_0)(1+\hat{\gamma})\mathbb{E}\left[\|\hat{e}_t\|^2\right] + \eta^2(1-q_0)(1+\hat{\gamma}^{-1})\mathbb{E}\left[\|v_t\|^2\right]$$

$$\mathbb{E}\left[\|v_t\|^2\right] = \mathbb{E}\left[\left\|\frac{1}{n}\sum_{j=1}^{n}\mathcal{C}_j\left(e_j^t + \sum_{\tau=0}^{E-1}\nu_{j\tau}^t\right)\right\|^2\right]$$

$$\overset{Jensen's+Young's}{\leq} \frac{2}{n}\sum_{j=1}^{n}\mathbb{E}\left[\left\|\mathcal{C}_j\left(e_j^t + \sum_{\tau=0}^{E-1}\nu_{j\tau}^t\right) - \left(e_j^t + \sum_{\tau=0}^{E-1}\nu_{j\tau}^t\right)\right\|^2\right]$$

$$+ \frac{2}{n}\sum_{j=1}^{n}\mathbb{E}\left[\left\|e_j^t + \sum_{\tau=0}^{E-1}\nu_{j\tau}^t\right\|^2\right]$$

$$\overset{Assump. 3}{\leq} \frac{2(2-q)}{n}\sum_{j=1}^{n}\mathbb{E}\left[\left\|e_j^t + \sum_{\tau=0}^{E-1}\nu_{j\tau}^t\right\|^2\right]$$

$$\overset{(2-q)\leq 2}{\leq} \frac{4}{n}\sum_{j=1}^{n}\mathbb{E}\left[\left\|e_j^t + \sum_{\tau=0}^{E-1}\nu_{j\tau}^t\right\|^2\right]$$

$$\overset{Young's}{\leq} \frac{8}{n}\sum_{j=1}^{n}\left[\mathbb{E}\left[\|e_j^t\|^2\right] + \mathbb{E}\left[\left\|\sum_{\tau=0}^{E-1}\nu_{j\tau}^t\right\|^2\right]\right]$$

$$\overset{Jensen's+G-Lip}{\leq} \frac{8}{n}\sum_{j=1}^{n}\left[\frac{4(1-q)}{q^2}E^2G^2 + E^2G^2\right]$$

$$= \frac{32(1-q)}{q^2}E^2G^2 + 8E^2G^2$$

$$\overset{(1-q)\leq 1}{\leq} \frac{40E^2G^2}{q^2}.$$

Then continuing again with 23, we have

$$\mathbb{E}[\|\hat{e}_{t+1}\|^2] \leq (1-q_0)(1+\hat{\gamma})\mathbb{E}\left[\|\hat{e}_t\|^2\right] + \eta^2(1-q_0)(1+\hat{\gamma}^{-1})\frac{40E^2G^2}{q^2}.$$

Same as before for $\hat{\gamma} = \frac{q_0}{2(1-q_0)}$, since $\hat{e}_0 = 0$, unrolling the recursion above we get

$$\mathbb{E}[\|\hat{e}_{t+1}\|^2] \leq \sum_{l=0}^{t}[(1-q_0)(1+\hat{\gamma})]^{t-l}\eta^2(1-q_0)(1+\hat{\gamma}^{-1})\frac{40E^2G^2}{q^2}$$

$$\overset{geometric\ sum}{\leq} \frac{(1-q_0)(1+\hat{\gamma}^-1)}{1-(1-q_0)(1+\hat{\gamma})}\eta^2\frac{40E^2G^2}{q^2}$$

$$\leq \eta^2\frac{160(1-q_0)E^2G^2}{q_0^2q^2}.$$

Therefore, $Term - B_2$, we have

$$\mathbb{E}_{\mathcal{C}_j,\mathcal{C}_0}\left[Term - B_2\right] \leq \frac{2\eta G}{n}\sum_{j=1}^{n}\sum_{\tau=0}^{E-1}\sqrt{\frac{160(1-q_0)E^2G^2}{q_0^2q^2}}$$

$$= 8\eta^2E^2G^2\frac{\sqrt{10(1-q_0)}}{q_0q}$$

Now, we take the expectation on both sides of eq. (22) with respect to the compression operators and plug in the value of $Term - B_1$, $Term - B_2$, and $Term - B_3$, we get

$$\|\hat{w}_{t+1} - w^*\|^2 \leq \|\hat{w}_t - w^*\|^2 + \eta^2E^2G^2 + 4\eta^2E^2G^2\frac{\sqrt{(1-q)}}{q} + 8\eta^2E^2G^2\frac{\sqrt{10(1-q_0)}}{q_0q}$$

$$+ \eta\alpha EG^2 + \frac{\eta}{\alpha n}\sum_{j=1}^{n}\sum_{\tau=0}^{E-1}\|w_t - w_{j,\tau}^t\|^2 + \frac{2\eta}{n}\sum_{j=1}^{n}\sum_{\tau=0}^{E-1}(f_j(w^*) - f_j(w_{j,\tau}^t))\mathbb{1}\{t \in \mathcal{A}\}$$

$$+ \frac{2\eta}{n}\sum_{j=1}^{n}\sum_{\tau=0}^{E-1}(g_j(w^*) - g_j(w_{j,\tau}^t))\mathbb{1}\{t \in \mathcal{B}\}$$

$$(24)$$

Now our goal is to handle the term $f_j(w^*) - f_j(w_{j,\tau}^t)$, so we first rewrite,

$$f_j(w_{j,\tau}^t) \geq f_j(w_t) + \langle\nabla f_j(w_t), w_{j,\tau}^t - w_t\rangle \quad \text{(by convexity)}$$

$$\Rightarrow f_j(w^*) - f_j(w_{j,\tau}^t) \leq f_j(w^*) - f_j(w_t) - \langle\nabla f_j(w_t), w_{j,\tau}^t - w_t\rangle.$$

Using Young's inequality with parameter $\alpha > 0$, we get

$$-\langle\nabla f_j(w_t), w_{j,\tau}^t - w_t\rangle \leq \frac{1}{2\alpha}\|w_{j,\tau}^t - w_t\|^2 + \frac{\alpha}{2}\|\nabla f_j(w_t^t)\|^2.$$

Thus, we get,

$$f_j(w^*) - f_j(w_{j,\tau}^t) \leq f_j(w^*) - f_j(w_t) + \frac{1}{2\alpha}\|w_t - w_{j,\tau}^t\|^2 + \frac{\alpha}{2}\|\nabla f_j(w_t)\|^2$$

$$\overset{G-Lip}{\leq} f_j(w^*) - f_j(w_t) + \frac{1}{2\alpha}\|w_t - w_{j,\tau}^t\|^2 + \frac{\alpha}{2}G^2.$$

Similarly, we can handle the other term with $g$ and get,

$$g_j(w^*) - g_j(w_{j,\tau}^t) \leq g_j(w^*) - g_j(w_t) + \frac{1}{2\alpha}\|w_t - w_{j,\tau}^t\|^2 + \frac{\alpha}{2}G^2.$$

So, putting these inequalities back in eq. (24) along with the use of Lemma 2 and eq. (1), we get

$$\|\hat{w}_{t+1} - w^*\|^2 \leq \|\hat{w}_t - w^*\|^2 + \eta^2E^2G^2 + 4\eta^2E^2G^2\frac{\sqrt{(1-q)}}{q} + 8\eta^2E^2G^2\frac{\sqrt{10(1-q_0)}}{q_0q}$$

$$+ 2\eta\alpha EG^2 + \frac{2}{3\alpha}\eta^3E^3G^2 + 2\eta E(f(w^*) - f(w_t))\mathbb{1}\{t \in \mathcal{A}\} + 2\eta E(g(w^*) - g(w_t))\mathbb{1}\{t \in \mathcal{B}\}.$$

Now, for $\alpha = \eta$, and rearranging the terms we get

$$(f(w_t) - f(w^*))\mathbb{1}\{t \in \mathcal{A}\} + (g(w_t) - g(w^*))\mathbb{1}\{t \in \mathcal{B}\} \leq \frac{\|\hat{w}_t - w^*\|^2 - \|\hat{w}_{t+1} - w^*\|^2}{2\eta E}$$

$$+ \frac{\eta}{2}EG^2 + 2\eta EG^2\frac{\sqrt{(1-q)}}{q} + 4\eta EG^2\frac{\sqrt{10(1-q_0)}}{q_0q} + \eta G^2 + \frac{1}{3}\eta E^2G^2$$

Defining $D := \|w_0 - w^*\|$ and summing the expression above for $t = 0, 1, \cdots, T - 1$ and then dividing by $T$, we get

$$\frac{1}{T} \sum_{t \in \mathcal{A}} (f(w_t) - f(w^*)) + \frac{1}{T} \sum_{t \in \mathcal{B}} (g(w_t) - g(w^*)) \leq \frac{D^2}{2\eta ET} + \frac{\eta}{2} EG^2 + \eta G^2 + \frac{1}{3} \eta E^2 G^2$$

$$+ 2\eta EG^2 \frac{\sqrt{(1-q)}}{q} + 4\eta EG^2 \frac{\sqrt{10(1-q_0)}}{q_0 q}.$$

Now choosing $\eta = \sqrt{\frac{D^2}{2G^2 ET\Gamma}}$, where $\Gamma = \frac{1}{2} E + 1 + \frac{1}{3} E^2 + 2E \frac{\sqrt{1-q}}{q} + 4E \frac{\sqrt{10(1-q_0)}}{q_0 q}$, we get

$$\frac{1}{T} \sum_{t \in \mathcal{A}} (f(w_t) - f(w^*)) + \frac{1}{T} \sum_{t \in \mathcal{B}} (g(w_t) - g(w^*)) \leq \sqrt{\frac{2D^2 G^2 \Gamma}{ET}}.$$

Note that when $\epsilon$ is sufficiently large, $\mathcal{A}$ is nonempty. Assuming an empty $\mathcal{A}$, we can find the largest "bad" $\epsilon$:

$$\epsilon_{bad} < \frac{1}{T} \sum_{t \in \mathcal{B}} g(w_t) - g(w^*) \leq \sqrt{\frac{2D^2 G^2 \Gamma}{ET}}$$

Thus, let us set $\epsilon = (N + 1)\sqrt{\frac{2D^2 G^2 \Gamma}{ET}}$ for some $N \geq 0$. With this choice, $\mathcal{A}$ is guaranteed to be nonempty.

Now, we consider two cases. Either $\sum_{t \in \mathcal{A}} f(w_t) - f(w^*) \leq 0$ which implies by the convexity of $f$ and $g$ for $\bar{w} = \frac{1}{|\mathcal{A}|} \sum_{t \in \mathcal{A}} w_t$ we have

$$f(\bar{w}) - f(w^*) \leq 0 < \epsilon, \qquad g(\bar{w}) \leq \epsilon. \tag{25}$$

Otherwise, if $\sum_{t \in \mathcal{A}} f(w_t) - f(w^*) > 0$, then

$$\sqrt{\frac{2D^2 G^2 \Gamma}{ET}} \geq \frac{1}{T} \sum_{t \in \mathcal{A}} f(w_t) - f(w^*) + \frac{1}{T} \sum_{t \in \mathcal{B}} g(w_t) - g(w^*)$$

$$> \frac{1}{T} \sum_{t \in \mathcal{A}} f(w_t) - f(w^*) + \frac{1}{T} \sum_{t \in \mathcal{B}} \epsilon$$

$$= \frac{|\mathcal{A}|}{T} \frac{1}{|\mathcal{A}|} \sum_{t \in \mathcal{A}} f(w_t) - f(w^*) + (1 - \frac{|\mathcal{A}|}{T})\epsilon \tag{26}$$

$$\geq \frac{|\mathcal{A}|}{T} \big(f(\bar{w}) - f(w^*)\big) + (1 - \frac{|\mathcal{A}|}{T})\epsilon.$$

By rearranging

$$\frac{|\mathcal{A}|}{T} \big(f(\bar{w}) - f(w^*) - \epsilon\big) < \sqrt{\frac{2D^2 G^2 \Gamma}{ET}} - \epsilon \leq -N\sqrt{\frac{2D^2 G^2 \Gamma}{ET}}, \tag{27}$$

implying $f(\bar{w}) - f(w^*) < \epsilon$ and further by convexity of $g$ for $\bar{w} = \frac{1}{|\mathcal{A}|} \sum_{t \in \mathcal{A}} w_t$, we also have $g(\bar{w}) \leq \epsilon$. $\qquad \square$

### C.4.2 HARD SWITCHING — PARTIAL PARTICIPATION

**Lemma 9.** *Assume Assumption 1 holds and also assume that the compressors $\{\mathcal{C}\}_{j=1}^n$ are deterministic, such as Top$-K$. Then, for $t \geq 0$ and $j \in [n]$, we have $\|\tilde{e}_t\| \leq \frac{n}{m} \frac{\sqrt{4(1-q) E^2 G^2}}{q}$ a.s.*

*Proof.* Fix a client $j$ and set $e_t := \|e_j^t\|^2$. With the chosen $\gamma = \frac{q}{2(1-q)}$ (using results from previous proofs and properties of the compressors),

$$e_{t+1} \leq \underbrace{(1-q)(1+\gamma)}_{=1-\frac{q}{2}} e_t + \underbrace{(1-q)\Big(1+\frac{1}{\gamma}\Big) E^2 G^2}_{= \frac{(1-q)(2-q)}{q} E^2 G^2} \quad \text{if } j \in \mathcal{S}_t \quad \text{and} \quad e_{t+1} = e_t \text{ otherwise.}$$

To find a bound that holds regardless of whether the client is selected in each round, we define a deterministic sequence $x_k$ that acts as a worst-case scenario. We can think of $k$ as the number of times the client has been selected for an update, starting with $x_0 = 0$.

$$x_{k+1} = \left(1 - \frac{q}{2}\right) x_k + \frac{(1-q)(2-q)}{q} E^2 G^2$$

This sequence describes the per-client worst-case scenario. By induction, we can prove that the client's actual error, $e_t$, is always bounded by this deterministic sequence's value at the number of updates the client has received, $N_t$: $e_t \leq x_{N_t}$ for all $t$ a.s. This is because the error $e_t$ either decreases or remains constant (if the client is not selected), while the deterministic sequence $x_k$ always progresses forward with each update.

Since the update factor $(1 - q/2)$ is less than 1, the deterministic sequence $x_k$ converges to a stable, maximum value as $k \to \infty$. This maximum value acts as a global upper bound for the error of any single client.

$$e_t \leq \lim_{k \to \infty} x_k = \frac{\frac{(1-q)(2-q)}{q} E^2 G^2}{1 - \left(1 - \frac{q}{2}\right)} = \frac{\frac{(1-q)(2-q)}{q} E^2 G^2}{\frac{q}{2}} = \frac{2(1-q)(2-q)}{q^2} E^2 G^2 \quad \text{for all } t \text{ a.s.}$$

This result provides a tight bound on the squared error for any individual client at any point in time. Now, we get

$$\|\tilde{e}_t\|^2 = \left\| \frac{1}{m} \sum_{j=1}^{n} e_j^t \right\|^2$$

$$\leq \left(\frac{n}{m}\right)^2 \frac{1}{n} \sum_{j=1}^{n} \|e_j^t\|^2$$

$$\leq \left(\frac{n}{m}\right)^2 \frac{2(1-q)(2-q)}{q^2} E^2 G^2$$

$$\overset{(2-q) \leq 2}{\leq} \left(\frac{n}{m}\right)^2 \frac{4(1-q)}{q^2} E^2 G^2.$$

$\square$

**Theorem 7.** *Consider the optimization problem in equation 1 and Algorithm 1 under Assumptions 1, 2, 4, and deterministic compressors $\{C_j\}_{j=0}^n$ (e.g. Top$-K$). Additionally, assume that at each global round, a subset of clients is sampled uniformly at random with replacement. Define*

$$\mathcal{A} := \left\{ t \in [T] \,\Big|\, \hat{G}(w_t) \leq \epsilon \right\}, \qquad \bar{w} := \frac{1}{|\mathcal{A}|} \sum_{t \in \mathcal{A}} w_t.$$

*Suppose the step size $\eta$ and constraint threshold $\epsilon$ are set as*

$$\eta = \sqrt{\frac{D^2}{2G^2 E T \Gamma}} \quad \text{and} \quad \epsilon = \sqrt{\frac{2D^2 G^2 \Gamma}{ET}} + \frac{n}{m} \frac{2DG\sqrt{1-q}}{qT} + \frac{4GD}{\sqrt{mT}} \sqrt{2 \log\left(\frac{3}{\delta}\right)} + 2\sigma \sqrt{\frac{2}{m} \log\left(\frac{6T}{\delta}\right)},$$

*where $\Gamma = 2E^2 + 16E \frac{n}{m} \frac{\sqrt{10(1-q)(1-q_0)}}{q_0 q^2} + 8E \frac{\sqrt{10(1-q_0)}}{q_0 q} + \frac{20E}{q^2} + \frac{n}{m} \frac{4E\sqrt{10(1-q)}}{q^2}$ and $\delta \in (0,1)$ is a confidence parameter. Then, with probability at least $1 - \delta$, the following hold:*

1. *The set $\mathcal{A}$ is non-empty, so $\bar{w}$ is well-defined.*

2. *The average iterate $\bar{w}$ satisfies*

$$f(\bar{w}) - f(w^*) \leq \epsilon, \qquad g(\bar{w}) \leq \epsilon.$$

*Proof.* Using Algorithm 1, we have that $x_{t+1} = \Pi_{\mathcal{X}}(x_t - \eta v_t)$. We also define $\tilde{e}_t := \frac{1}{m}\sum_{j=1}^{n} e_j^t$ and $\hat{e}_t = x_t - w_t$. Additionally, we follow all symbols used in the algorithm and other proofs as is. Any new notation will be mentioned when it comes.

$$\|x_{t+1} - w^*\|^2 \overset{(1-Lip\ \Pi_{\mathcal{X}})}{\leq} \|x_t - \eta v_t - w^*\|^2$$
$$= \|x_t - w^*\|^2 \underbrace{-2\eta \langle x_t - w^*, v_t\rangle}_{Term-1} + \underbrace{\eta^2 \|v_t\|^2}_{Term-2}$$

Now, we bound

$$\|v_t\|^2 = \left\| \frac{1}{m} \sum_{j \in \mathcal{S}_t} \mathcal{C}_j \left( e_j^t + \sum_{\tau=0}^{E-1} \nu_{j\tau}^t \right) \right\|^2$$

$$\overset{Jensen's+Young's}{\leq} \frac{2}{m} \sum_{j \in \mathcal{S}_t} \left\| \mathcal{C}_j \left( e_j^t + \sum_{\tau=0}^{E-1} \nu_{j\tau}^t \right) - \left( e_j^t + \sum_{\tau=0}^{E-1} \nu_{j\tau}^t \right) \right\|^2 + \frac{2}{m} \sum_{j \in \mathcal{S}_t} \left\| e_j^t + \sum_{\tau=0}^{E-1} \nu_{j\tau}^t \right\|^2$$

$$\leq \frac{2(2-q)}{m} \sum_{j \in \mathcal{S}_t} \left\| e_j^t + \sum_{\tau=0}^{E-1} \nu_{j\tau}^t \right\|^2$$

$$\overset{(2-q)\leq 2}{\leq} \frac{4}{m} \sum_{j \in \mathcal{S}_t} \left\| e_j^t + \sum_{\tau=0}^{E-1} \nu_{j\tau}^t \right\|^2$$

$$\overset{Young's}{\leq} \frac{8}{m} \sum_{j \in \mathcal{S}_t} \left[ \|e_j^t\|^2 + \left\| \sum_{\tau=0}^{E-1} \nu_{j\tau}^t \right\|^2 \right]$$

$$\overset{Jensen's+G-Lip+Lemma\ 9}{\leq} \frac{8}{m} \sum_{j \in \mathcal{S}_t} \left[ \frac{4(1-q)}{q^2} E^2 G^2 + E^2 G^2 \right]$$

$$= \frac{32(1-q)}{q^2} E^2 G^2 + 8E^2 G^2$$

$$\overset{(1-q)\leq 1}{\leq} \frac{40 E^2 G^2}{q^2}$$

So,

$$Term - 2 \leq \frac{40 \eta^2 E^2 G^2}{q^2}$$

Now, upper-bounding $Term - 1$.

$Term - 1 = -2\eta \langle x_t - w^*, v_t\rangle$

$= -2\eta \langle w_t - w^*, v_t\rangle - 2\eta \langle x_t - w_t, v_t\rangle$

$= -2\eta \langle w_t - w^*, v_t\rangle - 2\eta \langle \hat{e}_t, v_t\rangle$

$= -2\eta \left\langle w_t - w^*, \frac{1}{m} \sum_{j \in \mathcal{S}_t} (\Delta_j^t + e_j^t - e_j^{t+1}) \right\rangle - 2\eta \langle \hat{e}_t, v_t\rangle$

$= -2\eta \left\langle w_t - w^*, \frac{1}{m} \sum_{j \in \mathcal{S}_t} \Delta_j^t \right\rangle - 2\eta \left\langle w_t - w^*, \frac{1}{m} \sum_{j \in \mathcal{S}_t} (e_j^t - e_j^{t+1}) \right\rangle - 2\eta \langle \hat{e}_t, v_t\rangle$

$= -2\eta \langle w_t - w^*, \bar{\nu}_t\rangle + 2\eta \langle w_t - w^*, \bar{\nu}_t - \nu_t\rangle - 2\eta \left\langle w_t - w^*, \frac{1}{m} \sum_{j \in [n]} (e_j^t - e_j^{t+1}) \right\rangle - 2\eta \langle \hat{e}_t, v_t\rangle$

$= -2\eta \langle w_t - w^*, \bar{\nu}_t\rangle + 2\eta \langle w_t - w^*, \bar{\nu}_t - \nu_t\rangle - 2\eta \langle w_t - w^*, \tilde{e}_t - \tilde{e}_{t+1}\rangle - 2\eta \langle \hat{e}_t, v_t\rangle$

$= -2\eta \langle w_t - w^*, \bar{\nu}_t\rangle + 2\eta \langle w_t - w^*, \bar{\nu}_t - \nu_t\rangle - 2\eta \langle w_t - x_t, \tilde{e}_t - \tilde{e}_{t+1}\rangle - 2\eta \langle x_t - w^*, \tilde{e}_t - \tilde{e}_{t+1}\rangle$

$\quad - 2\eta \langle \hat{e}_t, v_t\rangle$

$$= -2\eta \langle w_t - w^*, \bar{\nu}_t \rangle + 2\eta \langle w_t - w^*, \bar{\nu}_t - \nu_t \rangle + \underbrace{2\eta \langle \hat{e}_t, \tilde{e}_t - \tilde{e}_{t+1} \rangle}_{Term-1-1} - 2\eta \langle x_t - w^*, \tilde{e}_t - \tilde{e}_{t+1} \rangle \underbrace{-2\eta \langle \hat{e}_t, v_t \rangle}_{Term-1-2}$$

Now bounding $Term - 1 - 1$.

$$\begin{aligned}
Term - 1 - 1 &= 2\eta \langle \hat{e}_t, \tilde{e}_t - \tilde{e}_{t+1} \rangle \\
&= 2\eta \langle \hat{e}_t, \tilde{e}_t \rangle + 2\eta \langle \hat{e}_t, \tilde{e}_{t+1} \rangle \\
&\overset{Cauchy-Schwarz}{\leq} 2\eta \|\hat{e}_t\| \|\tilde{e}_t\| + 2\eta \|\hat{e}_t\| \|\tilde{e}_{t+1}\| \\
&\overset{Lemma\ 9}{\leq} 8\eta E G \frac{n}{m} \frac{\sqrt{(1-q)}}{q} \|\hat{e}_t\|
\end{aligned}$$

Now we bound $\|\hat{e}_{t+1}\|^2$.

$$\begin{aligned}
\|\hat{e}_{t+1}\|^2 &= \|x_{t+1} - w_{t+1}\|^2 \\
&= \|x_{t+1} - w_t - \mathcal{C}_0(x_{t+1} - w_t)\|^2 \\
&\leq (1-q_0)\|x_{t+1} - w_t\|^2 \\
&\leq (1-q_0)\|x_{t+1} - x_t + x_t - w_t\|^2 \\
&\overset{Young's}{\leq} (1-q_0)(1+\hat{\gamma})\|\hat{e}_t\|^2 + (1-q_0)(1+\hat{\gamma}^{-1})\|x_{t+1} - x_t\|^2 \\
&\overset{1-Lip\ \Pi_{\mathcal{X}}}{\leq} (1-q_0)(1+\hat{\gamma})\|\hat{e}_t\|^2 + (1-q_0)(1+\hat{\gamma}^{-1})\|x_t - \eta v_t - x_t\|^2 \\
&\leq (1-q_0)(1+\hat{\gamma})\|\hat{e}_t\|^2 + \eta^2(1-q_0)(1+\hat{\gamma}^{-1})\|v_t\|^2
\end{aligned}$$

Plugging $\|v_t\|^2$, we have

$$\|\hat{e}_{t+1}\|^2 \leq (1-q_0)(1+\hat{\gamma})\|\hat{e}_t\|^2 + \eta^2(1-q_0)(1+\hat{\gamma}^{-1})\frac{40E^2G^2}{q^2}.$$

Same as before for $\hat{\gamma} = \frac{q_0}{2(1-q_0)}$, since $\hat{e}_0 = 0$, unrolling the recursion above we get

$$\begin{aligned}
\|\hat{e}_{t+1}\|^2 &\leq \sum_{l=0}^{t} [(1-q_0)(1+\hat{\gamma})]^{t-l} \eta^2(1-q_0)(1+\hat{\gamma}^{-1})\frac{40E^2G^2}{q^2} \\
&\overset{geometric\ sum}{\leq} \frac{(1-q_0)(1+\hat{\gamma}^{-1})}{1-(1-q_0)(1+\hat{\gamma})}\eta^2\frac{40E^2G^2}{q^2} \\
&\leq \eta^2\frac{160(1-q_0)E^2G^2}{q_0^2 q^2}.
\end{aligned}$$

Now, plugging the $\|\hat{e}_t\|$ back in $Term - 1 - 1$, we get

$$Term - 1 - 1 \leq 32\eta^2 E^2 G^2 \frac{n}{m} \frac{\sqrt{10(1-q)(1-q_0)}}{q_0 q^2}.$$

Similarly, we can bound $Term - 1 - 2$.

$$\begin{aligned}
Term - 1 - 2 &= -2\eta \langle \hat{e}_t, v_t \rangle \\
&\overset{Cauchy-Schwarz}{\leq} 2\eta \|\hat{e}_t\| \|v_t\| \\
&\leq 16\eta^2 E^2 G^2 \frac{\sqrt{10(1-q_0)}}{q_0 q}
\end{aligned}$$

So, for $Term - 1$, we get

$$\begin{aligned}
Term - 1 &\leq -2\eta \langle w_t - w^*, \bar{\nu}_t \rangle + 2\eta \langle w_t - w^*, \bar{\nu}_t - \nu_t \rangle - 2\eta \langle x_t - w^*, \tilde{e}_t - \tilde{e}_{t+1} \rangle \\
&\quad + 32\eta^2 E^2 G^2 \frac{n}{m} \frac{\sqrt{10(1-q)(1-q_0)}}{q_0 q^2} + 16\eta^2 E^2 G^2 \frac{\sqrt{10(1-q_0)}}{q_0 q}
\end{aligned}$$

Plugging $Term - 1$ and $Term - 2$ back into the starting inequality, we get

$$\|x_{t+1} - w^*\|^2 \leq \|x_t - w^*\|^2 - 2\eta \langle w_t - w^*, \bar{\nu}_t \rangle + 2\eta \langle w_t - w^*, \bar{\nu}_t - \nu_t \rangle - 2\eta \langle x_t - w^*, \tilde{e}_t - \tilde{e}_{t+1} \rangle$$

$$+ 32\eta^2 E^2 G^2 \frac{n}{m} \frac{\sqrt{10(1-q)(1-q_0)}}{q_0 q^2} + 16\eta^2 E^2 G^2 \frac{\sqrt{10(1-q_0)}}{q_0 q} + \frac{40\eta^2 E^2 G^2}{q^2}$$

Now, bounding $-2\eta \langle w_t - w^*, \bar{\nu}_t \rangle$, using Lemma 1 and Lemma 2 to bound LHS like we have done in previous proofs, we get

$$2\eta E(f(w_t) - f(w^*))\mathbb{1}\{t \in \mathcal{A}\} + 2\eta E(g(w_t) - g(w^*))\mathbb{1}\{t \in \mathcal{B}\} \leq \|x_t - w^*\|^2 - \|x_{t+1} - w^*\|^2$$

$$+ 2\eta^2 EG^2 + \frac{2}{3}\eta^2 E^3 G^2 + 32\eta^2 E^2 G^2 \frac{n}{m} \frac{\sqrt{10(1-q)(1-q_0)}}{q_0 q^2} + 16\eta^2 E^2 G^2 \frac{\sqrt{10(1-q_0)}}{q_0 q} + \frac{40\eta^2 E^2 G^2}{q^2}$$

$$+ 2\eta \langle w_t - w^*, \bar{\nu}_t - \nu_t \rangle - 2\eta \langle x_t - w^*, \tilde{e}_t - \tilde{e}_{t+1} \rangle .$$

Again add and subtract $\hat{G}(w_t)$ from the 2nd term on LHS and rearrange and we get,

$$2\eta E(f(w_t) - f(w^*))\mathbb{1}\{t \in \mathcal{A}\} + 2\eta E(\hat{G}(w_t) - g(w^*))\mathbb{1}\{t \in \mathcal{B}\} \leq \|x_t - w^*\|^2 - \|x_{t+1} - w^*\|^2$$

$$+ 2\eta^2 EG^2 + \frac{2}{3}\eta^2 E^3 G^2 + 32\eta^2 E^2 G^2 \frac{n}{m} \frac{\sqrt{10(1-q)(1-q_0)}}{q_0 q^2} + 16\eta^2 E^2 G^2 \frac{\sqrt{10(1-q_0)}}{q_0 q} + \frac{40\eta^2 E^2 G^2}{q^2}$$

$$+ 2\eta \langle w_t - w^*, \bar{\nu}_t - \nu_t \rangle - 2\eta \langle x_t - w^*, \tilde{e}_t - \tilde{e}_{t+1} \rangle + 2\eta E(\hat{G}(w_t) - g(w_t))\mathbb{1}\{t \in \mathcal{B}\}.$$

Now sum both sides over $t \in [T]$, we get

$$\sum_{t \in \mathcal{A}} 2\eta E(f(w_t) - f(w^*)) + \sum_{t \in \mathcal{B}} 2\eta E(\hat{G}(w_t) - g(w^*)) \leq D^2 + 2\eta^2 EG^2 T + \frac{2}{3}\eta^2 E^3 G^2 T$$

$$+ 32\eta^2 E^2 G^2 \frac{n}{m} \frac{\sqrt{10(1-q)(1-q_0)}}{q_0 q^2} T + 16\eta^2 E^2 G^2 \frac{\sqrt{10(1-q_0)}}{q_0 q} T + \frac{40\eta^2 E^2 G^2}{q^2} T$$

$$+ \underbrace{\sum_{t=0}^{T-1} 2\eta \langle w_t - w^*, \bar{\nu}_t - \nu_t \rangle - \sum_{t=0}^{T-1} 2\eta \langle x_t - w^*, \tilde{e}_t - \tilde{e}_{t+1} \rangle}_{Term-3} + \sum_{t \in \mathcal{A}} 2\eta E(\hat{G}(w_t) - g(w_t)).$$

Now, we bound $Term - 3$.

$$Term - 3 = -\sum_{t=0}^{T-1} 2\eta \langle x_t - w^*, \tilde{e}_t - \tilde{e}_{t+1} \rangle$$

$$= -2\eta \left[ \sum_{t=0}^{T-1} \langle x_t - w^*, \tilde{e}_t \rangle - \sum_{t=0}^{T-1} \langle x_t - w^*, \tilde{e}_{t+1} \rangle \right]$$

$$= -2\eta \left[ \sum_{t=0}^{T-1} \langle x_t - w^*, \tilde{e}_t \rangle - \sum_{t=1}^{T} \langle x_{t-1} - w^*, \tilde{e}_t \rangle \right]$$

$$= -2\eta \left[ \langle x_0 - w^*, \tilde{e}_0 \rangle + \sum_{t=1}^{T-1} \langle x_t - w^*, \tilde{e}_t \rangle - \sum_{t=1}^{T-1} \langle x_{t-1} - w^*, \tilde{e}_t \rangle - \langle x_{T-1} - w^*, \tilde{e}_T \rangle \right]$$

$$= -2\eta \langle x_0 - w^*, \tilde{e}_0 \rangle + 2\eta \langle x_{T-1} - w^*, \tilde{e}_T \rangle - 2\eta \sum_{t=1}^{T-1} \langle x_t - x_{t-1}, \tilde{e}_t \rangle$$

$$\overset{e_j^0 = 0, Cauchy-Schwarz}{\leq} 2\eta \|x_{T-1} - w^*\| \|\tilde{e}_T\| + 2\eta \sum_{t=1}^{T-1} \|x_t - x_{t-1}\| \|\tilde{e}_t\|$$

$$\overset{As.\ 2, 1-Lip\ \Pi_{\mathcal{X}}}{\leq} \frac{n}{m} \frac{4\eta DEG\sqrt{1-q}}{q} + 2\eta^2 \sum_{t=1}^{T-1} \|v_t\| \frac{n}{m} \frac{2\sqrt{1-q}EG}{q}$$

$$\leq \frac{n}{m} \frac{4\eta DEG\sqrt{1-q}}{q} + \frac{n}{m} \frac{8\eta^2 E^2 G^2 \sqrt{10(1-q)}}{q^2} T$$

Plugging $Term - 3$ back we get

$$\sum_{t \in \mathcal{A}} 2\eta E(f(w_t) - f(w^*)) + \sum_{t \in \mathcal{B}} 2\eta E(\hat{G}(w_t) - g(w^*)) \leq D^2 + 2\eta^2 EG^2 T + \frac{2}{3}\eta^2 E^3 G^2 T$$

$$+ 32\eta^2 E^2 G^2 \frac{n}{m} \frac{\sqrt{10(1-q)(1-q_0)}}{q_0 q^2} T + 16\eta^2 E^2 G^2 \frac{\sqrt{10(1-q_0)}}{q_0 q} T + \frac{40\eta^2 E^2 G^2}{q^2} T$$

$$+ \frac{n}{m} \frac{4\eta DEG\sqrt{1-q}}{q} + \frac{n}{m} \frac{8\eta^2 E^2 G^2 \sqrt{10(1-q)}}{q^2} T$$

$$+ \sum_{t \in \mathcal{A}} 2\eta \left\langle \frac{1}{n}\sum_{j=1}^{n}\sum_{\tau=0}^{E-1}\nabla f_j(w_{j,\tau}^t) - \frac{1}{m}\sum_{j \in \mathcal{S}_t}\sum_{\tau=0}^{E-1}\nabla f_j(w_{j,\tau}^t), w_t - w^* \right\rangle$$

$$+ \sum_{t \in \mathcal{B}} 2\eta \left\langle \frac{1}{n}\sum_{j=1}^{n}\sum_{\tau=0}^{E-1}\nabla g_j(w_{j,\tau}^t) - \frac{1}{m}\sum_{j \in \mathcal{S}_t}\sum_{\tau=0}^{E-1}\nabla g_j(w_{j,\tau}^t), w_t - w^* \right\rangle$$

$$+ \sum_{t \in \mathcal{B}} 2\eta E(\hat{G}(w_t) - g(w_t)).$$

Now, by using Lemmas 4 and 7, we can bound the term above for $\delta \in (0, 1)$ with probability at least $1 - \delta$

$$\sum_{t \in \mathcal{A}} 2\eta E(f(w_t) - f(w^*)) + \sum_{t \in \mathcal{B}} 2\eta E(\hat{G}(w_t) - g(w^*)) \leq D^2 + 2\eta^2 EG^2 T + \frac{2}{3}\eta^2 E^3 G^2 T$$

$$+ 32\eta^2 E^2 G^2 \frac{n}{m} \frac{\sqrt{10(1-q)(1-q_0)}}{q_0 q^2} T + 16\eta^2 E^2 G^2 \frac{\sqrt{10(1-q_0)}}{q_0 q} T + \frac{40\eta^2 E^2 G^2}{q^2} T$$

$$+ \frac{n}{m} \frac{4\eta DEG\sqrt{1-q}}{q} + \frac{n}{m} \frac{8\eta^2 E^2 G^2 \sqrt{10(1-q)}}{q^2} T + 8\eta EGD\sqrt{\frac{2T}{m}\log\left(\frac{3}{\delta}\right)} + 2\eta E\sigma T \sqrt{\frac{2}{m}\log\left(\frac{6T}{\delta}\right)}.$$

Dividing both sides by $2\eta ET$ we get

$$\frac{1}{T}\sum_{t \in \mathcal{A}}(f(w_t) - f(w^*)) + \frac{1}{T}\sum_{t \in \mathcal{B}}(\hat{G}(w_t) - g(w^*)) \leq \frac{D^2}{2\eta ET} + \eta G^2 + \frac{1}{3}\eta E^2 G^2$$

$$+ 16\eta EG^2 \frac{n}{m} \frac{\sqrt{10(1-q)(1-q_0)}}{q_0 q^2} + 8\eta EG^2 \frac{\sqrt{10(1-q_0)}}{q_0 q} + \frac{20\eta EG^2}{q^2}$$

$$+ \frac{n}{m} \frac{2DG\sqrt{1-q}}{qT} + \frac{n}{m} \frac{4\eta EG^2 \sqrt{10(1-q)}}{q^2} + \frac{4GD}{\sqrt{mT}}\sqrt{2\log\left(\frac{3}{\delta}\right)} + \sigma\sqrt{\frac{2}{m}\log\left(\frac{6T}{\delta}\right)}.$$

Now choosing $\eta = \sqrt{\frac{D^2}{2G^2 ET\Gamma}}$, where $\Gamma = 1 + \frac{1}{3}E^2 + 16E\frac{n}{m}\frac{\sqrt{10(1-q)(1-q_0)}}{q_0 q^2} + 8E\frac{\sqrt{10(1-q_0)}}{q_0 q} + \frac{20E}{q^2} + \frac{n}{m}\frac{4E\sqrt{10(1-q)}}{q^2}$, we get

$$\frac{1}{T}\sum_{t \in \mathcal{A}}(f(w_t) - f(w^*)) + \frac{1}{T}\sum_{t \in \mathcal{B}}(\hat{G}(w_t) - g(w^*)) \leq \sqrt{\frac{2D^2 G^2\Gamma}{ET}} + \frac{n}{m}\frac{2DG\sqrt{1-q}}{qT} + \frac{4GD}{\sqrt{mT}}\sqrt{2\log\left(\frac{3}{\delta}\right)}$$

$$+ \sigma\sqrt{\frac{2}{m}\log\left(\frac{6T}{\delta}\right)}.$$

Again, using a similar style of analysis as in previous proofs, if $\mathcal{A} = \phi$, then

$$\epsilon_{bad} < \sqrt{\frac{2D^2 G^2\Gamma}{ET}} + \frac{n}{m}\frac{2DG\sqrt{1-q}}{qT} + \frac{4GD}{\sqrt{mT}}\sqrt{2\log\left(\frac{3}{\delta}\right)} + \sigma\sqrt{\frac{2}{m}\log\left(\frac{6T}{\delta}\right)}.$$

Thus, if we set $\epsilon = \sqrt{\frac{2D^2 G^2\Gamma}{ET}} + \frac{n}{m}\frac{2DG\sqrt{1-q}}{qT} + \frac{4GD}{\sqrt{mT}}\sqrt{2\log\left(\frac{3}{\delta}\right)} + \sigma\sqrt{\frac{2}{m}\log\left(\frac{6T}{\delta}\right)}, \mathcal{A} \neq \phi$. Now, let's consider two different cases here as well. Either $\sum_{t \in \mathcal{A}} f(w_t) - f(w^*) \leq 0$ which implies by the convexity of $f$ for $\bar{w} = \frac{1}{|\mathcal{A}|}\sum_{t \in \mathcal{A}} w_t$, we have

$$f(\bar{w}) - f(w^*) \leq 0 < \epsilon. \tag{28}$$

Additionally, we know using the convexity of g that

$$g(\bar{w}) \leq \frac{1}{|\mathcal{A}|} \sum_{t \in \mathcal{A}} g(w_t)$$

$$\leq \frac{1}{|\mathcal{A}|} \sum_{t \in \mathcal{A}} (g(w_t) - \hat{G}(w_t)) + \frac{1}{|\mathcal{A}|} \sum_{t \in \mathcal{A}} \hat{G}(w_t)$$

$$\overset{Lem.\ 7}{\leq} \epsilon + \sigma \sqrt{\frac{2}{m} \log \left( \frac{6T}{\delta} \right)}, \text{ with probability at least } 1 - \delta.$$

Otherwise, if $\sum_{t \in \mathcal{A}} f(w_t) - f(w^*) > 0$, then

$$\sqrt{\frac{2D^2 G^2 \Gamma}{ET}} + \frac{n}{m} \frac{2DG\sqrt{1-q}}{qT} + \frac{4GD}{\sqrt{mT}} \sqrt{2 \log \left( \frac{3}{\delta} \right)} + \sigma \sqrt{\frac{2}{m} \log \left( \frac{6T}{\delta} \right)}$$

$$\geq \frac{1}{T} \sum_{t \in \mathcal{A}} f(w_t) - f(w^*) + \frac{1}{T} \sum_{t \in \mathcal{B}} \hat{G}(w_t) - g(w^*)$$

$$> \frac{1}{T} \sum_{t \in \mathcal{A}} f(w_t) - f(w^*) + \frac{1}{T} \sum_{t \in \mathcal{B}} \epsilon \qquad (29)$$

$$= \frac{|\mathcal{A}|}{T} \frac{1}{|\mathcal{A}|} \sum_{t \in \mathcal{A}} f(w_t) - f(w^*) + (1 - \frac{|\mathcal{A}|}{T})\epsilon$$

$$\geq \frac{|\mathcal{A}|}{T} \left( f(\bar{w}) - f(w^*) \right) + (1 - \frac{|\mathcal{A}|}{T})\epsilon.$$

This implies $f(\bar{w}) - f(w^*) < \epsilon$ and further using the same argument as above for $g$, we have for probability at least $1 - \delta$, $g(\bar{w}) \leq \epsilon + \sigma \sqrt{\frac{2}{m} \log \left( \frac{6T}{\delta} \right)}$. $\qquad \square$

### C.4.3 SOFT SWITCHING — FULL PARTICIPATION

**Theorem 8.** *Consider the problem in equation 1, where $\mathcal{X} = \mathbb{R}^d$ and Algorithm 1, under Assumption 1 and 3. Define $D := \|w_0 - w^*\|$ and*

$$\mathcal{A} = \{t \in [T] \mid g(w_t) < \epsilon\}, \qquad \bar{w} = \sum_{t \in \mathcal{A}} \alpha_t w_t,$$

*where*

$$\alpha_t = \frac{1 - \sigma_\beta(g(w_t) - \epsilon)}{\sum_{t \in \mathcal{A}} [1 - \sigma_\beta(g(w_t) - \epsilon)]}. \qquad (30)$$

*Then, if*

$$\epsilon = \sqrt{\frac{2D^2 G^2 \Gamma}{ET}}, \eta = \sqrt{\frac{D^2}{2G^2 ET\Gamma}}, \&\beta = \frac{2}{\epsilon} \text{ where } \Gamma = 2E^2 + 2E \frac{\sqrt{1-q}}{q} + 4E \frac{\sqrt{10(1-q_0)}}{q_0 q}$$

*it holds that $\mathcal{A}$ is nonempty, $\bar{w}$ is well-defined, and $\bar{w}$ is an $\epsilon$-solution for $P$.*

*Proof.* Starting with the update rule presented in eq. (21) along with the soft-switching update mechanism, we analyze the squared distance to the optimal point $w^*$ as follows,

$$\|\hat{w}_{t+1} - w^*\|^2 = \|\hat{w}_t - \eta \nu_t - w^*\|^2$$

$$= \|\hat{w}_t - w^*\|^2 + \underbrace{\eta^2 \|\nu_t\|^2}_{Term-A} \underbrace{-2\eta \langle \hat{w}_t - w^*, \nu_t \rangle}_{Term-B} \qquad (31)$$

Firstly, we start with upper bounding $Term - A$ where we use the fact that $\sigma_\beta^t = \sigma_\beta(g(w_t) - \epsilon) \in [0, 1]$.

$$Term - A = \eta^2 \|\nu_t\|^2 = \eta^2 \left\| \frac{1}{n} \sum_{j=1}^{n} \sum_{\tau=0}^{E-1} \nu_{j,\tau}^t \right\|^2 \overset{Jensen's}{\leq} \eta^2 \frac{1}{n} \sum_{j=1}^{n} E \sum_{\tau=0}^{E-1} \|\nu_{j,\tau}^t\|^2 \overset{G-Lip}{\leq} \eta^2 E^2 G^2$$

Now, we start to upper bound $Term - B$.

$$Term - B = -2\eta \langle \hat{w}_t - w^*, \nu_t \rangle$$

$$= -2\eta \langle \hat{w}_t - x_t + x_t - w^*, \nu_t \rangle$$

$$= -2\eta \langle \hat{w}_t - x_t, \nu_t \rangle - 2\eta \langle x_t - w^*, \nu_t \rangle$$

$$= 2\eta^2 \langle e_t, \nu_t \rangle + 2\eta \langle w_t - x_t, \nu_t \rangle - 2\eta \langle w_t - w^*, \nu_t \rangle$$

$$= \frac{2\eta^2}{n} \sum_{j=1}^{n} \sum_{\tau=0}^{E-1} \langle e_t, \nu_{j,\tau}^t \rangle + \frac{2\eta}{n} \sum_{j=1}^{n} \sum_{\tau=0}^{E-1} \langle -\hat{e}_t, \nu_{j,\tau}^t \rangle - \frac{2\eta}{n} \sum_{j=1}^{n} \sum_{\tau=0}^{E-1} \langle w_t - w^*, \nu_{j,\tau}^t \rangle$$

$$\overset{Cauchy-Schwarz}{\leq} \frac{2\eta^2}{n} \sum_{j=1}^{n} \sum_{\tau=0}^{E-1} \|e_t\| \|\nu_{j,\tau}^t\| + \frac{2\eta}{n} \sum_{j=1}^{n} \sum_{\tau=0}^{E-1} \|\hat{e}_t\| \|\nu_{j,\tau}^t\|$$

$$- \frac{2\eta}{n} \sum_{j=1}^{n} \sum_{\tau=0}^{E-1} \langle w_t - w^*, \nu_{j,\tau}^t \rangle$$

$$\overset{G-Lip}{\leq} \underbrace{\frac{2\eta^2 G}{n} \sum_{j=1}^{n} \sum_{\tau=0}^{E-1} \|e_t\|}_{Term-B_1} + \underbrace{\frac{2\eta G}{n} \sum_{j=1}^{n} \sum_{\tau=0}^{E-1} \|\hat{e}_t\|}_{Term-B_2} \underbrace{- \frac{2\eta}{n} \sum_{j=1}^{n} \sum_{\tau=0}^{E-1} \langle w_t - w^*, \nu_{j,\tau}^t \rangle}_{Term-B_3}$$

We will use the results for upper bounding $Term - B_1$ and $Term - B_2$ from the proof of Theorem 6, and we will borrow the result for upper bounding $Term - B_3$ from the proof of Theorem 5. Now, we take the expectation on both sides of eq. (31) with respect to the compression operators and plug in the value of $Term - B_1$, $Term - B_2$, and $Term - B_3$, we get

$$\|\hat{w}_{t+1} - w^*\|^2 \leq \|\hat{w}_t - w^*\|^2 + \eta^2 E^2 G^2 + 4\eta^2 E^2 G^2 \frac{\sqrt{(1-q)}}{q} + 8\eta^2 E^2 G^2 \frac{\sqrt{10(1-q_0)}}{q_0 q}$$

$$\frac{2\eta}{3\alpha} \eta^3 E^3 G^2 + 2\eta\alpha E G^2 + 2\eta E \sigma_\beta^t (g(w^*) - g(w_t)) + 2\eta E(1 - \sigma_\beta^t)(f(w^*) - f(w_t))$$

$$\overset{\alpha=\eta}{\leq} \|\hat{w}_t - w^*\|^2 + \eta^2 E^2 G^2 + 4\eta^2 E^2 G^2 \frac{\sqrt{(1-q)}}{q} + 8\eta^2 E^2 G^2 \frac{\sqrt{10(1-q_0)}}{q_0 q}$$

$$\frac{2\eta}{3} \eta^2 E^3 G^2 + 2\eta^2 E G^2 + 2\eta E \sigma_\beta^t (g(w^*) - g(w_t)) + 2\eta E(1 - \sigma_\beta^t)(f(w^*) - f(w_t))$$
$$(32)$$

Let $\mathcal{A} = \{t \in [T] | g(w_t) < \epsilon\}$ and $\mathcal{B} = [T] \setminus \mathcal{A} = \{t \in [T] | g(w_t) \geq \epsilon\}$. Note that for all $t \in \mathcal{B}$ it holds that $\sigma_\beta(g(w_t) - \epsilon) = 1$ and $g(w_t) - g(w^*) \geq \epsilon$. Further, for all $t \in \mathcal{A}$ if $\sigma_\beta(g(w_t) - \epsilon) \geq 0$ it holds that $g(w_t) - g(w^*) \geq g(w_t) \geq \epsilon - 1/\beta$. With these observations, using convexity of $f$ and $g$ and decomposing the sum over $t$ according to the definitions of $\mathcal{A}$ and $\mathcal{B}$ and division by $T$ yields,

$$\frac{D^2}{2\eta E} + \frac{1}{2}\eta E G^2 T + \eta G^2 T + \frac{1}{3}\eta E^2 G^2 T + 2\eta E G^2 \frac{\sqrt{(1-q)}}{q} + 4\eta E G^2 \frac{\sqrt{10(1-q_0)}}{q_0 q} \geq$$

$$\sum_{t \in \mathcal{A}} \sigma_\beta^t (g(w_t) - g(w^*)) + \sum_{t \in \mathcal{A}} (1 - \sigma_\beta^t)(f(w_t) - f(w^*)) + \sum_{t \in \mathcal{B}} (g(w_t) - g(w^*)).$$

Now choosing $\eta = \sqrt{\frac{D^2}{2G^2 E T \Gamma}}$, where $\Gamma = \frac{1}{2}E + 1 + \frac{1}{3}E^2 + 2E \frac{\sqrt{1-q}}{q} + 4E \frac{\sqrt{10(1-q_0)}}{q_0 q}$, we get

$$\sqrt{\frac{2D^2 G^2 T \Gamma}{E}} \geq \sum_{t \in \mathcal{A}} \sigma_\beta^t (g(w_t) - g(w^*)) + \sum_{t \in \mathcal{A}} (1 - \sigma_\beta^t)(f(w_t) - f(w^*))$$

$$+ \sum_{t \in \mathcal{B}} (g(w_t) - g(w^*)) \tag{33}$$

$$\geq \sum_{t \in \mathcal{A}} (1 - \sigma_\beta^t)(f(w_t) - f(w^*)) + \epsilon |\mathcal{B}| + \left(\epsilon - \frac{1}{\beta}\right) \sum_{t \in \mathcal{A}} \sigma_\beta^t.$$

Similar to the previous proofs, we first need to find the smallest $\epsilon$ to ensure $\mathcal{A}$ is non-empty. So, to find a lower bound on $\epsilon$, assume $\mathcal{A}$ is empty in eq. (33) and observe that as long as condition $\sqrt{\frac{2D^2 G^2 \Gamma}{ET}} < \epsilon$ is met, $\mathcal{A}$ is non-empty. We choose to set $\epsilon = 2\sqrt{\frac{2D^2 G^2 \Gamma}{ET}}$.

Now, like before, we consider two cases based on the sign of $\sum_{t \in \mathcal{A}} \left(1 - \sigma_\beta^t\right)\left(f(w_t) - f(w^*)\right)$. As before, when the sum is non-positive, we are done by the definition of $\mathcal{A}$, which implies $0 < 1 - \sigma_\beta(g(w_t) - \epsilon) \le 1$ and the convexity of $f$ and $g$.

Assuming the sum is positive, dividing eq. (33) by $\sum_{t \in \mathcal{A}} \left(1 - \sigma_\beta^t\right)$ (which by the definition of $\mathcal{A}$ is strictly positive), using convexity of $f$, and the definition of $\bar{w}$, we have

$$f(\bar{w}) - f(w^*) \le \frac{0.5\epsilon T - \epsilon|\mathcal{B}| - (\epsilon - \frac{1}{\beta}) \sum_{t \in \mathcal{A}} \sigma_\beta^t}{|\mathcal{A}| - \sum_{t \in \mathcal{A}} \sigma_\beta(g(w_t) - \epsilon)}$$

$$= \epsilon + \frac{-0.5\epsilon T + \beta^{-1} \sum_{t \in \mathcal{A}} \sigma_\beta^t}{|\mathcal{A}| - \sum_{t \in \mathcal{A}} \sigma_\beta^t},$$

where we used $|\mathcal{B}| = T - |\mathcal{A}|$.

Let us now find a lower bound on $\beta$ to ensure the second term in the bound is non-positive. Note this is done for simplicity, and as long as the second term is $\mathcal{O}(\epsilon)$, an $\epsilon$-solution can be found. Immediate calculations show the second term in the bound is non-positive when

$$\beta \ge \frac{2 \sum_{t \in \mathcal{A}} \sigma_\beta^t}{\epsilon T}.$$

Since $\sum_{t \in \mathcal{A}} \sigma_\beta^t < T$, a sufficient (and highly conservative) condition for all $T \ge 1$ is to set $\beta = 2/\epsilon$. Thus, we proved the sub-optimality gap result. The feasibility result is immediate given the definition of $\mathcal{A}$ and the convexity of $g$. $\qquad \square$

# D  STOCHASTIC FEDSGM

**Theorem 9.** *Consider the optimization problem in equation 1 and Algorithm 1 under Assumptions 1, 2, and 5. Additionally, assume that in each global round, $m$ distinct clients are sampled uniformly at random. Define $\tilde{G}(w_t) := \frac{1}{m} \sum_{j \in \mathcal{S}_t} g_j(w_t, \xi_t^j)$,*

$$\mathcal{A} := \left\{ t \in [T] \,\Big|\, \tilde{G}(w_t) \le \epsilon \right\}, \qquad \bar{w} := \frac{1}{|\mathcal{A}|} \sum_{t \in \mathcal{A}} w_t.$$

*Suppose the step size $\eta$ and constraint threshold $\epsilon$ are set as*

$$\eta = \sqrt{\frac{D^2}{2G_{eff}ET}} \qquad and$$

$$\epsilon = \sqrt{\frac{2D^2 G_{eff}}{ET}} + 4GD\sqrt{\frac{2}{mT} \log\left(\frac{7}{\delta}\right)} + 2D\sigma_\xi \sqrt{\frac{2}{mET} \log\left(\frac{7}{\delta}\right)} + 2\sqrt{\frac{2}{m}\left(\sigma^2 + \frac{\sigma_b^2}{N_b}\right) \log\left(\frac{14T}{\delta}\right)},$$

*where $G_{eff} = 3E^2G^2 + E^2\sigma_\xi^2 \left(1 + \log\left(\frac{7nTE}{\delta}\right)\right) + \frac{4}{m}(EG^2 + \sigma_\xi^2) \log\left(\frac{28T}{\delta}\right)$ and $\delta \in (0, 1)$ is a confidence parameter. Then, with probability at least $1 - \delta$, the following hold:*

1. *The set $\mathcal{A}$ is non-empty, so $\bar{w}$ is well-defined.*

2. *The average iterate $\bar{w}$ satisfies*

$$f(\bar{w}) - f(w^*) \le \epsilon, \qquad g(\bar{w}) \le \epsilon.$$

*Proof.* Using Algorithm 1, the update rule for the global model under partial participation is

$$w_{t+1} = \Pi_{\mathcal{X}} \left( w_t - \eta \cdot \frac{1}{m} \sum_{j \in \mathcal{S}_t} \sum_{\tau=0}^{E-1} \nu_{j,\tau}^t \right), \quad \text{where } \nu_{j,\tau}^t = \nabla f_j(w_{j,\tau}^t, \xi_{j,\tau}^t) \text{ or } g_j(w_{j,\tau}^t, \xi_{j,\tau}^t).$$

$$\text{(34)}$$

Now, we analyze the squared distance to the optimal point $w^*$ as follows,

$$\|w_{t+1} - w^*\|^2 \overset{1-Lip \ \Pi_{\mathcal{X}}}{\leq} \left\| w_t - \eta \cdot \frac{1}{m} \sum_{j \in \mathcal{S}_t} \sum_{\tau=0}^{E-1} \nu_{j,\tau}^t - w^* \right\|^2$$

$$= \|w_t - w^*\|^2 + \eta^2 \left\| \frac{1}{m} \sum_{j \in \mathcal{S}_t} \sum_{\tau=0}^{E-1} \nu_{j,\tau}^t \right\|^2 - 2\eta \left\langle w_t - w^*, \frac{1}{m} \sum_{j \in \mathcal{S}_t} \sum_{\tau=0}^{E-1} \nu_{j,\tau}^t \right\rangle$$

$$\overset{Young's}{\leq} \|w_t - w^*\|^2 + 2\eta^2 \left\| \frac{1}{n} \sum_{j \in [m]} \sum_{\tau=0}^{E-1} \bar{\nu}_{j,\tau}^t \right\|^2 + 2\eta^2 \left\| \frac{1}{n} \sum_{j \in [n]} \sum_{\tau=0}^{E-1} \bar{\nu}_{j,\tau}^t - \frac{1}{m} \sum_{j \in \mathcal{S}_t} \sum_{\tau=0}^{E-1} \nu_{j,\tau}^t \right\|^2$$

$$+ 2\eta \left\langle w_t - w^*, \frac{1}{n} \sum_{j=1}^n \sum_{\tau=0}^{E-1} \bar{\nu}_{j,\tau}^t - \frac{1}{m} \sum_{j \in \mathcal{S}_t} \sum_{\tau=0}^{E-1} \nu_{j,\tau}^t \right\rangle - 2\eta \left\langle w_t - w^*, \frac{1}{n} \sum_{j=1}^n \sum_{\tau=0}^{E-1} \bar{\nu}_{j,\tau}^t \right\rangle$$

$$\overset{G-Lip}{\leq} \|w_t - w^*\|^2 + 2\eta^2 E^2 G^2 + 2\eta^2 \left\| \frac{1}{n} \sum_{j \in [n]} \sum_{\tau=0}^{E-1} \bar{\nu}_{j,\tau}^t - \frac{1}{m} \sum_{j \in \mathcal{S}_t} \sum_{\tau=0}^{E-1} \nu_{j,\tau}^t \right\|^2$$

$$+ 2\eta \left\langle w_t - w^*, \frac{1}{n} \sum_{j=1}^n \sum_{\tau=0}^{E-1} \bar{\nu}_{j,\tau}^t - \frac{1}{m} \sum_{j \in \mathcal{S}_t} \sum_{\tau=0}^{E-1} \nu_{j,\tau}^t \right\rangle - 2\eta \left\langle w_t - w^*, \frac{1}{n} \sum_{j=1}^n \sum_{\tau=0}^{E-1} \bar{\nu}_{j,\tau}^t \right\rangle.$$

Now we can use Lemma 1 for $\alpha = \eta$ to bound LHS like we have done in previous proofs, we get

$$2\eta E(f(w_t) - f(w^*))\mathbb{1}_{t \in \mathcal{A}} + 2\eta E(g(w_t) - g(w^*))\mathbb{1}_{t \in \mathcal{B}} \leq \|w_t - w^*\|^2 - \|w_{t+1} - w^*\|^2 + 2\eta^2 E^2 G^2$$

$$+ 2\eta^2 E G^2 + \frac{2}{n} \sum_{j=1}^n \sum_{\tau=0}^{E-1} \|w_t - w_{j,\tau}^t\|^2 + 2\eta^2 \left\| \frac{1}{n} \sum_{j \in [n]} \sum_{\tau=0}^{E-1} \bar{\nu}_{j,\tau}^t - \frac{1}{m} \sum_{j \in \mathcal{S}_t} \sum_{\tau=0}^{E-1} \nu_{j,\tau}^t \right\|^2$$

$$+ 2\eta \left\langle \frac{1}{n} \sum_{j=1}^n \sum_{\tau=0}^{E-1} \nabla f_j(w_{j,\tau}^t) - \frac{1}{m} \sum_{j \in \mathcal{S}_t} \sum_{\tau=0}^{E-1} \nabla f_j(w_{j,\tau}^t, \xi_{j,\tau}^t), w_t - w^* \right\rangle \mathbb{1}_{t \in \mathcal{A}}$$

$$+ 2\eta \left\langle \frac{1}{n} \sum_{j=1}^n \sum_{\tau=0}^{E-1} \nabla g_j(w_{j,\tau}^t) - \frac{1}{m} \sum_{j \in \mathcal{S}_t} \sum_{\tau=0}^{E-1} \nabla g_j(w_{j,\tau}^t, \xi_{j,\tau}^t)), w_t - w^* \right\rangle \mathbb{1}_{t \in \mathcal{B}}.$$

Again add and subtract $\tilde{G}(w_t)$ from the 2nd term on LHS and rearrange and we get,

$$2\eta E(f(w_t) - f(w^*))\mathbb{1}_{t \in \mathcal{A}} + 2\eta E(\tilde{G}(w_t) - g(w^*))\mathbb{1}_{t \in \mathcal{B}} \leq \|w_t - w^*\|^2 - \|w_{t+1} - w^*\|^2$$

$$+ 2\eta^2 E^2 G^2 + 2\eta^2 E G^2 + \frac{2}{n} \sum_{j=1}^n \sum_{\tau=0}^{E-1} \|w_t - w_{j,\tau}^t\|^2 + 2\eta^2 \left\| \frac{1}{n} \sum_{j \in [n]} \sum_{\tau=0}^{E-1} \bar{\nu}_{j,\tau}^t - \frac{1}{m} \sum_{j \in \mathcal{S}_t} \sum_{\tau=0}^{E-1} \nu_{j,\tau}^t \right\|^2$$

$$+ 2\eta \left\langle \frac{1}{n} \sum_{j=1}^n \sum_{\tau=0}^{E-1} \nabla f_j(w_{j,\tau}^t) - \frac{1}{m} \sum_{j \in \mathcal{S}_t} \sum_{\tau=0}^{E-1} \nabla f_j(w_{j,\tau}^t, \xi_{j,\tau}^t), w_t - w^* \right\rangle \mathbb{1}_{t \in \mathcal{A}}$$

$$+ 2\eta \left\langle \frac{1}{n} \sum_{j=1}^n \sum_{\tau=0}^{E-1} \nabla g_j(w_{j,\tau}^t) - \frac{1}{m} \sum_{j \in \mathcal{S}_t} \sum_{\tau=0}^{E-1} \nabla g_j(w_{j,\tau}^t, \xi_{j,\tau}^t)), w_t - w^* \right\rangle \mathbb{1}_{t \in \mathcal{B}}$$

$$+ 2\eta E(\tilde{G}(w_t) - g(w_t))\mathbb{1}_{t \in \mathcal{B}}.$$

Now sum both sides over $t \in [T]$, we get

$$\sum_{t \in \mathcal{A}} 2\eta E(f(w_t) - f(w^*)) + \sum_{t \in \mathcal{B}} 2\eta E(\tilde{G}(w_t) - g(w^*)) \leq D^2 + 2\eta^2 E^2 G^2 T + 2\eta^2 E G^2 T$$

$$+ \sum_{t=0}^{T-1} \frac{2}{n} \sum_{j=1}^{n} \sum_{\tau=0}^{E-1} \|w_t - w_{j,\tau}^t\|^2 + \sum_{t=0}^{T-1} 2\eta^2 \left\| \frac{1}{n} \sum_{j \in [n]} \sum_{\tau=0}^{E-1} \bar{\nu}_{j,\tau}^t - \frac{1}{m} \sum_{j \in \mathcal{S}_t} \sum_{\tau=0}^{E-1} \nu_{j,\tau}^t \right\|^2$$

$$+ \sum_{t \in \mathcal{A}} 2\eta \left\langle \frac{1}{n} \sum_{j=1}^{n} \sum_{\tau=0}^{E-1} \nabla f_j(w_{j,\tau}^t) - \frac{1}{m} \sum_{j \in \mathcal{S}_t} \sum_{\tau=0}^{E-1} \nabla f_j(w_{j,\tau}^t, \xi_{j,\tau}^t), w_t - w^* \right\rangle$$

$$+ \sum_{t \in \mathcal{B}} 2\eta \left\langle \frac{1}{n} \sum_{j=1}^{n} \sum_{\tau=0}^{E-1} \nabla g_j(w_{j,\tau}^t) - \frac{1}{m} \sum_{j \in \mathcal{S}_t} \sum_{\tau=0}^{E-1} \nabla g_j(w_{j,\tau}^t, \xi_{j,\tau}^t)), w_t - w^* \right\rangle$$

$$+ \sum_{t \in \mathcal{B}} 2\eta E(\tilde{G}(w_t) - g(w_t)).$$

Now, by using Lemmas 3, 4, 5, 6, and 8, we can bound the term above for $\delta \in (0, 1)$ with probability at least $1 - \delta$

$$\sum_{t \in \mathcal{A}} 2\eta E(f(w_t) - f(w^*)) + \sum_{t \in \mathcal{B}} 2\eta E(\tilde{G}(w_t) - g(w^*)) \leq D^2 + 2\eta^2 E^2 G^2 T + 2\eta^2 E G^2 T$$

$$+ 2\eta^2 E^3 T \left[ \frac{2}{3} G^2 + \sigma_\xi^2 \left( 1 + \log\left( \frac{7nTE}{\delta} \right) \right) \right] + \frac{8\eta^2 ET}{m} \left( E G^2 + \sigma_\xi^2 \right) \log\left( \frac{28T}{\delta} \right)$$

$$+ 8\eta E G D \sqrt{\frac{2T}{m} \log\left( \frac{7}{\delta} \right)} + 4\eta D \sigma_\xi \sqrt{\frac{2ET}{m} \log\left( \frac{7}{\delta} \right)} + 2\eta ET \sqrt{\frac{2}{m} \left( \sigma^2 + \frac{\sigma_b^2}{N_b} \right) \log\left( \frac{14T}{\delta} \right)}.$$

Dividing both sides by $2\eta ET$, we get

$$\frac{1}{T} \sum_{t \in \mathcal{A}} (f(w_t) - f(w^*)) + \frac{1}{T} \sum_{t \in \mathcal{B}} (\tilde{G}(w_t) - g(w^*)) \leq \frac{D^2}{2\eta ET} + \eta E G^2 + \eta G^2 + \frac{2}{3} \eta E^2 G^2$$

$$+ \eta E^2 \sigma_\xi^2 \left( 1 + \log\left( \frac{7nTE}{\delta} \right) \right) + \frac{4\eta}{m} \left( E G^2 + \sigma_\xi^2 \right) \log\left( \frac{28T}{\delta} \right)$$

$$+ 4GD \sqrt{\frac{2}{mT} \log\left( \frac{7}{\delta} \right)} + 2D\sigma_\xi \sqrt{\frac{2}{mET} \log\left( \frac{7}{\delta} \right)} + \sqrt{\frac{2}{m} \left( \sigma^2 + \frac{\sigma_b^2}{N_b} \right) \log\left( \frac{14T}{\delta} \right)}$$

$$\leq \frac{D^2}{2\eta ET} + 3\eta E^2 G^2 + \eta E^2 \sigma_\xi^2 \left( 1 + \log\left( \frac{7nTE}{\delta} \right) \right) + \frac{4\eta}{m} \left( E G^2 + \sigma_\xi^2 \right) \log\left( \frac{28T}{\delta} \right)$$

$$+ 4GD \sqrt{\frac{2}{mT} \log\left( \frac{7}{\delta} \right)} + 2D\sigma_\xi \sqrt{\frac{2}{mET} \log\left( \frac{7}{\delta} \right)} + \sqrt{\frac{2}{m} \left( \sigma^2 + \frac{\sigma_b^2}{N_b} \right) \log\left( \frac{14T}{\delta} \right)}.$$

Now choosing $\eta = \sqrt{\frac{D^2}{2G_{eff}ET}}$, where $G_{eff} = 3E^2 G^2 + E^2 \sigma_\xi^2 \left( 1 + \log\left( \frac{7nTE}{\delta} \right) \right) + \frac{4}{m} (E G^2 + \sigma_\xi^2) \log\left( \frac{28T}{\delta} \right)$, we get

$$\frac{1}{T} \sum_{t \in \mathcal{A}} (f(w_t) - f(w^*)) + \frac{1}{T} \sum_{t \in \mathcal{B}} (\tilde{G}(w_t) - g(w^*)) \leq \sqrt{\frac{2D^2 G_{eff}}{ET}} + 4GD \sqrt{\frac{2}{mT} \log\left( \frac{7}{\delta} \right)}$$

$$+ 2D\sigma_\xi \sqrt{\frac{2}{mET} \log\left( \frac{7}{\delta} \right)} + \sqrt{\frac{2}{m} \left( \sigma^2 + \frac{\sigma_b^2}{N_b} \right) \log\left( \frac{14T}{\delta} \right)}.$$

Again, using a similar style of analysis as in previous proofs, if $\mathcal{A} = \phi$, then

$$\epsilon_{bad} < \sqrt{\frac{2D^2 G_{eff}}{ET}} + 4GD \sqrt{\frac{2}{mT} \log\left( \frac{7}{\delta} \right)} + 2D\sigma_\xi \sqrt{\frac{2}{mET} \log\left( \frac{7}{\delta} \right)} + \sqrt{\frac{2}{m} \left( \sigma^2 + \frac{\sigma_b^2}{N_b} \right) \log\left( \frac{14T}{\delta} \right)}.$$

Thus, if we set $\epsilon = \sqrt{\frac{2D^2 G_{eff}}{ET}} + 4GD\sqrt{\frac{2}{mT}\log\left(\frac{7}{\delta}\right)} + 2D\sigma_\xi\sqrt{\frac{2}{mET}\log\left(\frac{7}{\delta}\right)} + \sqrt{\frac{2}{m}\left(\sigma^2 + \frac{\sigma_b^2}{N_b}\right)\log\left(\frac{14T}{\delta}\right)}$, $\mathcal{A} \neq \phi$. Now, let's consider two different cases here as well. Either $\sum_{t\in\mathcal{A}} f(w_t) - f(w^*) \leq 0$ which implies by the convexity of $f$ for $\bar{w} = \frac{1}{|\mathcal{A}|}\sum_{t\in\mathcal{A}} w_t$, we have

$$f(\bar{w}) - f(w^*) \leq 0 < \epsilon. \tag{35}$$

Additionally, we know via convexity of g that

$$g(\bar{w}) \leq \frac{1}{|\mathcal{A}|}\sum_{t\in\mathcal{A}} g(w_t)$$

$$\leq \frac{1}{|\mathcal{A}|}\sum_{t\in\mathcal{A}}(g(w_t) - \tilde{G}(w_t)) + \frac{1}{|\mathcal{A}|}\sum_{t\in\mathcal{A}} \tilde{G}(w_t)$$

$$\overset{Lem.\ 8}{\leq} \epsilon + \sqrt{\frac{2}{m}\left(\sigma^2 + \frac{\sigma_b^2}{N_b}\right)\log\left(\frac{14T}{\delta}\right)}, \text{ with probability at least } 1 - \delta.$$

Otherwise, if $\sum_{t\in\mathcal{A}} f(w_t) - f(w^*) > 0$, then

$$\sqrt{\frac{2D^2 G_{eff}\Gamma}{ET}} + 4GD\sqrt{\frac{2}{mT}\log\left(\frac{7}{\delta}\right)} + 2D\sigma_\xi\sqrt{\frac{2}{mET}\log\left(\frac{5}{\delta}\right)} + \sqrt{\frac{2}{m}\left(\sigma^2 + \frac{\sigma_b^2}{N_b}\right)\log\left(\frac{10T}{\delta}\right)}$$

$$\geq \frac{1}{T}\sum_{t\in\mathcal{A}} f(w_t) - f(w^*) + \frac{1}{T}\sum_{t\in\mathcal{B}} \hat{G}(w_t) - g(w^*)$$

$$> \frac{1}{T}\sum_{t\in\mathcal{A}} f(w_t) - f(w^*) + \frac{1}{T}\sum_{t\in\mathcal{B}} \epsilon$$

$$= \frac{|\mathcal{A}|}{T}\frac{1}{|\mathcal{A}|}\sum_{t\in\mathcal{A}} f(w_t) - f(w^*) + (1 - \frac{|\mathcal{A}|}{T})\epsilon$$

$$\geq \frac{|\mathcal{A}|}{T}\left(f(\bar{w}) - f(w^*)\right) + (1 - \frac{|\mathcal{A}|}{T})\epsilon. \tag{36}$$

This implies $f(\bar{w}) - f(w^*) < \epsilon$ and further using the same argument as above for $g$, we have for probability at least $1 - \delta$, $g(\bar{w}) \leq \epsilon + \sqrt{\frac{2}{m}\left(\sigma^2 + \frac{\sigma_b^2}{N_b}\right)\log\left(\frac{14T}{\delta}\right)}$. $\qquad\square$

## E  WEAKLY CONVEX EXTENSION OF FEDSGM

We extend our proof presented in the main paper for the partial participation case to the case where $f$ is $\rho-$weakly convex. This analysis is complementary and serves as an addition to our work. This analysis is adapted from the work of Huang & Lin (2023) and extends it to the federated setting. A noteworthy point is that this is the first work that analyzes the rate of convergence for weakly convex objective with functional constraints in the federated setting. Another quick note is that (with a little abuse of notation) we will still use the $\nabla$ sign to denote subgradient.

Now, we introduce the standard assumption used in this setting. Firstly, we will borrow, convexity and Lipschitzness of $f_j$ and $g_j$ from Assumption 1, except that $f_j$ or $f$ is not convex anymore, and we will utilize Assumption 2. Here are the assumptions written formally,

**Assumption 6.** *Each local function $f_j$ is $\rho$-weakly convex on $\mathcal{X}$; hence $f$ is $\rho$-weakly convex.*

**Assumption 7.** *Each constraint function $g_j$ is convex on $\mathcal{X}$; hence $g$ is convex.*

**Assumption 8.** *Both $f_j$ and $g_j$ are $G-$Lipschitz continuous on $\mathcal{X}$.*

**Assumption 9.** *There exists $w_{\text{feas}} \in \text{relint}(\mathcal{X})$ such that $g(w_{\text{feas}}) < 0$.*

Now, we define the federated proximal constrained problem as the analysis of weakly convex function relies on a proximal regularization to induce strong convexity. So, given a point $w \in \mathcal{X}$ and a

parameter $\hat{\rho} > 2\rho$, define the federated constrained proximal subproblem:

$$\hat{w}(w) \in \arg\min_{y \in \mathcal{X}} \left\{ \frac{1}{n} \sum_{j=1}^{n} f_j(y) + \frac{\hat{\rho}}{2} \|y - w\|^2 \text{ s.t. } \frac{1}{n} \sum_{j=1}^{n} g_j(y) \leq 0 \right\}. \tag{37}$$

We observe that as $f$ is $\rho-$ weakly convex, the formulation in equation 37, is $(\hat{\rho} - \rho)-$strongly convex, and therefore has a unique minimizer denoted by $\hat{w}(w)$. For a more formal treatment of weakly convex functions, please refer to Huang & Lin (2023).

**KKT Conditions for the Federated Proximal Problem.** Let $\hat{y} = \hat{w}(w)$ denote the unique minimizer and let $\hat{\lambda} \geq 0$ be the optimal Lagrange multiplier. Choose sub-gradients

$$\nabla f_j(\hat{y}) \in \partial f_j(\hat{y}), \qquad \nabla g_j(\hat{y}) \in \partial g_j(\hat{y}),$$

and let $N_{\mathcal{X}}(\hat{y})$ denote the normal cone. The KKT conditions for equation 37 is:

$$\text{(stationarity)} \quad 0 \in \frac{1}{n} \sum_{j=1}^{n} \nabla f_j(\hat{y}) + \hat{\rho}(\hat{y} - w) + \hat{\lambda} \frac{1}{n} \sum_{j=1}^{n} \nabla g_j(\hat{y}) + N_{\mathcal{X}}(\hat{y}), \quad \text{(38)}$$

$$\text{(primal feasibility)} \quad g(\hat{y}) = \frac{1}{n} \sum_{j=1}^{n} g_j(\hat{y}) \leq 0, \tag{39}$$

$$\text{(dual feasibility)} \quad \hat{\lambda} \geq 0, \tag{40}$$

$$\text{(complementary slackness)} \quad \hat{\lambda} g(\hat{y}) = 0. \tag{41}$$

**Stationarity Measure and Federated Gradients** We adopt the standard weakly convex notation to denote the stationarity measure as $\|w - \hat{w}(w)\|$. This coincides with the criteria used in Huang & Lin (2023). We also define the sub-gradient notations used in the proof,

$\tilde{\nabla}_t := \frac{1}{m} \sum_{j \in \mathcal{S}_t} \sum_{\tau=0}^{E-1} \nu_{j,\tau}^t$ (Partial federated gradient)

$\zeta_t := \frac{1}{n} \sum_{j=1}^{n} \sum_{\tau=0}^{E-1} \nu_{j,\tau}^t$ (Full federated gradient).

Following Huang & Lin (2023), we define the weakly convex potential function as,

$$\varphi(w) := \min_{y \in \mathcal{X}} \left\{ f(y) + \frac{\hat{\rho}}{2} \|y - w\|^2 \text{ s.t. } g(y) \leq 0 \right\},$$

with $\hat{\rho} > 2\rho$. And the minimizer is denoted as,

$$\hat{w}_t = \hat{w}(w_t) := \arg\min_{y \in \mathcal{X}} \left\{ f(y) + \frac{\hat{\rho}}{2} \|y - w_t\|^2 \quad \text{s.t.} \quad g(y) \leq 0 \right\}.$$

**Lemma 10** (Bound on the Lagrange Multiplier Huang & Lin (2023)). *Under Assumptions 6–9 and 2, let $\hat{w}(w)$ be the unique minimizer of the proximal problem equation 37, and let $\hat{\lambda}$ denote its optimal Lagrange multiplier. Then the multiplier satisfies the uniform bound*

$$\hat{\lambda} \leq \Lambda := \frac{GD + \hat{\rho} D^2}{-g(w_{\text{feas}})}, \tag{42}$$

*where $D$ is the diameter of $\mathcal{X}$ and $M$ is the Lipschitz constant in Assumption 8. This bound is identical to the centralized result of Huang & Lin (2023).*

*Proof sketch.* The proof follows the argument of Lemma 4.2 in Huang & Lin (2023), applied to the aggregated functions $f(w) = \frac{1}{n} \sum_{j=1}^{n} f_j(w)$ and $g(w) = \frac{1}{n} \sum_{j=1}^{n} g_j(w)$, which satisfy $\rho$-weak convexity, convexity, and $G$-Lipschitzness by Assumptions 2, and 6–9.

Let $\hat{y} = \hat{w}(w)$ and $\hat{\lambda}$ satisfy the KKT conditions in equation 38–equation 41, and pick $\nabla f(\hat{y}) \in \partial f(\hat{y}), \nabla g(\hat{y}) \in \partial g(\hat{y})$ and $u \in N_{\mathcal{X}}(\hat{y})$ such that

$$\nabla f(\hat{y}) + \hat{\rho}(\hat{y} - w) + \hat{\lambda} \nabla g(\hat{y}) + u = 0.$$

Taking the inner product with $w_{\text{feas}} - \hat{y}$ and using: (i) weak convexity of $f$, (ii) convexity of $g$, (iii) the normal cone property $\langle u, w_{\text{feas}} - \hat{y} \rangle \leq 0$, (iv) Slater's condition $g(w_{\text{feas}}) < 0$, and (v) the Lipschitz and bounded-domain bounds $\|\nabla f(\hat{y})\| \leq G$, $\|\nabla g(\hat{y})\| \leq G$, $\|w_{\text{feas}} - \hat{y}\| \leq D$, $\|w - \hat{y}\| \leq D$, yields

$$\hat{\lambda}\left(-g(w_{\text{feas}})\right) \leq GD + \hat{\rho}D^2.$$

Dividing by $-g(w_{\text{feas}}) > 0$ gives the claimed bound. $\qquad\square$

**Lemma 11** (Lemma 1 for Weakly Convex: Bounding the Inner Product). *Under the Assumptions 6, 7, 8 and 2, for all rounds $t \in [T]$ and any $\alpha > 0$, the following bound holds.*

$$-\eta \left\langle \frac{1}{n} \sum_{j=1}^{n} \sum_{\tau=0}^{E-1} \nu_{j,\tau}^t, w_t - \hat{w}_t \right\rangle$$

$$\leq \frac{1}{n} \sum_{j=1}^{n} \sum_{\tau=0}^{E-1} \begin{cases} \left(\frac{\eta}{\alpha} + \frac{3\eta\rho}{2}\right) \|w_t - w_{j,\tau}^t\|^2 + \eta\alpha G^2 + \eta(f_j(\hat{w}_t) - f_j(w_t)) \\ \qquad\qquad\qquad\qquad\qquad\qquad + \eta\rho\|\hat{w}_t - w_t\|^2, \quad \text{if } t \in \mathcal{A}, \\ \frac{\eta}{\alpha} \|w_t - w_{j,\tau}^t\|^2 + \eta\alpha G^2 + \eta(g_j(\hat{w}_t) - g_j(w_t)), \quad \text{if } t \in \mathcal{B}. \end{cases}$$

*Proof.* We analyze the inner product term by decomposing it with respect to the local iterates $w_{j,\tau}^t$. The reference point $w^*$ is replaced by $\hat{w}_t$:

$$-2\eta \left\langle \frac{1}{n} \sum_{j=1}^{n} \sum_{\tau=0}^{E-1} \nu_{j,\tau}^t, w_t - \hat{w}_t \right\rangle$$

$$= \frac{1}{n} \sum_{j=1}^{n} \sum_{\tau=0}^{E-1} \left[ \underbrace{-2\eta \left\langle \nu_{j,\tau}^t, w_t - w_{j,\tau}^t \right\rangle}_{\text{Term A}} \underbrace{-2\eta \left\langle \nu_{j,\tau}^t, w_{j,\tau}^t - \hat{w}_t \right\rangle}_{\text{Term B}} \right].$$

**Bounding Term A** For any $\alpha > 0$, Young's inequality and the $G$-Lipschitz assumption give:

$$\text{Term A} = -2\eta \left\langle \nu_{j,\tau}^t, w_t - w_{j,\tau}^t \right\rangle \overset{\text{Young's}}{\leq} \eta\alpha\|\nu_{j,\tau}^t\|^2 + \frac{\eta}{\alpha}\|w_t - w_{j,\tau}^t\|^2$$

$$\overset{G\text{-Lip}}{\leq} \frac{\eta}{\alpha}\|w_t - w_{j,\tau}^t\|^2 + \eta\alpha G^2.$$

This bound holds for both $t \in \mathcal{A}$ (using $\nabla f_j$) and $t \in \mathcal{B}$ (using $\nabla g_j$).

**Bounding Term B (Weakly Convex Gap) Case 1:** $t \in \mathcal{A}$ **(Objective Update)** Here $\nu_{j,\tau}^t = \nabla f_j(w_{j,\tau}^t)$. We use the definition of $\rho$-weak convexity:

$$f_j(\hat{w}_t) \geq f_j(w_{j,\tau}^t) + \left\langle \nabla f_j(w_{j,\tau}^t), \hat{w}_t - w_{j,\tau}^t \right\rangle - \frac{\rho}{2}\|\hat{w}_t - w_{j,\tau}^t\|^2$$

Rearranging this to isolate the inner product gives,

$$-\left\langle \nabla f_j(w_{j,\tau}^t), w_{j,\tau}^t - \hat{w}_t \right\rangle \leq f_j(\hat{w}_t) - f_j(w_{j,\tau}^t) + \frac{\rho}{2}\|\hat{w}_t - w_{j,\tau}^t\|^2.$$

Multiplying by $2\eta$ to get our bound for Term B:

$$\text{Term B} \leq 2\eta\left(f_j(\hat{w}_t) - f_j(w_{j,\tau}^t)\right) + \eta\rho\|\hat{w}_t - w_{j,\tau}^t\|^2$$

$$\overset{Young's}{\leq} 2\eta\left(f_j(\hat{w}_t) - f_j(w_{j,\tau}^t)\right) + 2\eta\rho\|\hat{w}_t - w_t\|^2 + 2\eta\rho\|w_t - w_{j,\tau}^t\|^2$$

So, for $t \in \mathcal{A}$

$$-2\eta \left\langle \zeta_t, w_t - \hat{w}_t \right\rangle \leq \frac{1}{n} \sum_{j=1}^{n} \sum_{\tau=0}^{E-1} \left[ \left(\frac{\eta}{\alpha} + 2\eta\rho\right)\|w_t - w_{j,\tau}^t\|^2 + \eta\alpha G^2 + 2\eta(f_j(\hat{w}_t) - f_j(w_{j,\tau}^t)) \right.$$

$$\left. + 2\eta\rho\|\hat{w}_t - w_t\|^2 \right].$$

Now our goal is to handle the term $f_j(\hat{w}_t) - f_j(w_{j,\tau}^t)$, so we first rewrite,

$$f_j(w_{j,\tau}^t) \geq f_j(w_t) + \langle \nabla f_j(w_t), w_{j,\tau}^t - w_t \rangle - \frac{\rho}{2} \left\| w_t - w_{j,\tau}^t \right\|^2 \quad \text{(by } \rho\text{-weak convexity)}$$

$$\Rightarrow f_j(\hat{w}_t) - f_j(w_{j,\tau}^t) \leq f_j(\hat{w}_t) - f_j(w_t) - \langle \nabla f_j(w_t), w_{j,\tau}^t - w_t \rangle + \frac{\rho}{2} \left\| w_t - w_{j,\tau}^t \right\|^2.$$

Using Young's inequality with parameter $\alpha > 0$, we get

$$-\langle \nabla f_j(w_t), w_{j,\tau}^t - w_t \rangle \leq \frac{1}{2\alpha} \|w_{j,\tau}^t - w_t\|^2 + \frac{\alpha}{2} \|\nabla f_j(w_t^t)\|^2.$$

Thus, we get,

$$f_j(\hat{w}_t) - f_j(w_{j,\tau}^t) \leq f_j(\hat{w}_t) - f_j(w_t) + \frac{1}{2\alpha} \|w_t - w_{j,\tau}^t\|^2 + \frac{\alpha}{2} \|\nabla f_j(w_t)\|^2 + \frac{\rho}{2} \left\| w_t - w_{j,\tau}^t \right\|^2$$

$$\overset{G-Lip}{\leq} f_j(\hat{w}_t) - f_j(w_t) + \frac{1}{2\alpha} \|w_t - w_{j,\tau}^t\|^2 + \frac{\alpha}{2} G^2 + \frac{\rho}{2} \left\| w_t - w_{j,\tau}^t \right\|^2.$$

Now we have

$$-2\eta \langle \zeta_t, w_t - \hat{w}_t \rangle \leq \frac{1}{n} \sum_{j=1}^n \sum_{\tau=0}^{E-1} \left[ \left( \frac{2\eta}{\alpha} + 3\eta\rho \right) \|w_t - w_{j,\tau}^t\|^2 + 2\eta\alpha G^2 + 2\eta(f_j(\hat{w}_t) - f_j(w_t)) \right.$$

$$\left. + 2\eta\rho \|\hat{w}_t - w_t\|^2 \right].$$

**Case 2:** $t \in \mathcal{B}$ (**Constraint Update**) Here $\nu_{j,\tau}^t = \nabla g_j(w_{j,\tau}^t)$. The constraint function $g_j$ is still convex. Therefore, the original logic holds, just with $\hat{w}_t$ as the reference point:

$$g_j(\hat{w}_t) \geq g_j(w_{j,\tau}^t) + \langle \nabla g_j(w_{j,\tau}^t), \hat{w}_t - w_{j,\tau}^t \rangle$$

Rearranging gives:

$$-\langle \nabla g_j(w_{j,\tau}^t), w_{j,\tau}^t - \hat{w}_t \rangle \leq g_j(\hat{w}_t) - g_j(w_{j,\tau}^t)$$

Multiplying by $2\eta$:

$$\text{Term B} \leq 2\eta \left( g_j(\hat{w}_t) - g_j(w_{j,\tau}^t) \right)$$

Similar to previous steps we can bound

$$-2\eta \langle \zeta_t, w_t - \hat{w}_t \rangle \leq \frac{1}{n} \sum_{j=1}^n \sum_{\tau=0}^{E-1} \left[ \frac{2\eta}{\alpha} \|w_t - w_{j,\tau}^t\|^2 + 2\eta\alpha G^2 + 2\eta(g_j(\hat{w}_t) - g_j(w_t)) \right].$$

Substituting the bounds for Term A and Term B back into the original expression furnishes the proof. $\qquad \square$

**Theorem 10** (FEDSGM: Weakly Convex $f$, Convex $g$)**.** *Consider the optimization problem in equation 1 under Assumptions 2, 4, and Assumptions 6–9. Define $\mathcal{A} := \{t \in [T] \mid \hat{G}(w_t) \leq \epsilon\}$, and set step-size $\eta$ and constraint threshold $\epsilon$ appropriately.*

$$\eta = \sqrt{\frac{D^2}{4G^2 E^3 T}} \quad \text{and} \quad \epsilon = \sqrt{\frac{4D^2 G^2 E}{T} + \frac{D^2 \rho}{8ET} + \frac{4GD}{\sqrt{mT}} \sqrt{2 \log(\frac{4}{\delta})} + 2\sigma \sqrt{\frac{2}{m} \log\left(\frac{8T}{\delta}\right)} (1 + \Lambda)},$$

*where $\Lambda$ is from Lemma 10 and $\delta \in (0,1)$ is a confidence parameter. Then, with probability at least $1 - \delta$, the following hold:*

1. *The set $\mathcal{A}$ is non-empty, so $w_t$ for $t \in \mathcal{A}$ is well-defined.*

2. *The following holds*

$$\frac{1}{|\mathcal{A}|} \sum_{t \in \mathcal{A}} \|w_t - \hat{w}(w_t)\| \leq \sqrt{\frac{(1+\Lambda)\epsilon}{(\hat{\rho} - 2\rho)}}, \qquad \frac{1}{|\mathcal{A}|} \sum_{t \in \mathcal{A}} g(w_t) \leq \epsilon.$$

*Proof.* We start the analysis with the potential function $\varphi(w_t)$, which is standard for weakly convex problems. The single-step progress is given by (see Huang & Lin (2023)):

$$\varphi(w_{t+1}) \leq f(\hat{w}_t) + \frac{\hat{\rho}}{2}\|w_{t+1} - \hat{w}_t\|^2$$

$$\overset{1-Lip \ \Pi_{\mathcal{X}}}{\leq} = f(\hat{w}_t) + \frac{\hat{\rho}}{2}\|w_t - \eta\tilde{\nabla}_t - \hat{w}_t\|^2$$

$$\leq f(\hat{w}_t) + \frac{\hat{\rho}}{2}\left(\|w_t - \hat{w}_t\|^2 + \eta^2\|\tilde{\nabla}_t\|^2 - 2\eta\left\langle\tilde{\nabla}_t, w_t - \hat{w}_t\right\rangle\right)$$

$$\overset{(i)}{\leq} \varphi(w_t) + \frac{\hat{\rho}\eta^2 E^2 G^2}{2} - \eta\hat{\rho}\left\langle\tilde{\nabla}_t, w_t - \hat{w}_t\right\rangle,$$

where in $(i)$ we used $\varphi(w_t) = f(\hat{w}_t) + \frac{\hat{\rho}}{2}\|w_t - \hat{w}_t\|^2$ and $\|\tilde{\nabla}_t\|^2 \leq E^2 G^2$.

Rearranging and adding/subtracting the full federated gradient $\zeta_t$ gives:

$$\eta\hat{\rho}\left\langle\zeta_t, w_t - \hat{w}_t\right\rangle \leq \underbrace{\varphi(w_t) - \varphi(w_{t+1})}_{\text{Telescoping}} + \underbrace{\frac{\hat{\rho}\eta^2 E^2 G^2}{2}}_{\text{Noise}} + \underbrace{\eta\hat{\rho}\left\langle\zeta_t - \tilde{\nabla}_t, w_t - \hat{w}_t\right\rangle}_{\text{Client Sampling Error}}. \quad (43)$$

Using Lemma 11, we can bound the LHS and we get

$$\eta\hat{\rho}E(f(w_t) - f(\hat{w}_t) - \rho\|w_t - \hat{w}_t\|^2)\mathbb{1}\{t \in \mathcal{A}\} + \eta\hat{\rho}E(g(w_t) - g(\hat{w}_t))\mathbb{1}\{t \in \mathcal{B}\} \overset{\alpha=\eta}{\leq} \varphi(w_t) - \varphi(w_{t+1})$$

$$+ \frac{\hat{\rho}\eta^2 E^2 G^2}{2} + \frac{\eta^3 \rho\hat{\rho}E^3 G^2}{2} + \eta^2\hat{\rho}EG^2 + \frac{1}{3}\eta^2\hat{\rho}E^3 G^2 + \eta\hat{\rho}\left\langle\zeta_t - \tilde{\nabla}_t, w_t - \hat{w}_t\right\rangle$$

$$(44)$$

**Bounding the LHS terms:**

- If $t \in \mathcal{A}$:

$$f(w_t) - f(\hat{w}_t) \geq f(w_t) - f(\hat{w}_t) - \frac{\rho}{2}\|w_t - \hat{w}_t\|^2$$

$$= f(w_t) - f(\hat{w}_t) - \frac{\hat{\rho}}{2}\|w_t - \hat{w}_t\|^2 + \frac{\hat{\rho} - \rho}{2}\|w_t - \hat{w}_t\|^2$$

  Now, consider the potential function defined above with $w = w_t$. By Assumption 9 there exists a Lagrangian multiplier $\hat{\lambda}_t \geq 0$ such that $\hat{\lambda}_t g(\hat{w}_t) = 0$ (complementary slackness) and

$$\hat{w}_t = \underset{w \in \mathcal{X}}{\arg\min}\left\{\Phi(w) := f(w) + \frac{\hat{\rho}}{2}\|w - w_t\|^2 + \hat{\lambda}_t g(w)\right\}.$$

  Since the objective function, $\Phi(w)$ above is $(\hat{\rho} - \rho)$-strongly convex and $\hat{w}_t$ is its minimizer, the value at any other point must be larger by at least the strong convexity quadratic,

$$\Phi(w_t) \geq \Phi(\hat{w}_t) + \frac{\hat{\rho} - \rho}{2}\|w_t - \hat{w}_t\|^2.$$

  Now, we know $\Phi(w_t) = f(w_t) + \frac{\hat{\rho}}{2}\|w_t - w_t\|^2 + \hat{\lambda}_t g(w_t) = f(w_t) + \hat{\lambda}_t g(w_t)$ and $\Phi(\hat{w}_t) = f(\hat{w}_t) + \frac{\hat{\rho}}{2}\|\hat{w}_t - w_t\|^2 + \hat{\lambda}_t g(\hat{w}_t)$. So, plugging it back in the inequality above we will get

$$f(w_t) + \hat{\lambda}_t g(w_t) \geq \left(f(\hat{w}_t) + \frac{\hat{\rho}}{2}\|w_t - \hat{w}_t\|^2 + \hat{\lambda}_t g(\hat{w}_t)\right) + \frac{\hat{\rho} - \rho}{2}\|w_t - \hat{w}_t\|^2$$

$$f(w_t) + \hat{\lambda}_t g(w_t) \geq f(\hat{w}_t) + \frac{\hat{\rho}}{2}\|w_t - \hat{w}_t\|^2 + \frac{\hat{\rho} - \rho}{2}\|w_t - \hat{w}_t\|^2$$

$$f(w_t) - f(\hat{w}_t) - \frac{\hat{\rho}}{2}\|w_t - \hat{w}_t\|^2 \geq -\hat{\lambda}_t g(w_t) + \frac{\hat{\rho} - \rho}{2}\|w_t - \hat{w}_t\|^2.$$

Therefore, for $t \in \mathcal{A}$, we have $f(w_t) - f(\hat{w}_t) - \rho\|w_t - \hat{w}_t\|^2 \geq -\hat{\lambda}_t g(w_t) + (\hat{\rho} - 2\rho)\|w_t - \hat{w}_t\|^2$. Now, plugging this term back in equation 44, we get for $\hat{\rho} > 2\rho$

$$\eta\hat{\rho}E[(\hat{\rho} - 2\rho)\|w_t - \hat{w}_t\|^2 - \hat{\lambda}_t g(w_t)]\mathbb{1}\{t \in \mathcal{A}\} + \eta\hat{\rho}E(g(w_t) - g(\hat{w}_t))\mathbb{1}\{t \in \mathcal{B}\} \leq \varphi(w_t) - \varphi(w_{t+1})$$

$$+ \frac{\hat{\rho}\eta^2 E^2 G^2}{2} + \frac{\eta^3 \rho \hat{\rho} E^3 G^2}{2} + \eta^2 \hat{\rho} E G^2 + \frac{1}{3}\eta^2 \hat{\rho} E^3 G^2 + \eta \hat{\rho} \left\langle \zeta_t - \tilde{\nabla}_t, w_t - \hat{w}_t \right\rangle$$

Again add and subtract $\hat{G}(w_t)$ from both the first and second term on LHS and rearrange and we get,

$$\eta \hat{\rho} E[(\hat{\rho} - 2\rho)\|w_t - \hat{w}_t\|^2 - \hat{\lambda}_t \hat{G}(w_t)]\mathbb{1}\{t \in \mathcal{A}\} + \eta \hat{\rho} E(\hat{G}(w_t) - g(\hat{w}_t))\mathbb{1}\{t \in \mathcal{B}\} \le \varphi(w_t) - \varphi(w_{t+1})$$

$$+ \frac{\hat{\rho}\eta^2 E^2 G^2}{2} + \frac{\eta^3 \rho \hat{\rho} E^3 G^2}{2} + \eta^2 \hat{\rho} E G^2 + \frac{1}{3}\eta^2 \hat{\rho} E^3 G^2$$

$$+ \eta \hat{\rho} \left\langle \frac{1}{n}\sum_{j=1}^{n}\sum_{\tau=0}^{E-1} \nabla f_j(w_{j,\tau}^t) - \frac{1}{m}\sum_{j \in \mathcal{S}_t}\sum_{\tau=0}^{E-1} \nabla f_j(w_{j,\tau}^t), w_t - \hat{w}_t \right\rangle \mathbb{1}_{t \in \mathcal{A}}$$

$$+ \eta \hat{\rho} \left\langle \frac{1}{n}\sum_{j=1}^{n}\sum_{\tau=0}^{E-1} \nabla g_j(w_{j,\tau}^t) - \frac{1}{m}\sum_{j \in \mathcal{S}_t}\sum_{\tau=0}^{E-1} \nabla g_j(w_{j,\tau}^t), w_t - \hat{w}_t \right\rangle \mathbb{1}_{t \in \mathcal{B}}$$

$$+ \eta \hat{\rho} E \hat{\lambda}_t(g(w_t) - \hat{G}(w_t))\mathbb{1}\{t \in \mathcal{A}\}$$

$$+ \eta \hat{\rho} E(\hat{G}(w_t)) - g(w_t))\mathbb{1}\{t \in \mathcal{B}\}$$

Now sum both sides over $t \in [T]$, we get

$$\sum_{t=0}^{T-1} \eta \hat{\rho} E[(\hat{\rho} - 2\rho)\|w_t - \hat{w}_t\|^2 - \hat{\lambda}_t \hat{G}(w_t)]\mathbb{1}\{t \in \mathcal{A}\} + \sum_{t=0}^{T-1} \eta \hat{\rho} E(\hat{G}(w_t) - g(\hat{w}_t))\mathbb{1}\{t \in \mathcal{B}\}$$

$$\le \varphi(w_0) - \varphi(w_T) + \frac{\hat{\rho}\eta^2 E^2 G^2 T}{2} + \frac{\eta^3 \rho \hat{\rho} E^3 G^2 T}{2} + \eta^2 \hat{\rho} E G^2 T + \frac{1}{3}\eta^2 \hat{\rho} E^3 G^2 T$$

$$+ \sum_{t \in \mathcal{A}} \eta \hat{\rho} \left\langle \frac{1}{n}\sum_{j=1}^{n}\sum_{\tau=0}^{E-1} \nabla f_j(w_{j,\tau}^t) - \frac{1}{m}\sum_{j \in \mathcal{S}_t}\sum_{\tau=0}^{E-1} \nabla f_j(w_{j,\tau}^t), w_t - \hat{w}_t \right\rangle$$

$$+ \sum_{t \in \mathcal{B}} \eta \hat{\rho} \left\langle \frac{1}{n}\sum_{j=1}^{n}\sum_{\tau=0}^{E-1} \nabla g_j(w_{j,\tau}^t) - \frac{1}{m}\sum_{j \in \mathcal{S}_t}\sum_{\tau=0}^{E-1} \nabla g_j(w_{j,\tau}^t), w_t - \hat{w}_t \right\rangle$$

$$+ \sum_{t \in \mathcal{A}} \eta \hat{\rho} E \hat{\lambda}_t(g(w_t) - \hat{G}(w_t))$$

$$+ \sum_{t \in \mathcal{B}} \eta \hat{\rho} E(\hat{G}(w_t)) - g(w_t))$$

Let us simplify some terms in both LHS and RHS. First in the LHS observe that $\hat{G}\mathbb{1}\{t \in \mathcal{A}\} \le \epsilon \mathbb{1}\{t \in \mathcal{A}\}$ and similarly we have $\hat{G}\mathbb{1}\{t \in \mathcal{B}\} \ge \epsilon \mathbb{1}\{t \in \mathcal{B}\}$. Now, on the RHS, we observe that $\varphi(w_0) - \varphi(w_T) \le \frac{\hat{\rho}D^2}{2}$. And we can bound the last 4 terms in the RHS using Lemma 4 and Lemma 7. So finally, we get for $\delta \in (0,1)$ with probability at least $1 - \delta$

$$\sum_{t=0}^{T-1} \eta \hat{\rho} E[(\hat{\rho} - 2\rho)\|w_t - \hat{w}_t\|^2 - \hat{\lambda}_t \epsilon]\mathbb{1}\{t \in \mathcal{A}\} + \sum_{t=0}^{T-1} \eta \hat{\rho} E \epsilon \mathbb{1}\{t \in \mathcal{B}\}$$

$$\le \frac{\hat{\rho}D^2}{2} + \frac{\hat{\rho}\eta^2 E^2 G^2 T}{2} + \frac{\eta^3 \rho \hat{\rho} E^3 G^2 T}{2} + \eta^2 \hat{\rho} E G^2 T + \frac{1}{3}\eta^2 \hat{\rho} E^3 G^2 T$$

$$+ 4\eta \hat{\rho} E G D \sqrt{\frac{2T}{m}\log(\frac{4}{\delta})} + \sum_{t=0}^{T-1} \eta \hat{\rho} E \sigma \sqrt{\frac{2}{m}\log\left(\frac{8T}{\delta}\right)} \left[\hat{\lambda}_t \mathbb{1}\{t \in \mathcal{A}\} + \mathbb{1}\{t \in \mathcal{B}\}\right]$$

Notice that $0 \le \hat{\lambda}_t \le \Lambda$, we get

$$\sum_{t=0}^{T-1} \eta \hat{\rho} E[(\hat{\rho} - 2\rho)\|w_t - \hat{w}_t\|^2 - \Lambda_t \epsilon]\mathbb{1}\{t \in \mathcal{A}\} + \sum_{t=0}^{T-1} \eta \hat{\rho} E \epsilon \mathbb{1}\{t \in \mathcal{B}\}$$

$$\le \frac{\hat{\rho}D^2}{2} + \frac{\hat{\rho}\eta^2 E^2 G^2 T}{2} + \frac{\eta^3 \rho \hat{\rho} E^3 G^2 T}{2} + \eta^2 \hat{\rho} E G^2 T + \frac{1}{3}\eta^2 \hat{\rho} E^3 G^2 T$$

$$+ 4\eta\hat{\rho}EGD\sqrt{\frac{2T}{m}\log(\frac{4}{\delta})} + \eta\hat{\rho}ET\sigma\sqrt{\frac{2}{m}\log\left(\frac{8T}{\delta}\right)}(1+\Lambda)$$

Dividing both sides by $\eta\hat{\rho}ET$ we get

$$\frac{1}{T}\sum_{t=0}^{T-1}[(\hat{\rho}-2\rho)\|w_t - \hat{w}_t\|^2 - \Lambda\epsilon]\mathbb{1}\{t \in \mathcal{A}\} + \frac{1}{T}\sum_{t=0}^{T-1}\epsilon\mathbb{1}\{t \in \mathcal{B}\}$$

$$\leq \frac{D^2}{2\eta ET} + \frac{\eta EG^2}{2} + \frac{\eta^2\rho E^2 G^2}{2} + \eta G^2 + \frac{1}{3}\eta E^2 G^2 + \frac{4GD}{\sqrt{mT}}\sqrt{2\log(\frac{4}{\delta})} + \sigma\sqrt{\frac{2}{m}\log\left(\frac{8T}{\delta}\right)}(1+\Lambda)$$

Now choosing $\eta = \sqrt{\frac{D^2}{2G^2 ET\Gamma}}$, where $\Gamma = \frac{1}{2}E + 1 + \frac{1}{3}E^2$, we get

$$\frac{1}{T}\sum_{t=0}^{T-1}[(\hat{\rho}-2\rho)\|w_t - \hat{w}_t\|^2 - \Lambda\epsilon]\mathbb{1}\{t \in \mathcal{A}\} + \frac{1}{T}\sum_{t=0}^{T-1}\epsilon\mathbb{1}\{t \in \mathcal{B}\}$$

$$\leq \sqrt{\frac{2D^2 G^2 \Gamma}{ET}} + \frac{D^2\rho E}{4T\Gamma} + \frac{4GD}{\sqrt{mT}}\sqrt{2\log(\frac{4}{\delta})} + \sigma\sqrt{\frac{2}{m}\log\left(\frac{8T}{\delta}\right)}(1+\Lambda)$$

Again, using a similar style of analysis as in previous proofs, if $\mathcal{A} = \phi$, then

$$\epsilon_{bad} < \sqrt{\frac{2D^2 G^2 \Gamma}{ET}} + \frac{D^2\rho E}{4T\Gamma} + \frac{4GD}{\sqrt{mT}}\sqrt{2\log(\frac{4}{\delta})} + \sigma\sqrt{\frac{2}{m}\log\left(\frac{8T}{\delta}\right)}(1+\Lambda).$$

Thus, if we set $\epsilon = \sqrt{\frac{2D^2 G^2 \Gamma}{ET}} + \frac{D^2\rho E}{4T\Gamma} + \frac{4GD}{\sqrt{mT}}\sqrt{2\log(\frac{4}{\delta})} + \sigma\sqrt{\frac{2}{m}\log\left(\frac{8T}{\delta}\right)}(1+\Lambda)$, $\mathcal{A} \neq \phi$. Now, let's consider two different cases here as well.

**Case 1:** When $\sum_{t\in\mathcal{A}}[(\hat{\rho}-2\rho)\|w_t - \hat{w}_t\|^2 - \Lambda\epsilon] \leq 0$. Then, we have

$$\sum_{t\in\mathcal{A}}(\hat{\rho}-2\rho)\|w_t - \hat{w}_t\|^2 \leq \Lambda\epsilon|\mathcal{A}|$$

$$\frac{1}{|\mathcal{A}|}\sum_{t\in\mathcal{A}}\|w_t - \hat{w}_t\|^2 \leq \frac{(1+\Lambda)\epsilon}{(\hat{\rho}-2\rho)}, \quad \text{(Adding positive quantity)}$$

$$\left(\frac{1}{|\mathcal{A}|}\sum_{t\in\mathcal{A}}\|w_t - \hat{w}_t\|\right)^2 \leq \frac{(1+\Lambda)\epsilon}{(\hat{\rho}-2\rho)}, \quad \text{(by convexity of } \|\cdot\|^2 \text{ and Jensen's)}$$

$$\frac{1}{|\mathcal{A}|}\sum_{t\in\mathcal{A}}\|w_t - \hat{w}_t\| \leq \sqrt{\frac{(1+\Lambda)\epsilon}{(\hat{\rho}-2\rho)}}.$$

Additionally, we know

$$\frac{1}{|\mathcal{A}|}\sum_{t\in\mathcal{A}}g(w_t) \leq \frac{1}{|\mathcal{A}|}\sum_{t\in\mathcal{A}}(g(w_t) - \hat{G}(w_t)) + \frac{1}{|\mathcal{A}|}\sum_{t\in\mathcal{A}}\hat{G}(w_t)$$

$$\overset{Lem.\ 7}{\leq} \epsilon + \sigma\sqrt{\frac{2}{m}\log\left(\frac{8T}{\delta}\right)}, \text{ with probability at least } 1-\delta.$$

**Case 2:** Otherwise, if $\sum_{t\in\mathcal{A}}[(\hat{\rho}-2\rho)\|w_t - \hat{w}_t\|^2 - \Lambda\epsilon] > 0$, then

$$\epsilon T \geq \sum_{t=0}^{T-1}[(\hat{\rho}-2\rho)\|w_t - \hat{w}_t\|^2 - \Lambda\epsilon]\mathbb{1}\{t \in \mathcal{A}\} + \sum_{t=0}^{T-1}\epsilon\mathbb{1}\{t \in \mathcal{B}\}$$

$$\epsilon T \geq |\mathcal{A}|\left[\frac{1}{|\mathcal{A}|}\sum_{t\in\mathcal{A}}(\hat{\rho}-2\rho)\|w_t - \hat{w}_t\|^2\right] - \Lambda\epsilon|\mathcal{A}| + \epsilon|\mathcal{B}|$$

$$\epsilon T \geq |\mathcal{A}|(\hat{\rho} - 2\rho)\left(\frac{1}{|\mathcal{A}|}\sum_{t \in \mathcal{A}}\|w_t - \hat{w}_t\|\right)^2 - \Lambda\epsilon|\mathcal{A}| + \epsilon|\mathcal{B}|$$

$$\epsilon|\mathcal{A}| \geq |\mathcal{A}|(\hat{\rho} - 2\rho)\left(\frac{1}{|\mathcal{A}|}\sum_{t \in \mathcal{A}}\|w_t - \hat{w}_t\|\right)^2 - \Lambda\epsilon|\mathcal{A}|$$

$$\epsilon(1 + \Lambda) \geq (\hat{\rho} - 2\rho)\left(\frac{1}{|\mathcal{A}|}\sum_{t \in \mathcal{A}}\|w_t - \hat{w}_t\|\right)^2$$

$$\frac{\epsilon(1 + \Lambda)}{(\hat{\rho} - 2\rho)} \geq \left(\frac{1}{|\mathcal{A}|}\sum_{t \in \mathcal{A}}\|w_t - \hat{w}_t\|\right)^2$$

$$\sqrt{\frac{\epsilon(1 + \Lambda)}{(\hat{\rho} - 2\rho)}} \geq \frac{1}{|\mathcal{A}|}\sum_{t \in \mathcal{A}}\|w_t - \hat{w}_t\|.$$

And as before $\frac{1}{|\mathcal{A}|}\sum_{t \in \mathcal{A}} g(w_t) \leq \epsilon + \sigma\sqrt{\frac{2}{m}\log\left(\frac{8T}{\delta}\right)}$ with probability at least $1 - \delta$ for $\delta \in (0, 1)$. $\qquad\square$

## F  EXPERIMENTS

### F.1  CMDP EXPERIMENT DETAILS

Table 2: Detailed setting of CMDP experiment on Cartpole

| Hyperparameter | Value | Hyperparameter | Value |
|---|---|---|---|
| Local epochs (E) | 1 | Communication rounds (T) | 500 |
| Neural network dimension ($d$) | 17,410 | Total number of clients ($n$) | 10 |
| Batch size per client | 1,000 | Number of runs | 5 |
| Episode length | 200 | Discount factor ($\gamma$) | 1.0 |

Common details of Cartpole experiments are provided in Table 2. Constraint follows the two conditions used by Xu et al. (2021): a player is given a cost of 1 every time step for (1) violating five prohibited areas defined as $[-2.4, -2.2], [-1.3, -1.1], [-0.1, 0.1], [1.1, 1.3], [2.2, 2.4]$ or (2) exceeding the pole angle by more than 6 degrees.

### F.2  NP CLASSIFICATION EXPERIMENT DETAILS

**Hyperparameter** We tune the learning rate on a logarithmic scale ranging from 1 down to $1e - 4$. We set $\epsilon = 0.05$ for all our experiments and plot the best result. Additionally, we use the Top$-K$ compressor with $K/d = 0.1$ for both server and client compressors. Unless stated otherwise, we set global communication rounds $T = 500$, local epochs $E = 5$, total number of clients $n = 20$, and 50% as the participation rate. In the case of centralized training, we store all the training data locally and train the model for $T \times E$ epochs. Then, for a fair comparison, the results are reported every $E$ epochs, aligning with the frequency of model aggregation in FL. We report the experiments run for 3 different seed values and then report the mean with the variance bands.

We consider the constrained optimization problem in equation 1, where the objective is to minimize the empirical loss on the majority class while ensuring that the loss on the minority class remains below a prescribed tolerance. For each client $j$, the local objective and constraint are defined as

$$f_j(w) := \frac{1}{m_{j0}}\sum_{x \in \mathcal{D}_j^{(0)}}\phi(w; (x, 0)), \qquad g_j(w) := \frac{1}{m_{j1}}\sum_{x \in \mathcal{D}_j^{(1)}}\phi(w; (x, 1)), \qquad (45)$$

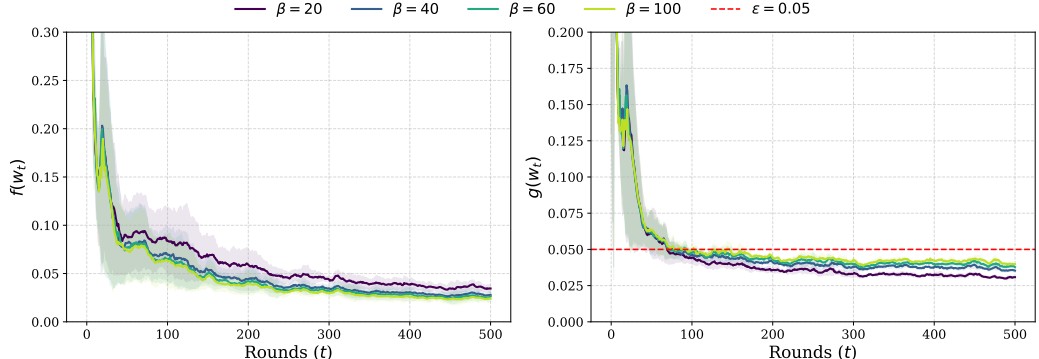

Figure 5: *NP classification:* Comparison of soft switching parameter in FedSGM. Solid lines and shaded areas indicate the mean and standard deviation over 3 runs with different random seeds respectively. Red dotted line indicates the constraint violation threshold $\epsilon$.

where $\mathcal{D}_j^{(0)}$ and $\mathcal{D}_j^{(1)}$ denote local samples of class 0 and class 1, respectively, and $m_{j0}, m_{j1}$ are their cardinalities. The function $\phi$ is the binary logistic loss,

$$\phi(w; (x,y)) = -y\,w^\top x + \log\left(1 + e^{w^\top x}\right), \quad y \in \{0, 1\}. \tag{46}$$

This captures the NP paradigm: $f(w)$ enforces performance on the majority class, while the constraint $g(w) \le \epsilon$ ensures that the loss on the minority class does not exceed the tolerance.

We use the breast cancer dataset (Wolberg et al., 1993), containing 569 samples with 30 features. We utilize 80% of the data for training and the rest for testing. The training data is then split in an IID fashion between $n = 20$ clients, such that each client receives an equal number of samples and the same class ratio.

For communication efficiency, both on the uplink and downlink sides, we evaluate the use of Top$-K$ (Haddadpour et al., 2021; Shulgin & Richtárik, 2022; Gruntkowska et al., 2024; Islamov et al., 2025) compressor, which transmits a fraction $K/d$ of coordinates. Performance is measured in terms of the majority-class objective loss $f(w)$, the minority-class constraint loss $g(w)$, and the number of rounds in which the feasibility condition $g(w) \le \epsilon$ is violated. From Figure 1, we observe that both hard and soft switching achieve convergence of the majority-class objective $f(w)$ while satisfying the constraint $g(w)$.

Finally, the soft switching parameter $\beta$ in FedSGM is exponentially easier to tune than penalty-based methods. In Figure 5, we compare differernt choices of $\beta$ around the value $\beta = 40$ suggested by our theoretical guarantee $\beta > \frac{2}{\epsilon}$. For lower $\beta = 20$, the convergence becomes noticeably less stable and more conservative on constraint satisfaction. For higher $\beta$, the behavior becomes asymptotically closer to its hard switching counterpart. In all experiments, we used the theoretically suggested value or tuned the value to about 2-3 times the point.

In the meantime, Figure 6 shows that the counterpart baseline (penalty-based FedAvg) is extremely unstable with the choice of penalty parameter $\rho$. Mild change in the parameter to $\rho = 0.5$ immediately makes the algorithm to converge to an infeasible solution. Although theory requires the penalty parameter to explode to infinity, larger $\rho$ prohibits the convergence performance.

### F.3  FAIR CLASSIFICATION EXPERIMENT

**Fair Classification with Demographic Parity.**    Fair classification task is formulated by the local objective of binary cross entropy loss and constraint of demographic parity:

$$f_j(w) := \frac{1}{m_j} \sum_{(x,y) \in \mathcal{D}_j} [y \log(\pi(x; w)) + (1-y) \log(1 - \pi(x; w))]$$

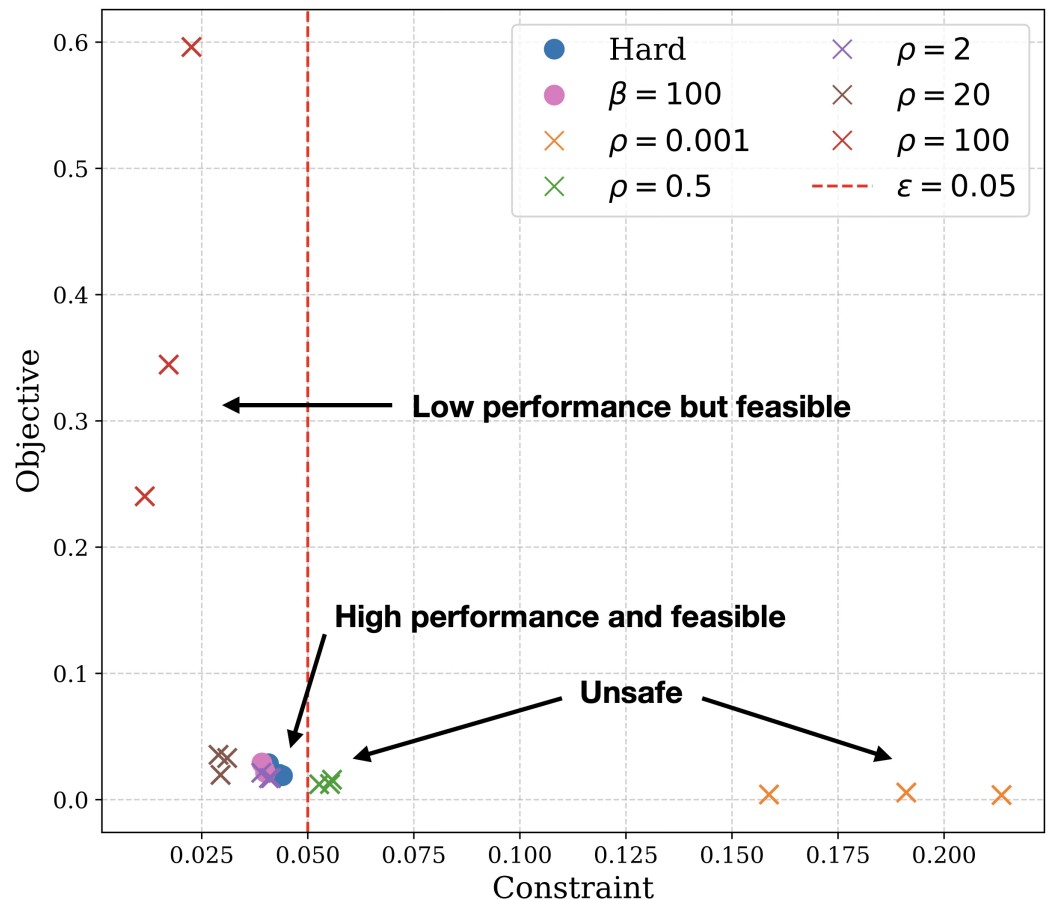

Figure 6: *NP classification:* Comparison of FedSGM against penalty-based FedAvg. Each point represents a result from a separate experimental run with different random seeds. Circles represent the FedSGM algorithm (soft and hard switching), while crosses ('X') represent penalty-based FedAvg with varying penalty parameters

$$g_j(w) := \left| \frac{1}{m_{j,p}} \sum_{x \in \mathcal{D}_{j,p}} \pi(x; w) - \frac{1}{m_{j,u}} \sum_{x \in \mathcal{D}_{j,u}} \pi(x; w) \right|,$$

where the subscript $p$ and $u$ represent protected and unprotected groups respectively. Since the demographic parity is defined as the absolute difference between the average logits of protected and unprotected groups, the aggregation at the server is treated with extra care. At the server, the average logits were aggregated instead of the final constraint value to recalculated the weighted average of logits for the global constraint calculation:

$$g(w) := \left| \frac{1}{m_{\mathcal{S},p}} \sum_{x \in \mathcal{D}_{\mathcal{S},p}} \pi(x; w) - \frac{1}{m_{\mathcal{S},u}} \sum_{x \in \mathcal{D}_{\mathcal{S},u}} \pi(x; w) \right|,$$

where the subscript $\mathcal{S}$ denote all sampled clients. Here, we explore the setting of stochastic data sampling, heterogeneous client data distribution, and deep neural network, making the problem highly non-convex, stochastic, skewed, and non-smooth. The experiments were conducted on Adult dataset (Kohavi & Becker, 1996) using $n = 10, m = 5, E = 2, T = 500, \epsilon = 0.05, \eta = 0.0001$, and $\alpha = 2.0$ for client selection from Dirichlet distribution. For the penalty-based methods, the penalty parameter was chosen from $\rho \in [0.1, 1.0, 10.0]$ and only the best results are presented. Aligned with the findings in Figure 6, higher values produced slow convergence and lower values produced infeasible solutions.

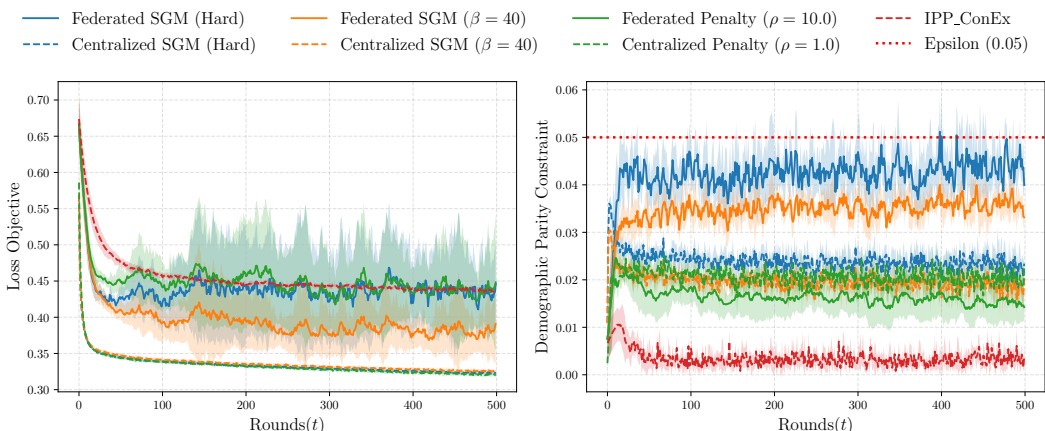

Figure 7: *Fair classification:* Comparison of FedSGM against centralized counterpart, centralized/federated penalty-based method and IPP ConEx. FedSGM (blue/orange) achieves the best balance of the trade-off between loss minimization and fairness violation without penalty parameter tuning.

In Figure 7, we report the mean and standard deviation bands over five runs with different random seeds. Centralized algorithms in dashed lines are more conservative than their federated version, while IPP ConEx being the most conservative. Centralized penalty-based and switching gradient methods achieve the fastest loss convergence while satisfying the fairness constraint. With appropriately tuned penalty parameter, penalty-based FedAvg (green solid line) bahaves comparably to FedSGM with hard switching (blue solid line). FedSGM with soft switching (orange solid line) outperforms both in terms of objective and constraint, as well as the stability of the training. This confirms our theoretical findings from Remark 1 that soft switching effectively mitigates the locally-induced skewness from the heterogeneity.

## G  RELATED WORKS

FL has become a foundation in privacy-preserving learning since McMahan et al. (2017) with key applications in healthcare (Rieke et al., 2020; Peng et al., 2024), battery management systems (Wang et al., 2024; Zhu et al., 2024; Banabilah et al., 2022), finance (Wen et al., 2023; Chatterjee et al., 2023) to name a few. While most existing FL algorithms focus on unconstrained optimization, many of these applications naturally involve constraints, such as fairness requirements, safety limits, or resource budgets (Du et al., 2021; Huang et al., 2024; Zhang et al., 2024; Salehi & Hossain, 2021; Yang et al., 2022). This motivates the development of constrained FL algorithms, which explicitly incorporate these constraints into the learning process to ensure reliable learning.

FedAvg, introduced in McMahan et al. (2017), remains the most widely used FL algorithm. In addition, there have been many variants of it to tackle different aspects of learning, such as data heterogeneity (Karimireddy et al., 2020; Seo et al., 2024; Morafah et al., 2024), system heterogeneity (Gong et al., 2022; Li et al., 2020a; Wu et al., 2024), and fairness (Li et al., 2021; Badar et al., 2024). Beyond FedAvg, ADMM-based FL algorithms have been explored to tackle heterogeneity in FL Acar et al. (2021); Gong et al. (2022); Zhou & Li (2023). Despite the rich FL literature proposed previously, the majority of works primarily focus on unconstrained FL problems, leaving a gap between constrained optimization and FL.

The most common setting in which 1 is analyzed is with $n = 1$, i.e., the centralized setting. Most works address this scenario by formulating it as a saddle-point problem and solving it using various primal-dual methods (Nemirovski, 2004; Chambolle & Pock, 2011; Bertsekas, 2014; Hamedani & Aybat, 2021; Zhang & Lan, 2022; Hounie et al., 2023; Boob & Khalafi, 2024). While theoretically grounded, these approaches can be highly sensitive to the tuning of the dual variables and typically require prior knowledge of an upper bound on the optimal dual variable. Moreover, many of

these methods involve projecting both primal and dual variables onto bounded sets, which are often unknown in practice, making implementation challenging.

Another primal-only approach is the switching (sub-)gradient method (SGM) proposed by Polyak (1967). This method has been extensively studied across different settings. For instance, Titov et al. (2018); Stonyakin et al. (2019); Alkousa (2020) studies mirror descent integrated with SGM. Additionally, Lan & Zhou (2020) extended SGM to the stochastic regime, establishing iteration complexity results under both convex and strongly convex assumptions. The analysis has also been broadened to weakly convex problems in works such as Huang & Lin (2023); Liu & Xu (2025); Ma et al. (2020); Jia & Grimmer (2025) extend the analysis of SGM to the weakly convex setting. In the federated learning context, Islamov et al. (2025) provides a seminal extension of SGM incorporating communication compression, though their framework does not address local multi-step updates or partial client participation. Building on these ideas, and drawing inspiration from Upadhyay et al. (2025), we, in addition to the hard switching, introduce a soft-switching variant of SGM and demonstrate its practical advantages.

## G.1 Novelty in Comparison to Baselines

FEDSGM is the first method that simultaneously and provably addresses functional constraints, multi-step local updates, bi-directional compression with error feedback, and partial participation in federated learning. Existing approaches do not apply to this setting, which is why direct comparisons to them are neither feasible nor relevant.

To examine the issue at its root and recognize the novelty of our method, we first clarify the difference between **set constraints** and **functional constraints**. Standard set constrained optimization often assumes a "simple" constraint set $\mathcal{X}$ (e.g., a box or ball) where projection is computationally cheap. However, in our work, we address functional constraints of the form $g(w) \leq 0$ like the loss function in our NP-classification experiment.

We will go into the details of existing methods, specifically projection, penalty, and the Frank-Wolfe method.

**Why FEDSGM is not comparable to projected or proximal methods?** We can observe that,

$$\min_w f(w) \quad \text{s.t.} \quad g(w) \leq 0,$$

is equivalent to an unconstrained problem with an indicator function as a regularizer (with $\epsilon-$ feasibility satisfaction):

$$\min_w f(w) + R(w), \quad \text{where} \quad R(w) = \begin{cases} 0 & \text{if } g(w) \leq \epsilon \\ \infty & \text{otherwise} \end{cases}.$$

The standard class of algorithms for this formulation is Proximal Gradient Descent (PGD) (Condat & Richtárik, 2022). A PGD step would be:

$$w_{t+1} = \text{prox}_{\eta R}(w_t - \eta \nabla f(w_t)).$$

However, the bottleneck of this computation is the proximal operator, $\text{prox}_{\eta R}(v)$. By definition, this is

$$\text{prox}_{\eta R}(v) = \arg\min_w (\eta R(w) + \frac{1}{2}\|w - v\|^2) = \arg\min_{\{w \text{ s.t. } g(w) \leq \epsilon\}} \|w - v\|^2. \quad (Projection)$$

This is the Euclidean projection onto the feasible set $\mathcal{W}_\epsilon^g = \{w \mid g(w) \leq \epsilon\}$. In case of full participation, the server can compute $g$ by doing the inner solve where it asks the clients to send $g_j(w)$ multiple times (ours only require one constraint evaluation and communication) to solve $(Projection)$. For a general, non-linear convex function $g(w)$, this projection does not have a closed-form solution. Rather, it is a complex constrained optimization problem in itself that must be solved in every iteration and requires multiple communication rounds. This bottleneck is precisely what is violated in our problem setting.

It is the motivation of our paper to develop a method that avoids this intractable projection. Our FEDSGM framework is explicitly projection-free and primal-only.

**Why penalty-based FedAvg is not an appropriate baseline?**   Consider the following penalty-based formulation instead of our problem (1) in the paper,

$$\min_w f(w) + \rho[g(w) - \epsilon]_+ \quad (Penalty)$$

It is evident that when $g(w) \leq \epsilon$ the (sub-)gradient is $\nabla f(w)$ while when $g(w) > \epsilon$ the (sub-)gradient is $\nabla f(w) + \rho \nabla g(w)$. Thus, for large $\rho$ such that $\nabla f(w) + \rho \nabla g(w) \approx \rho \nabla g(w)$, the penalty formulation in $(Penalty)$ would be a simplified approximation of the SGM dynamics. To precisely emulate the dynamics of hard switching, we must set $\rho$ to be very large. However, making $\rho$ large induces ill-conditioning, slowing optimization.

On the other hand, any fixed $\rho$ that is too small can leave persistent violations. The "right" $\rho$ depends on the relative scale and geometry of $\nabla f$ and $\nabla g$. FEDSGM avoids this hyperparameter tuning by switching directions (objective vs. constraint), i.e., no penalty weight to tune, no conditioning blow-up. Therefore, this penalty-based FedAvg is not an appropriate baseline.

However, to elucidate the above points, we create the heuristics to run a penalty-based federated algorithm where in each round the clients update their gradient as

$$\nu_{j,\tau}^t = \nabla f_j(w_{j,\tau}^t) + \mathbb{1}_{\{\hat{G}(w_t) > \epsilon\}} \rho \nabla g_j(w_{j,\tau}^t).$$

We compare FEDSGM against the "Penalty-FedAVG" with varying $\rho \in [0.001, 0.5, 2, 20, 100]$. We present the result in Figure 6. The result confirms that $\rho$ has a significant impact on performance as we expected. A small $\rho$ such as $0.001$ or $0.5$ does not satisfy the feasibility criteria, while a large $\rho = 100$ (which essentially should have been indicative of switching dynamics) makes the problem ill-conditioned and gives low performance.

**Why additional comparison with Frank-Wolfe is not suitable?**   Frank-Wolfe Pokutta (2024); Dadras et al. (2024) requires access to a Linear Minimization Oracle (LMO) over the constraint set at each iteration, formally written as,

$$s_t = \arg\min_{s \in \boldsymbol{\mathcal{X}}} \langle s, \nabla f(w_t) \rangle. \quad (Frank - Wolfe)$$

While Frank-Wolfe is efficient for "simple" set constraints such as a polytope, box, or simplex, for general non-smooth convex functional constraints $g(w) \leq 0$, solving the LMO is computationally infeasible. For our case in FL, where $g(w) = \frac{1}{n} \sum g_i(w)$, the constraint boundary is defined by distributed clients. Therefore, solving the LMO would require an iterative subroutine (like projected gradient descent), where every inner iteration requires a full communication round. Thus, just like projection-based methods, a Frank-Wolfe approach is communication-prohibitive in the functional constraint regime.

As no existing method in FL supports functional constraints with partial participation, local updates, and compression with error feedback simultaneously, a direct comparison is not feasible. Our work bridges this gap by characterizing the interplay between factors that are well motivated in federated learning, establishing realistic, functionally constrained framework.

