# OpenReview forum: "FedSGM: A Unified Framework for Constraint Aware, Bidirectionally Compressed, Multi-Step Federated Optimization"
_ICLR.cc/2026/Conference — Submitted to ICLR 2026_

### Official Review · Reviewer_Q7y9 · 2025-10-28

**Soundness:** 3
**Presentation:** 3
**Contribution:** 3
**Rating:** 6
**Confidence:** 3

**Summary:**

The authors propose FedSGM, a unified framework for federated constrained optimization that addresses four challenges in federated learning: functional constraints, bidirectional communication compression, multi-step local updates, and partial client participation. Building on the Switching Gradient Method (SGM), FedSGM performs primal-only, projection-free updates that alternate between minimizing the objective and the constraint, enabling feasibility without dual variables. The authors derive convergence guarantees for both hard and soft switching regimes, proving an optimal $\mathcal{O}(1/\sqrt{T})$ rate even with biased compression (via error feedback) and random client sampling. Extensive theoretical analysis covers convex functional constraints, communication noise, and participation variance. Empirical results on Neyman–Pearson classification and constrained MDP tasks demonstrate that FedSGM achieves constraint satisfaction and stable convergence under heterogeneous and lossy federated settings.

**Strengths:**

- This work tackles a challenge: federated optimization with constraints (such as fairness or safety requirements) in realistic federated learning (FL) conditions, which involve limited communication and sporadic client participation. Previous FL methods primarily address either heterogeneity (FedAvg variants) or constraints (e.g., FedAvg with projection) or compression.

- By building on the Switching Gradient Method, FedSGM avoids expensive projections or dual updates needed in prior constrained FL approaches (e.g., penalty/ADMM methods).

- The paper provides a rigorous theoretical analysis covering multiple challenging aspects equally. It proves convergence for both hard and soft switching regimes under a broad range of settings (full or partial client participation, with or without compression, and multiple local steps).

- The paper is generally well-written and structured.

**Weaknesses:**

- The theoretical guarantees rely on the convexity of both the objective and constraint functions (Assumption 1), along with bounded Lipschitz constants and sub-Gaussian noise assumptions. This is a standard assumption for proving $O(1/\sqrt{T})$ rates, but it limits the immediate applicability to realistic federated learning settings (most real federated tasks, such as deep neural network training, are nonconvex).

- Combining constraints, compression, local updates, and partial clients in a single framework is beneficial, but the technical novelty of each component may appear incremental compared to prior art. FedSGM essentially merges well-known techniques, such as switching gradient updates for constraints, error-feedback compression, and multi-step local SGDs (similar to FedAvg), and extends existing analyses to their combined application.

- Although the two case studies are relevant, the experiments are somewhat limited in diversity and scale.

**Questions:**

- In practice, how does FedSGM handle non-convex objectives? While the theory assumes convex $f_i$ and $g_i$, the experiments involve neural networks and reinforcement learning (both non-convex). Does the algorithm still converge reliably in these settings?

- Could the authors provide guidance or intuition on selecting the soft switching parameters (such as the smoothing coefficient $\beta$ or threshold $\epsilon$ in the sigmoid function)? How sensitive is FedSGM’s performance to this choice?

- How does FedSGM compare against simpler baseline approaches for constrained federated learning? For instance, one could run FedAvg with a penalty for constraint violation or project the global model onto the constraint set each round (if feasible).

- The analysis introduces the factor $\Gamma(q,q_0)$ to capture compression in presence of local steps. Could the authors elaborate on how error feedback interacts with multiple local SGD steps?

---

> ### Author Response · Authors · 2025-11-21
> **Response to Reviewer Q7y9 (Part 1/2)**
>
> We thank the reviewer for recognizing that $FedSGM$ tackles the critical challenge of constrained federated optimization under realistic conditions like limited communication and partial participation. We appreciate the recognition of our theoretical rigor in proving convergence across these diverse settings without relying on expensive projections.
>
> >**Weakness (1) and (2)**: *Theoretical guarantees, limiting applicability due to non-convexity....*
>
> We agree that limiting the analysis to convexity restricts applicability to deep learning. To address this, we have significantly expanded our theoretical analysis in the Appendix (D and E). As detailed in the General Response, we now provide a convergence proof for $\rho$-Weakly Convex objectives (a standard class of non-convex functions covering neural networks), demonstrating that $FedSGM$ is robust to both non-convex setting.
>
> Regarding Lipschitz continuity and sub-Gaussian noise, we clarify that these are not a limitation of our framework, but rather standard and necessary assumptions in optimization (also used in FedAvg, SCAFFOLD, etc.) to bound gradient variance and ensure convergence in partial participation settings. In unconstrained FL, a standard assumption widely used in the analysis is bounded client dissimilarity or bounded global variance, i.e, $\mathbb{E}[\\|\nabla f_i(w) - \nabla f(w)\\|^2] \le \sigma^2$. In constrained FL with functional constraints present in the optimization problem it is necessary to impose an analogous bound on the heterogeneity of the constraint function $g(w)$ across clients. Assumption 3, which states that $\hat{G} - g$ is sub-Gaussian, is simply the high-probability equivalent of this standard dissimilarity bound present in literature (such as FedAVG, SCAFFOLD, FedProx, etc.) and applied to $g$. Without bounding this heterogeneity, estimating the global constraint $\sum g_i(w)$ from a subset of clients would be theoretically impossible (as the variance would be unbounded). Thus, this assumption defines its validity in FL problems rather than restricting the algorithm's theory.
>
>
> >**Weakness (3)**: *Technical Novelty....*
>
> We thank the reviewer for their question regarding the technical novelty of this work. We emphasize that the combination of these components is not trivial, especially with partial client participation and local updates.
> As the reviewer noted, analyzing these four components is the baseline requirement for modern Federated Learning. Local updates, partial participation, and compression are requisistes of the resource (bandwidth/latency) constrained edge devices, while functional constraints are required for modern training paradigms (Safety/Fairness). The server must decide whether to optimize $f$ or $g$ based on a stochastic estimate $\hat{G}(w)$ derived from a subset of clients. Near the constraint boundary, sampling noise causes the server to make erroneous switching decisions (e.g., minimizing $f$ when the global state is infeasible). We tackle this by utilizing the soft switching which ensures that this instability around the constraint boundary remains controlled. Also, standard expectation based analysis fails to handle these binary, stochastic decision flips. Our novelty lies in utilizing the sub-Gaussian Assumption (Assumption 3) to establish rigorous High-Probability bounds.
>
> Additionally, we have also added a convergence result for **$\rho$-weakly convex objectives** with convex constraints under partial participation and local updates. This covers a broad class of non-convex problems standard in neural network training. The primary challenge in this setting is the absence of unique minimizer which is not guaranteed in non-convex settings. We overcame this by utilizing classical Moreau envelope based analysis for weakly convex functions. Please refer to Appendix (E) for the complete proof.
>
>
> >**Weakness (4)**: *Limited Experiments...*
>
>
> Our existing Neyman Pearson classification experiment covered the case of classical constrained optimization problem under federated learning setting by applying simple logistic regression. On the other hand, the deep RL experiment showcased that FedSGM also works on highly non-convex, stochastic and heterogeneous environments.
>
> Based on the reviewer's suggestion, to further strengthen our applicability we introduced a new fair classification experiment on ADULT dataset using a Deep NN under heterogeneous (non-IID) data distributions and stochastic gradients. Please refer to [New: Figure 2](https://imgur.com/ppeEMIt), General Response, and Appendix (F).
>
> References:
>
> 1.  Boob et al., "Stochastic first-order methods for convex and nonconvex functional constrained optimization" (Mathematical Programming 2022).
>
> >**Question (1)**: *Non-convexity and Experiments....*
>
> Please refer to the answer in Weakness (1),(3) and Weakness(4).
>
> $\small(continued\ in\ the\ 2nd\ part)$

---

> ### Author Response · Authors · 2025-11-21
> **Response to Reviewer Q7y9 (Part 2/2)**
>
> >**Question (2)**: *Intuition for Soft-switching....*
>
> We thank the reviewer for this practical question. One strength of our framework is that parameter selection is theory-guided rather than heuristic. Our analysis provides explicit recipes for both $\epsilon$ and $\beta$. We set our $\epsilon \approx \boldsymbol{\mathcal{O}}\left(1/\sqrt{T}\right)$. This ensures the constraint tolerance matches the expected optimization error. In our experiments with $T=500$, this yields $\epsilon \approx 0.045$, which we rounded to 0.05. Additionally, Theorem 2 establishes the condition $\beta \geq \frac{2}{\epsilon}$. This ensures that the switching function is sharp enough to approximate the indicator while maintaining sufficient smoothness for convergence. For $\epsilon=0.05$, this lower bound is $\beta \geq 40$.
>
> To validate this theoretical results, we conducted a smoothing coefficient ablation varying $\beta \in [20, 40, 60, 100]$. As predicted by the theory, performance degrades for $\beta < 40$. However, for $\beta \geq 40$, the performance is highly stable and robust. This confirms that we can simply follow the theoretical lower bound ($\beta = 2/\epsilon$) without extensive tuning. Please see [New: Figure 3](https://imgur.com/0qVLmRN).
>
> >**Question (3)**: *Other baselines....*
>
> Please refer to the General Response for this. Additonally, based on reviewers suggestion we added 2 new experiments where we do compare against the heuristic based Penalty method. Please see [New: Figure 1](https://imgur.com/HPhwHK9) and [New: Figure 2](https://imgur.com/ppeEMIt).
>
> >**Question (4)**: *Additional factor $\Gamma(q,q_0)$....*
>
>
> This is a key technical aspect of the analysis. The interaction between error feedback (EF) and multiple local updates ($E>1$) is non-trivial, as the "client drift" caused by local steps and the "compression error" maintained by EF become coupled. This coupling appears explicitly in how we bound the compression residual $e_{t+1}$ (Appendix B.4).
>
> **Bounding the uplink error:**
> When a client performs $E$ local updates before communication, its cumulative local trajectory $\sum_{\tau=0}^{E-1}\nu_{j,\tau}^t$ interacts with the previous compression residual $e_j^t$. The recursive EF relation yields:
>
> $$
> \mathbb{E}\left[\\|e_{t+1}\\|^2\right] \le (1-q)(1+\gamma)\mathbb{E}\left[\\|e_{t}\\|^2\right]+ (1-q)(1+\gamma^{-1})\mathbb{E}\left[\Big\\|\sum_{\tau=0}^{E-1}\nu_{j,\tau}^{t}\Big\\|^{2}\right].
> $$
>
> By the Lipschitz gradient assumption, $\left\\|\sum_{\tau=0}^{E-1}\nu_{j,\tau}^{t}\right\\|^{2}\le E^{2}G^{2}$, so the compression residual scales jointly with the number of local steps ($E$) and the compression accuracy ($q$). Propagating this bound through the full EF recursion produces coupled terms such as
>
> $$
> \mathbb{E}[\\|e_t\\|] \leq \mathcal{O} \left(E G\frac{\sqrt{1-q}}{q}\right),
> $$
>
> and similarly we can bound the downlink error $\hat{e}_t$ as
>
> $$
> \mathbb{E}[\\|\hat{e}_t\\|] \leq \mathcal{O} \left(E G \frac{\sqrt{(1-q_0)}}{q_0q}\right).
> $$
>
> Together, they form the factor $\Gamma(q,q_0)$. Hence, $\Gamma(q,q_0)$ does not merely sum the effects of local drift and compression noise; it quantifies their coupling through EF. This term reflects how residual error is affected by multiple local updates and is bounded globally.

---

### Official Review · Reviewer_PoVG · 2025-10-29

**Soundness:** 3
**Presentation:** 3
**Contribution:** 3
**Rating:** 4
**Confidence:** 3

**Summary:**

The paper proposes FEDSGM, a unified framework for federated constrained optimization that simultaneously handles four major challenges: (1) functional constraints, (2) bi-directional communication compression, (3) multiple local updates per round, and (4) partial client participation. Built upon the Switching Gradient Method, FEDSGM adopts a projection-free, primal-only update scheme that avoids dual variable tuning and inner-loop projections. The framework incorporates error-feedback mechanisms to correct compression bias and establishes an $O(1/\sqrt{T})$ convergence rate under convex objectives. Experiments on Neyman–Pearson classification and constrained MDPs demonstrate convergence and constraint satisfaction.

**Strengths:**

1. The work is the first to combine constraint handling, compression, multiple local steps, and partial participation in a single theoretical framework.
2. The convergence proof carefully separates optimization and sampling errors and analyzes the bias introduced by compression and local drift. Both hard and soft switching variants are theoretically justified.
3. FEDSGM avoids inner projections and dual updates, leading to a lightweight and practical approach for resource-constrained federated systems.
4. Experiments on Neyman–Pearson classification and CMDP tasks confirm the feasibility and stability of the proposed method.

**Weaknesses:**

1. The theory assumes convex objectives and constraints, which restricts applicability to deep federated learning scenarios. The extension to nonconvex settings is not discussed.
2. Experiments are restricted to small datasets and tabular RL tasks. Results on large-scale or nonconvex benchmarks (e.g., image classification or language models) would greatly strengthen the empirical claims.
3. The experimental section lacks strong comparisons with recent constrained FL baselines (e.g., primal-dual or penalty-based methods), making it difficult to evaluate relative performance.
4. No experiments under heterogeneous data distributions.

**Questions:**

See weaknesses above.

---

> ### Author Response · Authors · 2025-11-21
> **Response to PoVG**
>
> We thank the reviewer to acknowledge the novelty as being first work to handle the four components. Indeed the method leads to lightweight and practical approach for resource constrained FL optimization with functional constraints.
>
> >**Weakness (1)**: *Non-convexity....*
>
> We agree that limiting the analysis to convexity restricts applicability to deep learning. To address this, we have significantly expanded our theoretical analysis in the Appendix (D and E). As detailed in the General Response, we now provide a convergence proof for $\rho$-Weakly Convex objectives (a standard class of non-convex functions covering neural networks), demonstrating that $FedSGM$ is robust to non-convex setting.
>
> >**Weakness (2), (3) and (4)**: *Experimental restriction....*
>
> We appreciate the suggestion to strengthen empirical validation. We first clarify that our existing RL experiment is not tabular, but utilizes Deep Neural Network policies in a continuous environment, inherently covering the non-convex and stochastic setting along with extreme data heterogeneity. We made this point clear and explicit in the revised paper.
>
> To further address the call for heterogeneous data distributions (Weakness (4)) and strong baselines (Weakness (3)), we introduced a new fair classification experiment on ADULT dataset using a Deep NN under heterogeneous (non-IID) data distributions and stochastic gradients. Please refer to [New: Figure 2](https://imgur.com/ppeEMIt), Geneeral Response and Appendix (F).

---

> > ### Comment · Reviewer_PoVG · 2025-11-25
> >
> > I appreciate the authors' efforts during rebuttal, especially the new proofs and new experiments in Appendix.
> >
> > However, I have some follow-up questions.
> >
> > Follow-up Q1: Why does the main paper focus originally on convex analysis? What is the necessity of convex theory under your task? Do the original experiments satisfy the requirements of the convexity problem?
> >
> > Follow-up Q2: The added proof focus on the weakly convex objective. However, to my knowledge, weak convexity is not not general nonconvex. And the experiments the authors added specifically for deep neural networks satisfy the non-convex, but not weak convexity. Please correct me if I am wrong.
> >
> > After reading other reviewers' comments, I am also curious: What is the practical necessity of simultaneously handling all four components? Could the authors provide further explanation of how these four components coexist in federated learning? Are there any interdependencies among these components?

---

> > > ### Author Response · Authors · 2025-11-26
> > > **Response to Reviewer PoVG**
> > >
> > > We thank the reviewer for their positive assessment of our rebuttal extensions.
> > >
> > > >Follow-up Q1: Why does the main paper focus originally on convex analysis? What is the necessity of convex theory under your task? Do the original experiments satisfy the requirements of the convexity problem?
> > >
> > > In constrained optimization, analysis under convexity assumption serves as the starting point. Our primary aim was to develop a practical federated framework with provable theoretical guarantees that can handle functional constraints. Our analysis was guided by the intuition that if the algorithm failed to recover standard convergence rates in the convex setting, it would not be worth exploring the non-convex setting. Thus, establishing these fundamental guarantees (global optimality and feasibility) was the necessary prerequisite to validate the core mechanism of $FedSGM$, specifically, that the interplay between compression, local updates, and switching does not cause divergence.
> > >
> > > Now, to answer the question about experiments. Yes, the original experiment on NP classification utilizes binary logistic loss; hence both the objective and constraints are convex, satisfying the convexity assumption. This serves as the direct empirical verification of our main theorem. This is presented in Figure 1 of the main paper. Additionally, the RL and ADULT experiments were included to demonstrate that the algorithm performs well even when these convexity assumptions are relaxed (non-convex neural networks), highlighting practical robustness beyond the theoretical scope.
> > >
> > > >Follow-up Q2: The added proof focus on the weakly convex objective. However, to my knowledge, weak convexity is not not general nonconvex. And the experiments the authors added specifically for deep neural networks satisfy the non-convex, but not weak convexity. Please correct me if I am wrong.
> > >
> > > We would like to clarify the relationship between weak convexity and the broader notion of non-convexity found in the optimization literature. The reviewer is correct that weak convexity is a stronger condition than general non-convexity. Our theoretical results rely on weak convexity only to establish the convergence guarantees of the proposed algorithm. Importantly, weak convexity is not required for our empirical evaluation.
> > >
> > > Regarding the reviewer's concern about restrictiveness, we emphasize that weak convexity is in fact more general than the classical smoothness(Lipschitz-gradient) assumption commonly used in optimization theory. A standard result states that any function with $L-$ Lipschitz continuous gradients is automatically $L-$weakly convex. Thus, smoothness implies weak convexity, showing that weak convexity is not an overly restrictive assumption within the context of theoretical analysis.
> > >
> > > Regarding experiments, our empirical evaluation does not rely on the objective being weakly convex. As we indicated earlier, we verified our method in the convex setting (NP Classification) to confirm consistency with theory. We then presented additional experiments on highly non-convex deep neural networks to demonstrate practical behaviour in realistic settings. These deep-learning objectives are not assumed to be weakly convex, nor do we claim that modern architectures satisfy weak convexity. This separation between theoretical assumptions (for convergence proofs) and empirical evaluation (on general non-convex problems) follows standard practice in optimization research.
> > >
> > >
> > > >After reading other reviewers' comments, I am also curious: What is the practical necessity of simultaneously handling all four components? Could the authors provide further explanation of how these four components coexist in federated learning? Are there any interdependencies among these components?
> > >
> > >
> > > We thank the reviewer for asking the question about practical relevance. We argue that we are analyzing these four components in tandem not as a theoretical exercise but rather as the requirement for modern FL. The first three components of local updates, partial participation, and compression are the necessities in the field for resource constrained edge devices. A natural addition to the three components is the functional constraint.
> > >
> > > A simple yet concrete example of this practical necessity is mobile keyboard prediction, for example Gboard, across millions of devices. The literature has already established that partial participation, local updates, and compression are mandatory to handle bandwidth and latency limits. However, modern deployments also demand fairness. For instance, ensuring that the model does not degrade performance for minority language speakers requires enforcing a functional constraint (e.g. demographic parity). Our algorithm is the first to provide provable guarantees for this realistic scenario where efficiency and functional constraints must coexist.

---

### Official Review · Reviewer_BCyy · 2025-11-01

**Soundness:** 3
**Presentation:** 1
**Contribution:** 3
**Rating:** 2
**Confidence:** 3

**Summary:**

The paper proposes a unified theoretical framework for federated learning optimization that jointly captures several key aspects: constraints, partial participation, local computation, and bidirectional communication compression with error feedback. The framework generalizes many existing FL methods and, if it indeed recovers their best-known convergence rates across all settings (as claimed), it would constitute a strong theoretical contribution. The paper centers on the functional constraints and considers both hard and soft switching for satisfying them.

**Strengths:**

- The unification of these aspects of federated learning (local steps, communication compression \& error feedback, partial participation) under one framework is meaningful and relevant. If the framework recovers the best-known convergence rates for all covered scenarios of literature (as claimed), it is worth publishing on its own.

- The authors provide high-probability convergence guarantees for both hard and soft switching.

**Weaknesses:**

**Constraint formulation**: The novelty and significance of the constraint formulation seem to be overstated. Assumption 3 restricts the generality of the framework, limiting the scope of the unification.
Hard and soft switching closely resemble well-known approaches for minimizing an unconstrained regularized objective, with the regularizer R chosen as: R(w)=0 if G(w)<=\eps and \infty otherwise (see the formulation [1]). In this viewpoint, the literature review seems to be inadequate.

> [1] Condat, Laurent, and Peter Richtárik. "Murana: A generic framework for stochastic variance-reduced optimization." Mathematical and Scientific Machine Learning. PMLR, 2022.

**Practical relevance:**
Practical relevance of the unified framework is unclear: FL is a diverse field, and it is not evident which real-world FL setups require all of the newly-enabled components simultaneously.

**Experimental comparison:**
Experiments are limited and mostly illustrative. I think that there is a lot of missed potential. Having the unified framework, one can also compare the different aspects of the framework to answer fundamental questions of federated learning -- what is more important: to have more clients in partial participation, or to have more bits in the compresion schemes, or to do more local steps?

**Presentation issues:** The paper needs a major revision before considering publishing.  There are missing definitions (convexity, Lipschitz continuity), a lack of formal rigor, and a disjointed and difficult-to-follow narrative.

**Questions:**

- How is the proposed notion of hard/soft constraint switching connected to projection (or inexact projection) onto the feasible set {w:G(w)<= \eps}?

- Is there a practically significant configuration (e.g., combination of compression, partial participation, local steps, and constraints) that is new and not covered by prior FL works?

- Can the authors provide insights (both theoretical and empirical) on the relative importance of **i)** number of clients in partial participation, **ii)** number of bits in compression, and **iii)*** number of local updates?

---

> ### Author Response · Authors · 2025-11-21
> **Response to Reviewer BCyy (Part 1/2)**
>
> We thank the reviewer for recognizing that our unification of federated challenges is "meaningful, relevant, and worth publishing on its own." **The reviewer specifically noted that "the work is valuable if it recovers best-known convergence rates."** We confirm that $FedSGM$ recovers the best-known canonical optimization rates of $\boldsymbol{\mathcal{O}}\left(\frac{1}{\sqrt{T}}\right)$ while decoupling the estimation error from the high-probability analysis for all covered scenarios. Furthermore, we have significantly expanded the applicability of our framework by adding new theoretical analyses for Stochastic Gradients (SGD) and $\rho$-Weakly Convex objectives (see General Response) in the Appendix D and E respectively. These additions demonstrate that our unified framework not only recovers standard rates in convex settings but also extends robustly to modern non-convex scenarios.
>
> >**Weakness (1)**: *Constraint formulation....*
>
> We thank the reviewer for pointing out the connection to regularized optimization and the connection to MURANA paper. The reviewer's intuition is right in saying that one could construct an indicator function based on the constraint feasibility and this reformulation would lead to the mathematically equivalent optimization problem. However, this reformulation does not yield a feasible algorithm for FL as we are solving the problem of functional constraints. MURANA relies on the proximal operator of $R(w)$. As detailed in our General Response, the proximal operator for a functional constraint corresponds to the Euclidean Projection onto the feasible set. In a federated setting, computing this projection requires an iterative inner loop involving multiple rounds of vector communication per global step. Thus, while the mathematical formulations are equivalent, the proximal algorithms are communication restrictive. The novelty of $FedSGM$ lies in developing a primal-only switching framework that satisfies the constraint while avoiding the massive communication cost of the proximal operator that makes standard proximal methods inapplicable. We kindly refer the reviewer to see General Response for a more formal and extensive treatment of not only MURANA(projection) but Frank-Wolfe as well as Penalty method.
>
> **On Literature Review:** We have expanded our literature review to explicitly include the discussion in General Response and it is now present in the Appendix (G).
>
> **On Assumption 3:**
> We clarify that Assumption 3 is not a limitation of our framework, but rather the standard theoretical characterization of Client Heterogeneity in partial participation settings. In unconstrained FL, a standard assumption widely used in the analysis is bounded client dissimilarity or bounded global variance, i.e, $\mathbb{E}[\\|\nabla f_i(w) - \nabla f(w)\\|^2] \le \sigma^2$. In constrained FL with functional constraints, it is necessary to impose an analogous bound on the heterogeneity of the constraint function $g(w)$ across clients. Assumption 3, which states that $\hat{G} - g$ is sub-Gaussian, is simply the high-probability equivalent of this standard dissimilarity bound present in literature, such as FedAVG, SCAFFOLD, FedProx, etc., and applied to $g$. Without bounding this heterogeneity, estimating the global constraint $\sum g_i(w)$ from a subset of clients would be theoretically impossible (as the variance would be unbounded). Thus, this assumption defines its validity in FL problems rather than restricting the algorithm's theory.
>
>
> >**Weakness (2) and Question (2)**: *Practical relevance....*
>
>
> We thank the reviewer for asking the question about practical relevance. We argue that we are analyzing these four components in tandem not as a theoretical excercise but rather as the requirement for modern FL. The first three components of local updates, partial participation, and compression is a necessity in the field for resource constrained edge devices. A natural addition to the three components is the functional constraint. While previous works like MURANA analyzed constrained FL under set constraints, they cannot handle functional constraints required for modern ML objectives.
>
> A simple yet concrete example of this practical necessity is mobile keyboard prediction, for e.g., Gboard, across millions of devices. The literature has already established that partial participation, local updates, and compression are mandatory to handle bandwidth and latency limits. However, modern deployments also demand fairness. For instance, ensuring the model does not degrade performance for minority language speakers requires enforcing a functional constraint (e.g. demographic parity). Our algorithm is the first to provide provable guarantees for this realistic scenario where efficiency and fairness constraints must coexist.
>
> $\small(continued\ in\ the\ 2nd\ part)$

---

> > ### Author Response · Authors · 2025-11-21
> > **Response to Reviewer BCyy (Part 2/2)**
> >
> > >**Weakness (3) and Question (3)**: *Experimental comparison and theoretical insight....*
> >
> > We thank the reviewer for this interesting perspective on the fundamental trade-offs. We view these factors through the lens of System Constraints vs. Algorithmic Control:
> >
> > 1. System Constraints (Participation & Compression): Although we control these with hyperparametrs, these are largely dictated by the system or environment; for example, client availability and communication bottleneck. In our case, with functional constraints, low participation induces a noise floor that slows the convergence. Our experiments (see NP CLassification Figure 2: Row 2) confirm that reducing participation causes frequent violation of the constraints while hindering the convergence.
> >
> > While compression is also a system constraint, our results show the algorithm is robust to it. Thanks to error feedback where noise is corrected over time rather than getting accumulated, we can handle aggressive compression with minimal performance loss, as observed in Figure 2: Row 3.
> >
> > 2. Algorithmic Control (Local Updates): Local updates (E) is a hyperparameter that can be dictated by the server. In our theoretical guarantees for both optimality and feasibility, we have $\sqrt{E}$ in the upper bound, capturing the effect of client drift. So, increasing $E$ aggresively can harm more where the "Client Drift" error outweighs the communication savings, destabilizing the constraint satisfaction. We clearly observed this trade-off in our experiments see Figure 2: Row 1.
> >
> > >**Weakness (4)**: *Presentation....*
> >
> > We sincerely thank the reviewer for holding our manuscript to a high standard. We have now revised the paper to address these concerns. We have now added the definition of convexity and Lipschtness in Assumption 1. Furthermore, we have added the discussion presented in General Response as part of the related work in Appendix (G) incorporating reviewers questions about MURANA. We believe these changes address the presentation concerns. We welcome specific pointers on any other concerns during the discussion period.

---

### Official Review · Reviewer_RqPZ · 2025-11-03

**Soundness:** 3
**Presentation:** 3
**Contribution:** 3
**Rating:** 4
**Confidence:** 3

**Summary:**

This paper introduces FEDSGM, a unified framework for federated constrained optimization designed to simultaneously address four major challenges: functional constraints, communication bottlenecks, multiple local updates, and partial client participation. Proposed approach is built on the Switching Gradient Method (SGM). FEDSGM provides projection-free, primal-only updates, avoiding the need for expensive dual-variable tuning. To manage communication limits, it incorporates bidirectional error feedback to correct bias from compression, and its theoretical analysis explicitly models the interaction between this compression noise and the client drift from multiple local steps. The authors derive convergence guarantees showing the averaged iterate achieves the canonical $\mathcal{O}(1/\sqrt{T})$ rate, with high-probability bounds that isolate the sampling noise from partial participation. Additionally, a "soft switching" variant is proposed to stabilize updates near the feasibility boundary. The framework's efficacy is validated empirically on Neyman-Pearson classification and constrained Markov decision process (CMDP) tasks.

**Strengths:**

1. The paper provides a rigorous and extensive theoretical analysis of its framework.
2. Well-motivated problems and applications.

**Weaknesses:**

1. Lack of Empirical Validation: The authors have not compared their proposed method with the state-of-the-art. The experiments (Section 4) only compare FEDSGM against a "Centralized" (i.e., non-federated) version of itself which is an ablation study.

2. It demonstrates that the federated setting introduces a performance cost (which is expected) but tells us nothing about whether FEDSGM is better than any other existing method.

3. Communication Efficiency:  In this paper, the authors does not propose a new communication-efficient technique. It merely "incorporates" standard  methods like Top-K compression and Error Feedback. Moreover, their method added new communication overhead. Algorithm 1 explicitly requires a "Constraint query" (lines 3-4) where all clients send their constraint value $g_j(w_t)$ to the server, which then aggregates and broadcasts the switching decision. This is a full, separate communication round-trip that must happen before the main gradient/model update. This added synchronisation step directly contradicts the goal of reducing communication bottlenecks.

**Questions:**

1. I want to see a detailed comparative analysis with SOTA (e.g., Federated Frank-Wolfe, and other projection-free methods ) to prove the superiority of the proposed approach.

---

> ### Author Response · Authors · 2025-11-21
> **Response to Reviewer RqPZ**
>
> We would like to thank the reviewer RqPZ and are encouraged that they found our work rigorous and extensive. We acknowledge that this problem is extremely timely and well motivated, particularly given the increasing need to balance performance with fairness and safety constraints in modern distributed learning systems.
>
>
> >**Weakness (1) and Question (1)**: *Empirical validation and baselines...*
>
> We have addressed the reviewer's concern on comparative analysis as follows:
>
> 1. **Infeasibility of Frank-Wolfe/Projection Baselines:** As detailed in our General Response, methods like Federated Frank-Wolfe or projected/proximal methods relies on solving optimization over a feasible set. For functional constraints defined in the distributed setting ($g(w) = \frac{1}{n}\sum g_i(w) \le 0$), solving over a feaseible set requires an iterative subroutine where multiple inner iterations require multiple communication rounds with the server. Thus, Frank Wolfe is communication prohibitive for functional constraints, rendering a direct runtime comparison unfair.
>
> 2. **Comparison with Penalty-Based Baselines:** The most viable alternative is a penalty-based approach (e.g., applying FedAvg on $f(x) + \rho [g(x)]_+$). We have added a comparison against Penalty-FedAvg as in [New: Figure 1](https://imgur.com/HPhwHK9). The results demonstrate that penalty scheduling is a heuristic task without theoretical backing in this setting. As shown in the figure, small penalties yield infeasible solutions. Conversely, in order to emulate the theoretical hard constraint requires setting $\rho$ very large, which renders the problem ill-conditioned. This failure mode is explicitly visible in [New: Figure 1](https://imgur.com/HPhwHK9) with the poor convergence for $\rho=100$. $FedSGM$ satisfies constraints without this manual tuning and the instability followed by it.
>
> To further strengthen our claim, we run additional experiments on ADULT dataset with SGD and heterogeneous data distribution, and compared the result against various centralized benchmarks (serving as the empirical best) to solve functional constrained problem.
>
>
> 3. **Comparison with Safe-EF:** We clarify that Safe-EF is a special case of our framework under $E=1$ and full participation. Our ablation studies effectively cover this regime (comparing $E=1$ vs $E>1$), showing that our framework generalizes the Safe-EF to the more practical federated setting with partial participation and local updates, while refining their theoretical analysis.
>
> >**Weakness (2)**: *Comparison against other methods...*
>
>
> We emphasize that $FedSGM$ is the first work to simultaneously address functional constraints, partial participation, local updates, and compression. As detailed in the General Response, existing methods in the literature (such as Federated Frank-Wolfe or Projection-based methods) are either mathematically incompatible with this setting or require prohibitive communication overhead (inner loops) to solve the functional constraints. Thus, a direct comparison to "existing methods" in the literature, which cannnot solve Problem (1), is not feasible.
>
> To address the need for extra comparison, we constructed the Penalty-FedAvg baseline (see [New: Figure 1](https://imgur.com/HPhwHK9)), which represents the closest adaptation of existing literature to this setting. The results confirm $FedSGM's$ superiority over this adapted baseline. Additionally, we conducted an additional experiment on the ADULT dataset with data heterogeneity and SGD, validating that $FedSGM$ discovers feasible solutions in realistic settings and comparing against various other centralized baseline.
>
>
> >**Weakness (3)**: *Communication Efficiency....*
>
> We acknowledge that $FedSGM$ does not introduce a new compression mechanism. Our contribution, however, is a rigorous theoretical analysis of functional constraints in resource constrained FL setting. The novelty lies in the unification and proof that the four key challenges can coexist while maintaining convergence guarantees.
>
> Regarding the "Constraint Query" overhead, we emphasize the distinction between Scalar Communication and Model Communication. The constraint query involves transmitting a single scalar (4 bytes) per client. This bandwidth cost is negligible compared to the model updates (megabytes/gigabytes) that we compress.
> While this adds a synchronization step, it avoids the far greater cost of alternative methods. As described in the General Response, solving a functional constraint, for example by Frank-Wolfe, requires an iterative inner loop. This implies multiple rounds of vector communication per global step. Meanwhile, the overall bottleneck is drastically reduced by the use of bidirectional compression and multi-step local updates ($E > 1$), which far outweigh the minor cost of the constraint check involving scalar communication.

---

### Official Review · Reviewer_yKoE · 2025-11-04

**Soundness:** 3
**Presentation:** 3
**Contribution:** 3
**Rating:** 6
**Confidence:** 3

**Summary:**

The paper introduces a federated algorithm for IV analysis (FEDIV) via federated GMM (FEDGMM). It formulates FEDGMM as a federated zero-sum game defined by a non-convex non-concave minimax optimization problem. It also shows that the solutions to the federated game satisfies Stackelberg equilibrium satisfying client-local equilibria up to a heterogeneity bias. The work is the first work on federated IV using federated GMM.

**Strengths:**

1. The work is the first work formulating federated IV via federated GMM.
2. The solid theoretical results strengthen the work.
3. The experimental results shows that the federated IV analysis framework is efficient in recovering the GMM estimators.

**Weaknesses:**

The experimental results in the main body of the paper are limited. The authors could move some of the results from the appendix for the final version of the draft.

**Questions:**

In Table 1, when using GDA, Fed-DeepGMM performs closer to the DeepGMM compared to when SGDA is used for both. Is there an intuitive reason behind this observation?

---

> ### Author Response · Authors · 2025-11-21
> **Responses to Reviewer ykoE**
>
> >**Weakness (1,2) & Question (2)**: *Limited theory, GD, and non-convexity...*
>
> Please refer to the **Reformulation and Strengthened Theoretical Analysis** in the general response.
>
> >**Weakness (3)**: *Fixed partial participation...*
>
> We respectfully disagree with this point. We would like to clarify that assuming a fixed participation size $m$ is a notational convention used in FL analysis, (see [1], [2], and [3]). Extending this to variable participation ($m_t$) is complementary to our core contribution of solving functional constraints. Our theoretical framework naturally supports this extension by simply bounding the sampling noise with the minimum participation count ($m_{\min}$), without altering the convergence rate order.
>
> References:
>
> 1.  Karimireddy et al., "SCAFFOLD: Stochastic Controlled Averaging for Federated Learning" (ICML 2020).
> 2.  Li et al., "Federated Optimization in Heterogeneous Networks" (MLSys 2020).
> 3.  Reddi et al., "Adaptive Federated Optimization" (ICLR 2021).
>
> >**Weakness (4)**: *Limited experiments...*
>
> We acknowledge that our initial evaluation focused on Centralized methods. This choice was driven by the fact that no existing FL algorithm supports general functional constraints combined with partial participation and compression (please refer to our General Response). As we explained in the General Response, this is the first method to deal with functional constraints in FL and comparing against projection based, penalty based, and Frank-Wolfe methods may not be relevant. To show that, we conducted an additional experiment on NP Classification and compared against Penalty-FedAVG which is a heuristical method without any theoretical guarantee. The results for that is present in [New: Figure 1](https://imgur.com/HPhwHK9). Here we show that penalty scheduling is a tedious task without any theoretical ground. If we try to tune the penalty coefficient to emulate the behaviour of hard switching, the problem becomes ill-conditioned as shown in the figure.
>
> To further strengthen our claim, we run additional experiments on ADULT dataset with SGD and heterogeneous data distribution and compared the result against various centralized benchmarks to solve a fair classification problem.
>
>
> >**Question (1)**: *Contribution of the work...*
>
> We thank the reviewer for asking this question. The core deliverable of this work is to unlock the potential for realistic edge deployment with safety constraints. We showcase this in our experiments with Cartpole and NP classification as well. We provide a provable framework for Safe-FL on resource constrained edge. Prior to this work, one could have Safety (Constrained FL) or Efficiency (Compressed/Partial FL), but not both at the same time. Our work is the first to prove that safe, fair algorithms can be deployed on bandwidth-limited, unreliable edge networks. Furthermore, with the new theoretical results on SGD and weak convexity, we have significantly increased the practicality of our framework.

---

> ### Comment · Reviewer_yKoE · 2025-11-25
> **Official Comment**
>
> I would like to thank the authors for their response. I have read the comments and decide to keep my score. Although the work has some contributions, it still needs to be improved, e.g. in experimental results to fully support the theoretical results. Authors could also address my other comments as well. Therefore, I will keep my current score.

---

> > ### Author Response · Authors · 2025-11-26
> > **Response to Reviewer ykoE**
> >
> > We thank the reviewer for the follow-up. To ensure clarity and demonstrate that no "other comments" were left unaddressed, we provide the following point-by-point summary of revisions mapping your specific requests to our exact responses and changes in the paper:
> >
> > > The theoretical results are limited to when both objective functions as well as the constraints are convex, which strongly limits their applicability.
> >
> > We added a theorem to the **General Response part 3**, extending our analysis to weakly convex objectives, following the standard in analyzing non-convex non-smooth scenarios.
> >
> > > The results are obtained for when gradient descent (GD) is used for clients' local updates, which makes it more limited.
> >
> > We added a theorem to the **General Response part 3**, providing guarantees for Stochastic Gradient Descent (SGD).
> >
> > > Although the work considers partial participation, the number of participating clients in each round is fixed to $m$, which highlights another limitation.
> >
> > We clarified that this is a standard notation convention (referencing SCAFFOLD, FedProx, FedOpt). Our analysis holds for variable participation by bounding with $m_{\min}$. Please find the detail in our original response.
> >
> > > The experimental results in the main body of the paper are limited. The proposed approach is evaluated against only a centralized method (no FL).
> >
> > We added a comparison against **Penalty-FedAvg** (a heuristic FL baseline) on the NP Classification task ([New: Figure 1](https://imgur.com/HPhwHK9)) and ADULT dataset ([New: Figure 2](https://imgur.com/ppeEMIt)). Results show FedSGM outperforms the heuristic baseline significantly. As stated previously, please note that there are no existing FL baselines that handle general functional constraints, and Penalty-FedAvg was implemented specifically to address your request. In ADULT experiments ([New: Figure 2](https://imgur.com/ppeEMIt)), we compared our result against three SOTA centralized baselines, which serves as the empirical best.
> >
> > > What is the most valuable deliverable of this work? Is it the developed theory for when we have all the compression, partial participation, constraints on objectives and multiple local updates? If so, despite being valuable, the applicability of the results is still limited.
> >
> > The core deliverable of this work is to unlock the potential for realistic edge deployment with safety constraints. Our work is the first to provide a safe, fair algorithm that can be deployed on bandwidth-limited, unreliable edge networks with provable global optimality and feasibility guarantees.
> >
> > > Introducing non-convexity and SGD will make the analysis harder, but more applicable. Im wondering if the authors have explored these directions? and what is the difficulty?
> >
> > We accepted this challenge and have successfully completed the analysis. We added two theorems in the **General Response part 3** with rigorous proofs handling the complexity of stochastic gradients and weakly convex objectives. Additionally, we conducted new fair classification experiment on ADULT dataset ([New: Figure 2](https://imgur.com/ppeEMIt)). The primary challenge in this setting is the absence of unique minimizer which is not guaranteed in non-convex settings. We overcame this by utilizing classical Moreau envelope based analysis for weakly convex functions.
> >
> > > **New comment:** I would like to thank the authors for their response. I have read the comments and decide to keep my score. Although the work has some contributions, it still needs to be improved, e.g. in experimental results to fully support the theoretical results. Authors could also address my other comments as well. Therefore, I will keep my current score.
> >
> > We ask the reviewer for clarification on what the reviewer means about **experimental results to fully support the theoretical results.** For our NP Classification experiment (Figure 1 in the main paper), both the objective and the constraint are convex, and the results fully supports the theoretical claims made in the main paper. Similarly, in Figure 2 (in the main paper), the ablation study shows the impact of each component of our theoretical result, and we explained in detail in **Discussion: NP Classification** in page 7 of the main paper.
> >
> > Additionally, to show the robustness of our algorithm in general deep learning scenarios, we tested on RL and ADULT dataset experiments. These deep-learning objectives are not assumed to be weakly convex. This separation between theoretical assumptions (for convergence proofs) and empirical evaluation (on general non-convex problems) follows the standard practice in optimization research.
> >
> > We respectfully request that the reviewer specify which **remaining points they feel were not addressed**. We believe we have comprehensively resolved every concern raised in the initial review, specifically through the addition of new experiments and theoretical proofs.

---

### Author Response · Authors · 2025-11-21
**General Response - Comparison SOTA & Others (Part 1/3)**

We sincerely thank all the reviewers and the Area Chair for their time and constructive feedback. We are encouraged that reviewers found our work "rigorous, extensive and well-motivated" (RqPZ), "meaningful and relevant, it is worth publishing on its own" (BCyy), and "rigorous, well-written and structured" (Q7y9).

Several concerns revolve around the novelty of the constraint formulation, the relationship to projection or penalty-based methods, and the absence of comparisons to certain baselines. We address all of these points together below, as they stem from a single underlying issue: **$FedSGM$ is the first method that simultaneously and provably addresses functional constraints, multi-step local updates, bi-directional compression with error feedback, and partial participation in federated learning.** Existing approaches do not apply to this setting, which is why direct comparisons to them are neither feasible nor relevant.

To examine the issue at its root and recognize the novelty of our method, we first clarify the difference between **Set Constraints** and **Functional Constraints**. Standard set constrained optimization often assumes a "simple" constraint set $\boldsymbol{\mathcal{X}}$ (e.g., a box or ball) where projection is computationally cheap. However, in our work we address functional constraints of the form $g(w) \leq 0$ like the loss function in our NP-classification experiment.

We will go into the details of existing methods brought on by the reviewers, specifically **projection, penalty, and Frank-Wolfe method.** Please refer to individual sections for detailed analysis.

>**Why $FedSGM$ is not comparable to projected or proximal methods?**

The reviewers correctly note that our constrained problem (Problem 1),
$$\min_w f(w) \quad \text{s.t.} \quad g(w) \le 0,$$
is equivalent to an unconstrained problem with an indicator function as a regularizer (with $\epsilon-$ feasibility satisfaction):
$$\min_w f(w) + R(w), \quad \text{where} \quad R(w) = \begin{cases} 0 & \text{if } g(w) \le \epsilon \\\\ \infty & \text{otherwise} \end{cases}.$$

The standard class of algorithms for this formulation is Proximal Gradient Descent (PGD), as mentioned in the MURANA paper [1]. A PGD step would be:

$$w_{t+1} = \text{prox}_{\eta R}(w_t - \eta \nabla f(w_t)).$$

However, the bottleneck of this computation is the proximal operator, $\text{prox}_{\eta R}(v)$. By definition, this is

$$\text{prox}_{\eta R}(v) = \underset{w}{\operatorname{argmin}} \left(\eta R(w) + \frac{1}{2} \\|w - v\\|^2\right) = \underset{\{w \text{ s.t. } g(w) \le \epsilon\}}{\operatorname{argmin}} \\|w - v\\|^2.\   \ \ \ \ \ \ (Projection)$$

This is the Euclidean projection onto the feasible set $\boldsymbol{\mathcal{W}}^g_{\epsilon} = \\{w \mid g(w) \le \epsilon\\}$. In case of full participation, the server can compute $g$ by doing the inner solve where it asks the clients to send $g_j(w)$ multiple times (ours only require one constraint evaluation and communication) to solve $(Projection)$. For a general, non-linear convex function $g(w)$, this projection does not have a closed-form solution. Rather, it is a complex constrained optimization problem in itself that must be solved in every iteration and requires multiple communication rounds. The MURANA paper [1] fundamentally assumes that the proximal operator of the regularizer $R$ is "easy to compute". It is evident in their experimental setup as well, where they make $R =0$. This assumption is precisely what is violated in our problem setting.

It is the motivation of our paper to develop a method that avoids this intractable projection. Our ${FedSGM}$ framework is explicitly projection-free and primal-only.

$\small(continued\ in\ the\ 2nd\ part)$

---

> ### Author Response · Authors · 2025-11-21
> **General Response - Comparison SOTA & Others (Part 2/3)**
>
> >**Why penalty-based FedAvg is not an appropriate baseline?**
>
> Consider the following penalty-based formulation instead of our problem (1) in the paper,
> $$\min_{w} f(w)+\rho[g(w) - \epsilon]_+ \ \ \ \ (Penalty)$$
>
> It is evident that when $g(w) \leq \epsilon$ the (sub-)gradient is $\nabla f(w)$ while when $g(w) > \epsilon$ the (sub-)gradient is $\nabla f(w) + \rho \nabla g(w)$. Thus, for large $\rho$ such that $\nabla f(w) + \rho \nabla g(w) \approx \rho \nabla g(w)$, the penalty formulation in $(Penalty)$ would be a simplified approximation of the SGM dynamics. To precisely emulate the dynamics of hard switching, we must set $\rho$ to be very large. However, making $\rho$ large induces ill-conditioning, slowing optimization.
>
> On the other hand, any fixed $\rho$ that is too small can leave persistent violations. The "right" $\rho$ depends on the relative scale and geometry of $\nabla f$ and $\nabla g$. ${FedSGM}$ avoids this hyperparameter tuning by **switching directions** (objective vs. constraint), i.e., no penalty weight to tune, no conditioning blow-up. Therefore, this penalty-based FedAvg is not an appropriate baseline.
>
> However, to elucidate the above points we create the heuristics to run a penalty based federated algorithm where in each round the clients update their gradient as
>
> $$\nu_{j,\tau}^t = \nabla f_j(w_{j,\tau}^t) + \mathbb{I}{\\{\hat{G}(w_t) > \epsilon\\}}  \rho \nabla g_j(w_{j,\tau}^t).$$
>
> We compare ${FedSGM}$ against the "Penalty-FedAVG" with varying $\rho \in [0.001, 0.5, 2, 20, 100]$. We present the result in [New: Figure 1](https://imgur.com/HPhwHK9). The result confirms that $\rho$ has a significant impact on performance as we expected. A small $\rho$ such as $0.001$ or $0.5$ does not satisfy the feasibility criteria, while a large $\rho = 100$ (which essentially should have been indicative of switching dynamics) makes the problem ill-conditioned and gives low performance.
>
>
> >**Why additional comparison with Frank-Wolfe is not suitable?**
>
>
> **Frank-Wolfe** requires access to a Linear Minimization Oracle (LMO) over the constraint set at each iteration, formally written as,
> $$s_t = {\displaystyle\arg\min}_{s \in \boldsymbol{\mathcal{X}}} \langle s, \nabla f(w_t) \rangle. \ \ \ \ (Frank-Wolfe)$$
> While Frank-Wolfe is efficient for "simple" set constraints such as a polytope, box, or simplex, for general non-smooth convex functional constraints $g(w) \leq 0$, solving the LMO is computationally infeasible. For our case in FL where $g(w) = \frac{1}{n}\sum g_i(w)$, the constraint boundary is defined by distributed clients. Therefore, solving the LMO would require an iterative subroutine (like projected gradient descent), where every inner iteration requires a full communication round. Thus, just like projection-based methods, a Frank-Wolfe approach is communication-prohibitive in the functional constraint regime.
>
> As no existing method in FL supports functional constraints with partial participation, local updates, and compression with error feedback simultaneously, a direct comparison is not feasible. Our work bridges this gap, by characterizing the interplay between factors that are well motivated in federated learning, establishing realistic, functionally constrained framework.
>
> >**Fair classification with SGD, Deep NN and heterogeneous data distribution**
>
> To further address the call for additional experiment with strong baselines, and heterogeneous data distributions, we introduce a new fair classification experiment on ADULT dataset using a Deep NN under heterogeneous (non-IID) data distributions and stochastic gradients. Please refer to [New: Figure2](https://imgur.com/ppeEMIt).
>
> By explicitly comparing against Penalty-FedAvg and IPP-ConEx (Centralized SOTA, [1]), we demonstrate that $FedSGM$ achieves the best trade-off between objective and constraint satisfaction, avoiding the instability from penalty tuning. Full details of the experiments are provided in Appendix (F).
>
> References:
>
> 1.  Boob et al., "Stochastic first-order methods for convex and nonconvex functional constrained optimization" (Mathematical Programming 2022).
>
> $\small(continued\ in\ the\ 3rd\ part)$

---

> > ### Author Response · Authors · 2025-11-21
> > **General Response - Comparison SOTA & Others (Part 3/3)**
> >
> > >**Refined Formulation and Strengthened Theoretical Analysis**
> >
> > We are grateful to the reviewers highlighting the references using constraint set geometry. Based on this, we refined our problem (1) formulation in the paper to explicitly minimize over compact convex set ${\mathcal{X}}$, while satisfying functional constraint $g(w) \leq 0$. Formally, we define it as
> > $$w^* = \underset{w \in \mathcal{X}}{\operatorname{argmin}} \\{f(w) := \frac{1}{n}\sum_{j=1}^n f_j(w) \text{ s.t. } g(w) := \frac{1}{n}\sum_{j=1}^n g_j(w) \leq 0\\}.$$
> >
> > Acccordingly, we adapted the main theorems in the paper while maintaining the canonical convergence rate of ${\mathcal{O}\left(\frac{1}{\epsilon^2}\right)}$.
> >
> > Moreover, we agree with reviewer that extending the analysis to stochastic data (SGD) and non-convex settings significantly improves the applicability of our framework to modern deep learning scenarios. Inspired by the reviewer's suggestion, we have added two new theorems in the Appendix(D and E).
> >
> > Firstly, we analyzed **$FedSGM$ under stochastic gradient noise** and, crucially, stochastic constraint evaluation. The primary challenge here is the interaction between noise and switching. In the constrained setting, stochasticity in the constraint evaluation ($g(w)$) comes from two sources, data sampling(SGD) as well as client sampling(partial participation). This creates a "noisy switching" process where the algorithm might minimize the wrong objective (optimizing $f$ when infeasible, or $g$ when feasible). We resolved this by establishing concentration bounds, proving that these erroneous switchings are bounded and do not derail convergence. Please refer to Appendix (D) in the paper for the complete proof. Now, we present the statement of the theorem.
> >
> > > **Theorem (Stochastic Constraints & Gradients):**
> >
> > > Consider the optimization problem in (1) and **Algorithm 1** under **Assumptions 1, 2, and 5**. Additionally, assume that in each global round, $m$ distinct clients are sampled uniformly at random. Define $\tilde{G}(w_t):=\frac{1}{m}\sum_{j \in {\mathcal{S}}_t} g_j(w_t,\xi_t^j)$,
> > >
> > >$$\mathcal{A} := \left\\{ t \in [T] \middle|\tilde{G}(w_t) \leq \epsilon \right\\}, \qquad \bar{w} := \frac{1}{|{\mathcal{A}}|} \sum_{t \in {\mathcal{A}}} w_t. $$
> > > Suppose the step size $\eta$ and constraint threshold $\epsilon$ are set as $$\eta = \sqrt{\frac{D^2}{2G_{eff} E T}} \qquad \text{and }$$
> > $$\epsilon = \sqrt{\frac{{2 D^2 G_{eff}}}{{E T}}} + 4 GD \sqrt{\frac{2}{{mT}}\log\left(\frac{7}{\delta}\right)} + 2 D \sigma_\xi \sqrt{\frac{2}{mET}\log\left(\frac{7}{\delta}\right)} + 2\sqrt{\frac{2}{m}\left({\sigma^2} + \frac{\sigma^2_b}{N_b}\right) \log\left(\frac{14T}{\delta}\right)},$$
> > where $$ G_{eff} = 3E^2 G^2 + E^{2} \sigma_{\xi}^{2} \left(1 + \log\left(\frac{7 n T E}{\delta}\right)\right)+ \frac{4}{m}(EG^{2}+\sigma_{\xi}^{2}) \log\left(\frac{28 T}{\delta}\right)$$ and $\delta \in (0,1)$ is a confidence parameter. Then, with probability at least $1-\delta$, the following hold:
> > > 1. The set ${\mathcal{A}}$ is non-empty, so $\bar{w}$ is well-defined.
> > > 2. The average iterate $\bar{w}$ satisfies $$f(\bar{w}) - f(w^*) \leq \epsilon, \qquad g(\bar{w}) \leq \epsilon.$$
> >
> > Additionally, we have also added a convergence result for **$\rho$-weakly convex objectives** with convex constraints under partial participation and local updates. This covers a broad class of non-convex problems standard in neural network training. The primary challenge in this setting is the absence of unique minimizer which is not guaranteed in non-convex settings. We overcame this by utilizing classical Moreau envelope based analysis for weakly convex functions. Please refer to Appendix (E) for the complete proof. Now, we present the statement of the theorem.
> >
> > > **Theorem (FedSGM: Weakly Convex $f$, Convex $g$):**
> >
> > > Consider the optimization problem in **(1)** under **Assumptions 2, 4, and 6-9**. Define $\hat{G}(w_{t}):=\frac{1}{m}\sum_{j \in \{\mathcal{S}}_t} g_j(w_t)$,
> > >
> > > $$\mathcal{A} := \left\\{t\in [T] \middle|\hat{G}(w_{t})\leq \epsilon\right\\},$$
> > >and set step-size $\eta$ and constraint threshold $\epsilon$ appropriately:$$
> > > \eta = \sqrt{\frac{D^2}{4G^2 E^3 T}} \quad \text{and}$$
> > > $$\epsilon = \sqrt{\frac{{4 D^2 G^2 E}}{{T}}} + \frac{D^2 \rho}{8 E T} + \frac{4GD}{\sqrt{mT}}\sqrt{2\log\left(\frac{4}{\delta}\right)}+ 2\sigma\sqrt{\frac{2}{m} \log\left(\frac{8T}{\delta}\right)}(1 + \Lambda), $$
> > >
> > > where $\Lambda$ is the bound on the Lagrange multiplier and $\delta \in (0,1)$ is a confidence parameter. Then, with probability at least $1 -\delta$, the following hold:
> > > 1. The set ${\mathcal{A}}$ is non-empty, so $w_t$ for $t \in {\mathcal{A}}$ is well-defined.
> > > 2. The following bounds hold for the average iterate:$$\frac{1}{|{\mathcal{A}}|} \sum_{t \in {\mathcal{A}}}\\|w_t - \hat{w}(w_t)\\| \leq \sqrt{\frac{(1+\Lambda)\epsilon}{(\hat{\rho} - 2\rho}}\quad \frac{1}{|{\mathcal{A}}|} \sum_{t \in {\mathcal{A}}}g(w_t) \leq \epsilon.$$

---

### Meta-Review · Area_Chair_6tLD · 2026-01-06

**Summary:**

Some of the major concerns include
1. Limited experiments
2. Techinques used for communication efficiency are not new
3. Limited practical relevance
4. assumption of convexity is restrictive
5. Merging several existing techniques has limited technical novelty

**Reviewer Concerns:**

Limited novelty (combining several existing techniques) and the restrictive assumption of convexity have not been sufficiently addressed.

**Reviewer Scores:**

unchanged

---

### Decision · Program_Chairs · 2026-01-26

Reject